# Independent mechanism analysis, a new concept?

**Luigi Gresele**[*1]     **Julius von Kügelgen**[*1,2]     **Vincent Stimper** [1,2]

**Bernhard Schölkopf** [1]     **Michel Besserve** [1]

[1] Max Planck Institute for Intelligent Systems, Tübingen, Germany     [2] University of Cambridge
{luigi.gresele,jvk,vincent.stimper,bs,besserve}@tue.mpg.de

## Abstract

Independent component analysis provides a principled framework for unsupervised representation learning, with solid theory on the identifiability of the latent code that generated the data, given only observations of mixtures thereof. Unfortunately, when the mixing is nonlinear, the model is provably nonidentifiable, since statistical independence alone does not sufficiently constrain the problem. Identifiability can be recovered in settings where additional, typically observed variables are included in the generative process. We investigate an alternative path and consider instead including assumptions reflecting the principle of *independent causal mechanisms* exploited in the field of causality. Specifically, our approach is motivated by thinking of each source as independently influencing the mixing process. This gives rise to a framework which we term independent mechanism analysis. We provide theoretical and empirical evidence that our approach circumvents a number of nonidentifiability issues arising in nonlinear blind source separation.

## 1   Introduction

One of the goals of unsupervised learning is to uncover properties of the data generating process, such as latent structures giving rise to the observed data. Identifiability [55] formalises this desideratum: under suitable assumptions, a model learnt from observations should match the ground truth, up to well-defined ambiguities. Within representation learning, identifiability has been studied mostly in the context of independent component analysis (ICA) [17, 40], which assumes that the observed data $\mathbf{x}$ results from mixing unobserved *independent* random variables $s_i$ referred to as *sources*. The aim is to recover the sources based on the observed mixtures alone, also termed *blind source separation* (BSS). A major obstacle to BSS is that, in the nonlinear case, independent component estimation does not necessarily correspond to recovering the *true* sources: it is possible to give counterexamples where the observations are transformed into components $y_i$ which are independent, yet still mixed with respect to the true sources $s_i$ [20, 39, 98]. In other words, nonlinear ICA is not identifiable.

In order to achieve identifiability, a growing body of research postulates additional supervision or structure in the data generating process, often in the form of *auxiliary variables* [28, 30, 37, 38, 41]. In the present work, we investigate a different route to identifiability by drawing inspiration from the field of *causal inference* [71, 78] which has provided useful insights for a number of machine learning tasks, including semi-supervised [87, 103], transfer [6, 23, 27, 31, 61, 72, 84, 85, 97, 102, 107], reinforcement [7, 14, 22, 26, 53, 59, 60, 106], and unsupervised [9, 10, 54, 70, 88, 91, 104, 105] learning. To this end, we *interpret the ICA mixing as a causal process* and apply the principle of independent causal mechanisms (ICM) which postulates that the generative process consists of independent modules which do not share information [43, 78, 87]. In this context, "independent" does not refer to *statistical* independence of random variables, but rather to the notion that the distributions and functions composing the generative process are chosen independently by Nature [43, 48]. While a formalisation of ICM [43, 57] in terms of algorithmic (Kolmogorov) complexity [51] exists, it is not computable, and hence applying ICM in practice requires assessing such non-statistical independence

---

[*]Equal contribution. Code available at: https://github.com/lgresele/independent-mechanism-analysis

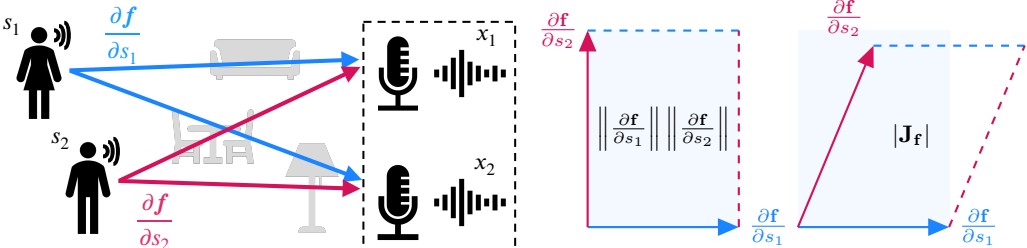

Figure 1: *(Left)* For the cocktail party problem, the ICM principle *as traditionally understood* would say that the content of speech $p_{\mathbf{s}}$ is independent of the mixing or recording process $\mathbf{f}$ (microphone placement, room acoustics). IMA refines, or extends, this idea *at the level of the mixing function* by postulating that the contributions $\partial\mathbf{f}/\partial s_i$ of each source to $\mathbf{f}$, as captured by the speakers' positions relative to the recording process, should not be fine-tuned to each other. *(Right)* We formalise this independence between the $\partial\mathbf{f}/\partial s_i$, which are the columns of the Jacobian $\mathbf{J_f}$, as an *orthogonality condition*: the absolute value of the determinant $|\mathbf{J_f}|$, i.e., the volume of the parallelepiped spanned by $\partial\mathbf{f}/\partial s_i$, should decompose as the product of the norms of the $\partial\mathbf{f}/\partial s_i$.

with suitable domain specific criteria [96]. The goal of our work is thus to *constrain the nonlinear ICA problem, in particular the mixing function, via suitable ICM measures*, thereby ruling out common counterexamples to identifiability which intuitively violate the ICM principle.

Traditionally, ICM criteria have been developed for causal discovery, where *both cause and effect are observed* [18, 45, 46, 110]. They enforce an independence between (i) the cause (source) distribution and (ii) the conditional or mechanism (mixing function) generating the effect (observations), and thus rely on the fact that the *observed* cause distribution is informative. As we will show, this renders them insufficient for nonlinear ICA, since the constraints they impose are satisfied by common counterexamples to identifiability. With this in mind, we introduce a new way to characterise or *refine* the ICM principle for unsupervised representation learning tasks such as nonlinear ICA.

**Motivating example.** To build intuition, we turn to a famous example of ICA and BSS: the cocktail party problem, illustrated in Fig. 1 *(Left)*. Here, a number of conversations are happening in parallel, and the task is to recover the individual voices $s_i$ from the recorded mixtures $x_i$. The mixing or recording process $\mathbf{f}$ is primarily determined by the room acoustics and the locations at which microphones are placed. Moreover, each speaker influences the recording through their positioning in the room, and we may think of this influence as $\partial\mathbf{f}/\partial s_i$. Our independence postulate then amounts to stating that the speakers' positions are not fine-tuned to the room acoustics and microphone placement, or to each other, i.e., *the contributions $\partial\mathbf{f}/\partial s_i$ should be independent (in a non-statistical sense).*[1]

**Our approach.** We formalise this notion of independence between the contributions $\partial\mathbf{f}/\partial s_i$ of each source to the mixing process (i.e., the columns of the Jacobian matrix $\mathbf{J_f}$ of partial derivatives) as an orthogonality condition, see Fig. 1 *(Right)*. Specifically, the absolute value of the determinant $|\mathbf{J_f}|$, which describes the local change in infinitesimal volume induced by mixing the sources, should factorise or decompose as the product of the norms of its columns. This can be seen as a decoupling of the local influence of each partial derivative in the pushforward operation (mixing function) mapping the source distribution to the observed one, and gives rise to a novel framework which we term independent mechanism analysis (IMA). IMA can be understood as a refinement of the ICM principle that applies the idea of independence of mechanisms at the level of the mixing function.

**Contributions.** The structure and contributions of this paper can be summarised as follows:

- we review well-known obstacles to identifiability of nonlinear ICA (§ 2.1), as well as existing ICM criteria (§ 2.2), and show that the latter do not sufficiently constrain nonlinear ICA (§ 3);

- we propose a more suitable ICM criterion for unsupervised representation learning which gives rise to a new framework that we term independent mechanism analysis (IMA) (§ 4); we provide geometric and information-theoretic interpretations of IMA (§ 4.1), introduce an IMA contrast function which is invariant to the inherent ambiguities of nonlinear ICA (§ 4.2), and show that it rules out a large class of counterexamples and is consistent with existing identifiability results (§ 4.3);

- we experimentally validate our theoretical claims and propose a regularised maximum-likelihood learning approach based on the IMA constrast which outperforms the unregularised baseline (§ 5); additionally, we introduce a method to learn nonlinear ICA solutions with triangular Jacobian and a metric to assess BSS which can be of independent interest for the nonlinear ICA community.

---

[1]For additional intuition and possible violations in the context of the cocktail party problem, see Appendix B.4.

## 2 Background and preliminaries

Our work builds on and connects related literature from the fields of independent component analysis (§ 2.1) and causal inference (§ 2.2). We review the most important concepts below.

### 2.1 Independent component analysis (ICA)

Assume the following data-generating process for independent component analysis (ICA)

$$\mathbf{x} = \mathbf{f}(\mathbf{s})\,, \qquad\qquad p_{\mathbf{s}}(\mathbf{s}) = \prod_{i=1}^{n} p_{s_i}(s_i)\,, \qquad\qquad (1)$$

where the *observed mixtures* $\mathbf{x} \in \mathbb{R}^n$ result from applying a *smooth and invertible mixing function* $\mathbf{f} : \mathbb{R}^n \to \mathbb{R}^n$ to a set of *unobserved, independent signals or sources* $\mathbf{s} \in \mathbb{R}^n$ with smooth, factorised density $p_{\mathbf{s}}$ with connected support (see illustration Fig. 2b). The goal of ICA is to learn an *unmixing function* $\mathbf{g} : \mathbb{R}^n \to \mathbb{R}^n$ such that $\mathbf{y} = \mathbf{g}(\mathbf{x})$ has independent components. *Blind source separation* (BSS), on the other hand, aims to recover the true unmixing $\mathbf{f}^{-1}$ and thus the true sources $\mathbf{s}$ (up to tolerable ambiguities, see below). Whether performing ICA corresponds to solving BSS is related to the concept of *identifiability* of the model class. Intuitively, identifiability is the desirable property that *all models which give rise to the same mixture distribution should be "equivalent" up to certain ambiguities*, formally defined as follows.

**Definition 2.1** ($\sim$-identifiability). Let $\mathcal{F}$ be the set of all smooth, invertible functions $\mathbf{f} : \mathbb{R}^n \to \mathbb{R}^n$, and $\mathcal{P}$ be the set of all smooth, factorised densities $p_{\mathbf{s}}$ with connected support on $\mathbb{R}^n$. Let $\mathcal{M} \subseteq \mathcal{F} \times \mathcal{P}$ be a *subspace of models* and let $\sim$ be an *equivalence relation* on $\mathcal{M}$. Denote by $\mathbf{f}_* p_{\mathbf{s}}$ the *push-forward density* of $p_{\mathbf{s}}$ via $\mathbf{f}$. Then the generative process (1) is said to be $\sim$-*identifiable on* $\mathcal{M}$ if

$$\forall (\mathbf{f}, p_{\mathbf{s}}), (\tilde{\mathbf{f}}, p_{\tilde{\mathbf{s}}}) \in \mathcal{M} : \qquad \mathbf{f}_* p_{\mathbf{s}} = \tilde{\mathbf{f}}_* p_{\tilde{\mathbf{s}}} \qquad \Longrightarrow \qquad (\mathbf{f}, p_{\mathbf{s}}) \sim (\tilde{\mathbf{f}}, p_{\tilde{\mathbf{s}}})\,. \qquad (2)$$

If the true model belongs to the model class $\mathcal{M}$, then $\sim$-identifiability ensures that any model in $\mathcal{M}$ learnt from (infinite amounts of) data will be $\sim$-equivalent to the true one. An example is *linear* ICA which is identifiable up to permutation and rescaling of the sources on the subspace $\mathcal{M}_{\mathrm{LIN}}$ of pairs of (i) invertible matrices (constraint on $\mathcal{F}$) and (ii) factorizing densities for which at most one $s_i$ is Gaussian (constraint on $\mathcal{P}$) [17, 21, 93], see Appendix A for a more detailed account.

In the nonlinear case (i.e., without constraints on $\mathcal{F}$), identifiability is much more challenging. If $s_i$ and $s_j$ are independent, then so are $h_i(s_i)$ and $h_j(s_j)$ for any functions $h_i$ and $h_j$. In addition to permutation-ambiguity, such *element-wise* $\mathbf{h}(\mathbf{s}) = (h_1(s_1), ..., h_n(s_n))$ can therefore not be resolved either. We thus define the desired form of identifiability for nonlinear BSS as follows.

**Definition 2.2** ($\sim_{\mathrm{BSS}}$). The equivalence relation $\sim_{\mathrm{BSS}}$ on $\mathcal{F} \times \mathcal{P}$ defined as in Defn. 2.1 is given by

$$(\mathbf{f}, p_{\mathbf{s}}) \sim_{\mathrm{BSS}} (\tilde{\mathbf{f}}, p_{\tilde{\mathbf{s}}}) \iff \exists \mathbf{P}, \mathbf{h} \quad \text{s.t.} \quad (\mathbf{f}, p_{\mathbf{s}}) = (\tilde{\mathbf{f}} \circ \mathbf{h}^{-1} \circ \mathbf{P}^{-1}, (\mathbf{P} \circ \mathbf{h})_* p_{\tilde{\mathbf{s}}}) \qquad (3)$$

where $\mathbf{P}$ is a permutation and $\mathbf{h}(\mathbf{s}) = (h_1(s_1), ..., h_n(s_n))$ is an invertible, element-wise function.

A fundamental obstacle—and a crucial difference to the linear problem—is that in the nonlinear case, different mixtures of $s_i$ and $s_j$ can be independent, i.e., solving ICA is *not* equivalent to solving BSS. A prominent example of this is given by the *Darmois construction* [20, 39].

**Definition 2.3** (Darmois construction). The *Darmois construction* $\mathbf{g}^{\mathrm{D}} : \mathbb{R}^n \to (0, 1)^n$ is obtained by recursively applying the conditional cumulative distribution function (CDF) transform:

$$g_i^{\mathrm{D}}(\mathbf{x}_{1:i}) := \mathbb{P}(X_i \leq x_i | \mathbf{x}_{1:i-1}) = \int_{-\infty}^{x_i} p(x_i' | \mathbf{x}_{1:i-1}) dx_i' \qquad (i = 1, ..., n). \qquad (4)$$

The resulting *estimated* sources $\mathbf{y}^{\mathrm{D}} = \mathbf{g}^{\mathrm{D}}(\mathbf{x})$ are mutually-independent uniform r.v.s by construction, see Fig. 2a for an illustration. However, they need not be meaningfully related to the *true* sources $\mathbf{s}$, and will, in general, still be a nonlinear mixing thereof [39].[2] Denoting the mixing function corresponding to (4) by $\mathbf{f}^{\mathrm{D}} = (\mathbf{g}^{\mathrm{D}})^{-1}$ and the uniform density on $(0, 1)^n$ by $p_{\mathbf{u}}$, the *Darmois solution* $(\mathbf{f}^{\mathrm{D}}, p_{\mathbf{u}})$ thus allows construction of counterexamples to $\sim_{\mathrm{BSS}}$-identifiability on $\mathcal{F} \times \mathcal{P}$.[3]

*Remark* 2.4. $\mathbf{g}^{\mathrm{D}}$ has lower-triangular Jacobian, i.e., $\partial g_i^{\mathrm{D}} / \partial x_j = 0$ for $i < j$. Since the order of the $x_i$ is arbitrary, applying $\mathbf{g}^{\mathrm{D}}$ after a permutation yields a different Darmois solution. Moreover, (4) yields independent components $\mathbf{y}^{\mathrm{D}}$ even if the sources $s_i$ were not independent to begin with.[4]

---

[2]Consider, e.g., a mixing $\mathbf{f}$ with full Jacobian which yields a contradiction to Defn. 2.2, due to Remark 2.4.

[3]By applying a change of variables, we can see that the transformed variables in (4) are uniformly distributed in the open unit cube, thereby corresponding to independent components [69, § 2.2].

[4]This has broad implications for unsupervised learning, as it shows that, for i.i.d. observations, not only factorised priors, but *any* unconditional prior is insufficient for identifiability (see, e.g., [49], Appendix D.2).

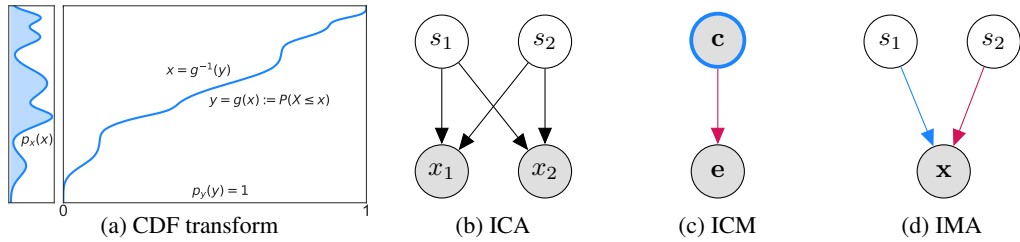

(a) CDF transform        (b) ICA        (c) ICM        (d) IMA

Figure 2: (a) Any observed density $p_x$ can be mapped to a uniform $p_y$ via the CDF transform $g(x) = \mathbb{P}(X \leq x)$; Darmois solutions $(\mathbf{f}^D, p_{\mathbf{u}})$ constructed from (4) therefore automatically satisfy the independence postulated by IGCI (6). (b) ICA setting with $n = 2$ sources (shaded nodes are observed, white ones are unobserved). (c) Existing ICM criteria typically enforce independence between an observed input or cause distribution $p_{\mathbf{c}}$ and a mechanism $p_{\mathbf{e}|\mathbf{c}}$ (independent objects are highlighted in blue and red). (d) IMA enforces independence between the contributions of different sources $s_i$ to the mixing function $\mathbf{f}$ as captured by $\partial \mathbf{f}/\partial s_i$.

Another well-known obstacle to identifiability are *measure-preserving automorphisms* (MPAs) of the source distribution $p_{\mathbf{s}}$: these are functions $\mathbf{a}$ which map the source space to itself without affecting its distribution, i.e., $\mathbf{a}_* p_{\mathbf{s}} = p_{\mathbf{s}}$ [39]. A particularly instructive class of MPAs is the following [49, 58].

**Definition 2.5** ("Rotated-Gaussian" MPA). Let $\mathbf{R} \in O(n)$ be an orthogonal matrix, and denote by $\mathbf{F}_{\mathbf{s}}(\mathbf{s}) = (F_{s_1}(s_1), ..., F_{s_n}(s_n))$ and $\boldsymbol{\Phi}(\mathbf{z}) = (\Phi(z_1), ..., \Phi(z_n))$ the element-wise CDFs of a smooth, factorised density $p_{\mathbf{s}}$ and of a Gaussian, respectively. Then the "rotated-Gaussian" MPA $\mathbf{a}^{\mathbf{R}}(p_{\mathbf{s}})$ is

$$\mathbf{a}^{\mathbf{R}}(p_{\mathbf{s}}) = \mathbf{F}_{\mathbf{s}}^{-1} \circ \boldsymbol{\Phi} \circ \mathbf{R} \circ \boldsymbol{\Phi}^{-1} \circ \mathbf{F}_{\mathbf{s}} . \tag{5}$$

$\mathbf{a}^{\mathbf{R}}(p_{\mathbf{s}})$ first maps to the (rotationally invariant) standard isotropic Gaussian (via $\boldsymbol{\Phi}^{-1} \circ \mathbf{F}_{\mathbf{s}}$), then applies a rotation, and finally maps back, without affecting the distribution of the estimated sources. Hence, if $(\tilde{\mathbf{f}}, p_{\tilde{\mathbf{s}}})$ is a valid solution, then so is $(\tilde{\mathbf{f}} \circ \mathbf{a}^{\mathbf{R}}(p_{\tilde{\mathbf{s}}}), p_{\tilde{\mathbf{s}}})$ for any $\mathbf{R} \in O(n)$. Unless $\mathbf{R}$ is a permutation, this constitutes another common counterexample to $\sim_{\text{BSS}}$-identifiability on $\mathcal{F} \times \mathcal{P}$.

Identifiability results for nonlinear ICA have recently been established for settings where an auxiliary variable $\mathbf{u}$ (e.g., environment index, time stamp, class label) renders the sources *conditionally* independent [37, 38, 41, 49]. The assumption on $p_{\mathbf{s}}$ in (1) is replaced with $p_{\mathbf{s}|\mathbf{u}}(\mathbf{s}|\mathbf{u}) = \prod_{i=1}^{n} p_{s_i|\mathbf{u}}(s_i|\mathbf{u})$, thus restricting $\mathcal{P}$ in Defn. 2.1. In most cases, $\mathbf{u}$ is assumed to be observed, though [30] is a notable exception. Similar results exist given access to a second noisy view $\tilde{\mathbf{x}}$ [28].

## 2.2 Causal inference and the principle of independent causal mechanisms (ICM)

Rather than relying only on additional assumptions on $\mathcal{P}$ (e.g., via auxiliary variables), we seek to further constrain (1) by also placing assumptions on the set $\mathcal{F}$ of mixing functions $\mathbf{f}$. To this end, we draw inspiration from the field of causal inference [71, 78]. Of central importance to our approach is the *Principle of Independent Causal Mechanisms* (ICM) [43, 56, 87].

**Principle 2.6** (ICM principle [78])**.** *The causal generative process of a system's variables is composed of autonomous modules that do not inform or influence each other.*

These "modules" are typically thought of as the conditional distributions of each variable given its direct causes. Intuitively, the principle then states that these *causal conditionals* correspond to *independent mechanisms of nature* which do not share information. Crucially, here "independent" does not refer to *statistical* independence of random variables, but rather to independence of the underlying distributions as *algorithmic* objects. For a bivariate system comprising a cause $\mathbf{c}$ and an effect $\mathbf{e}$, this idea reduces to an independence of cause and mechanism, see Fig. 2c. One way to formalise ICM uses Kolmogorov complexity $K(\cdot)$ [51] as a measure of algorithmic information [43].

However, since Kolmogorov complexity is is not computable, using ICM in practice requires assessing Principle 2.6 with other suitable proxy criteria [9, 11, 34, 42, 45, 65, 75–78, 90, 110].[5] Allowing for deterministic relations between cause (sources) and effect (observations), the criterion which is most closely related to the ICA setting in (1) is *information-geometric causal inference* (IGCI) [18, 46].[6] IGCI assumes a nonlinear relation $\mathbf{e} = \mathbf{f}(\mathbf{c})$ and formulates a notion of indepen-

---

[5]"This can be seen as an algorithmic analog of replacing the empirically undecidable question of statistical independence with practical independence tests that are based on assumptions on the underlying distribution" [43].

[6]For a similar criterion which assumes linearity [45, 110] and its relation to linear ICA, see Appendix B.1.

dence between the cause distribution $p_\mathbf{c}$ and the deterministic mechanism $\mathbf{f}$ (which we think of as a degenerate conditional $p_{\mathbf{e}|\mathbf{c}}$) via the following condition (in practice, assumed to hold approximately),

$$C_{\text{IGCI}}(\mathbf{f}, p_\mathbf{c}) := \int \log |\mathbf{J}_\mathbf{f}(\mathbf{c})| \, p_\mathbf{c}(\mathbf{c}) d\mathbf{c} - \int \log |\mathbf{J}_\mathbf{f}(\mathbf{c})| \, d\mathbf{c} = 0 \,, \tag{6}$$

where $(\mathbf{J}_\mathbf{f}(\mathbf{c}))_{ij} = \partial f_i / \partial c_j (\mathbf{c})$ is the Jacobian matrix and $|\cdot|$ the absolute value of the determinant. $C_{\text{IGCI}}$ can be understood as the covariance between $p_\mathbf{c}$ and $\log |\mathbf{J}_\mathbf{f}|$ (viewed as r.v.s on the unit cube w.r.t. the Lebesgue measure), so that $C_{\text{IGCI}} = 0$ rules out a form of fine-tuning between $p_\mathbf{c}$ and $|\mathbf{J}_\mathbf{f}|$. As its name suggests, IGCI can, from an information-geometric perspective, also be seen as an orthogonality condition between cause and mechanism in the space of probability distributions [46], see Appendix B.2, particularly eq. (19) for further details.

## 3 Existing ICM measures are insufficient for nonlinear ICA

Our aim is to use the ICM Principle 2.6 to further constrain the space of models $\mathcal{M} \subseteq \mathcal{F} \times \mathcal{P}$ and rule out common counterexamples to identifiability such as those presented in § 2.1. Intuitively, both the Darmois construction (4) and the rotated Gaussian MPA (5) give rise to "*non-generic*" solutions which should violate ICM: the former, $(\mathbf{f}^D, p_\mathbf{u})$, due the triangular Jacobian of $\mathbf{f}^D$ (see Remark 2.4), meaning that each observation $x_i = f_i^D(\mathbf{y}_{1:i})$ only depends on a subset of the inferred independent components $\mathbf{y}_{1:i}$, and the latter, $(\mathbf{f} \circ \mathbf{a}^\mathbf{R}(p_\mathbf{s}), p_\mathbf{s})$, due to the dependence of $\mathbf{f} \circ \mathbf{a}^\mathbf{R}(p_\mathbf{s})$ on $p_\mathbf{s}$ (5).

However, the ICM criteria described in § 2.2 were developed for the task of cause-effect inference where *both variables are observed*. In contrast, in this work, we consider an unsupervised representation learning task where *only the effects* (mixtures $\mathbf{x}$) *are observed*, but the causes (sources $\mathbf{s}$) are not. It turns out that this renders existing ICM criteria insufficient for BSS: they can easily be satisfied by spurious solutions which are not equivalent to the true one. We can show this for IGCI. Denote by $\mathcal{M}_{\text{IGCI}} = \{(\mathbf{f}, p_\mathbf{s}) \in \mathcal{F} \times \mathcal{P} : C_{\text{IGCI}}(\mathbf{f}, p_\mathbf{s}) = 0\} \subset \mathcal{F} \times \mathcal{P}$ the class of nonlinear ICA models satisfying IGCI (6). Then the following negative result holds.

**Proposition 3.1** (IGCI is insufficient for $\sim_{\text{BSS}}$-identifiability). (1) *is not* $\sim_{\text{BSS}}$-*identifiable on* $\mathcal{M}_{\text{IGCI}}$.

*Proof.* IGCI (6) is satisfied when $p_\mathbf{s}$ is uniform. However, the Darmois construction (4) yields uniform sources, see Fig. 2a. This means that $(\mathbf{f}^D \circ \mathbf{a}^\mathbf{R}(p_\mathbf{u}), p_\mathbf{u}) \in \mathcal{M}_{\text{IGCI}}$, so IGCI can be satisfied by solutions which do not separate the sources in the sense of Defn. 2.2, see footnote 2 and [39]. □

As illustrated in Fig. 2c, condition (6) and other similar criteria enforce a notion of "genericity" or "decoupling" of the mechanism w.r.t. the *observed* input distribution.[7] They thus rely on the fact that the cause (source) distribution is informative, and are generally not invariant to reparametrisation of the cause variables. In the (nonlinear) ICA setting, on the other hand, the *learnt* source distribution may be fairly uninformative. This poses a challenge for existing ICM criteria since any mechanism is generic w.r.t. an uninformative (uniform) input distribution.

## 4 Independent mechanism analysis (IMA)

As argued in § 3, enforcing independence between the input distribution and the mechanism (Fig. 2c), as existing ICM criteria do, is insufficient for ruling out spurious solutions to nonlinear ICA. We therefore propose a new ICM-inspired framework which is more suitable for BSS and which we term *independent mechanism analysis* (IMA).[8] All proofs are provided in Appendix C.

### 4.1 Intuition behind IMA

As motivated using the cocktail party example in § 1 and Fig. 1 *(Left)*, our main idea is to enforce a notion of *independence between the contributions or influences of the different sources* $s_i$ *on the observations* $\mathbf{x} = \mathbf{f}(\mathbf{s})$ as illustrated in Fig. 2d—as opposed to between the source distribution and mixing function, cf. Fig. 2c. These contributions or influences are captured by the vectors of partial derivatives $\partial \mathbf{f} / \partial s_i$. IMA can thus be understood as a *refinement of ICM at the level of the mixing* $\mathbf{f}$: in addition to *statistically independent components* $s_i$, we look for a mixing with *contributions* $\partial \mathbf{f} / \partial s_i$ *which are independent*, in a non-statistical sense which we formalise as follows.

**Principle 4.1** (IMA). *The mechanisms by which each source* $s_i$ *influences the observed distribution, as captured by the partial derivatives* $\partial \mathbf{f} / \partial s_i$, *are independent of each other in the sense that for all* $\mathbf{s}$:

$$\log |\mathbf{J}_\mathbf{f}(\mathbf{s})| = \sum_{i=1}^{n} \log \left\| \frac{\partial \mathbf{f}}{\partial s_i}(\mathbf{s}) \right\| \tag{7}$$

---

[7]In fact, many ICM criteria can be phrased as special cases of a unifying group-invariance framework [9].

[8]The title of the present work is thus a reverence to Pierre Comon's seminal 1994 paper [17].

**Geometric interpretation.** Geometrically, the IMA principle can be understood as an *orthogonality condition*, as illustrated for $n = 2$ in Fig. 1 *(Right)*. First, the vectors of partial derivatives $\partial \mathbf{f}/\partial s_i$, for which the IMA principle postulates independence, are the *columns* of $\mathbf{J_f}$. $|\mathbf{J_f}|$ thus measures the volume of the $n-$dimensional parallelepiped spanned by these columns, as shown on the right. The product of their norms, on the other hand, corresponds to the volume of an $n$-dimensional box, or rectangular parallelepiped with side lengths $\|\partial \mathbf{f}/\partial s_i\|$, as shown on the left. The two volumes are equal if and only if all columns $\partial \mathbf{f}/\partial s_i$ of $\mathbf{J_f}$ are orthogonal. Note that (7) is trivially satisfied for $n = 1$, i.e., if there is no mixing, further highlighting its difference from ICM for causal discovery.

**Independent influences and orthogonality.** In a high dimensional setting (large $n$), this orthogonality can be intuitively interpreted from the ICM perspective as *Nature choosing the direction of the influence of each source component in the observation space independently and from an isotropic prior*. Indeed, it can be shown that the scalar product of two independent isotropic random vectors in $\mathbb{R}^n$ vanishes as the dimensionality $n$ increases (equivalently: two high-dimensional isotropic vectors are typically orthogonal). This property was previously exploited in other linear ICM-based criteria (see [44, Lemma 5] and [45, Lemma 1 & Thm. 1]).[9] The principle in (7) can be seen as a constraint on the function space, enforcing such orthogonality between the columns of the Jacobian of $\mathbf{f}$ at all points in the source domain, thus approximating the high-dimensional behavior described above.[10]

**Information-geometric interpretation and comparison to IGCI.** The additive contribution of the sources' influences $\partial \mathbf{f}/\partial s_i$ in (7) suggests their local *decoupling at the level of the mechanism* $\mathbf{f}$. Note that IGCI (6), on the other hand, postulates a different type of decoupling: one between $\log |\mathbf{J_f}|$ and $p_\mathbf{s}$. There, dependence between cause and mechanism can be conceived as a fine tuning between the derivative of the mechanism and the input density. The IMA principle leads to a complementary, non-statistical measure of independence between the influences $\partial \mathbf{f}/\partial s_i$ of the individual sources on the vector of observations. Both the IGCI and IMA postulates have an information-geometric interpretation related to the influence of ("non-statistically") independent modules on the observations: both lead to an *additive decomposition of a KL-divergence between the effect distribution and a reference distribution.* For IGCI, independent modules correspond to the cause distribution and the mechanism mapping the cause to the effect (see (19) in Appendix B.2). For IMA, on the other hand, these are the influences of each source component on the observations in an interventional setting (under soft interventions on individual sources), as measured by the KL-divergences between the original and intervened distributions. See Appendix B.3, and especially (22), for a more detailed account.

We finally remark that while recent work based on the ICM principle has mostly used the term "mechanism" to refer to causal Markov kernels $p(X_i|PA_i)$ or structural equations [78], we employ it in line with the broader use of this concept in the philosophical literature.[11] To highlight just two examples, [86] states that *"Causal processes, causal interactions, and causal laws provide the mechanisms by which the world works; to understand why certain things happen, we need to see how they are produced by these mechanisms"*; and [99] states that *"Mechanisms are events that alter relations among some specified set of elements"*. Following this perspective, we argue that a causal mechanism can more generally denote any process that describes the way in which causes influence their effects: the partial derivative $\partial \mathbf{f}/\partial s_i$ thus reflects a causal mechanism in the sense that it describes the infinitesimal changes in the observations $\mathbf{x}$, when an infinitesimal perturbation is applied to $s_i$.

### 4.2 Definition and useful properties of the IMA contrast

We now introduce a contrast function based on the IMA principle (7) and show that it possesses several desirable properties in the context of nonlinear ICA. First, we define a local contrast as the difference between the two integrands of (7) for a particular value of the sources $\mathbf{s}$.

**Definition 4.2** (Local IMA contrast)**.** The local IMA contrast $c_{\text{IMA}}(\mathbf{f}, \mathbf{s})$ of $\mathbf{f}$ at a point $\mathbf{s}$ is given by

$$c_{\text{IMA}}(\mathbf{f}, \mathbf{s}) = \sum_{i=1}^{n} \log \left\| \frac{\partial \mathbf{f}}{\partial s_i}(\mathbf{s}) \right\| - \log |\mathbf{J_f}(\mathbf{s})| \ . \tag{8}$$

*Remark* 4.3. This corresponds to the left KL measure of diagonality [2] for $\sqrt{\mathbf{J_f}(\mathbf{s})^\top \mathbf{J_f}(\mathbf{s})}$.

---

[9]This has also been used as a *"leading intuition"* [sic] to interpret IGCI in [46].

[10]To provide additional intuition on how IMA differs from existing principles of independence of cause and mechanism, we give examples, both technical and pictorial, of violations of both in Appendix B.4.

[11]See Table 1 in [62] for a long list of definitions from the literature.

The local IMA contrast $c_{\mathrm{IMA}}(\mathbf{f}, \mathbf{s})$ quantifies the extent to which the IMA principle is violated at a given point $\mathbf{s}$. We summarise some of its properties in the following proposition.

**Proposition 4.4** (Properties of $c_{\mathrm{IMA}}(\mathbf{f}, \mathbf{s})$)**.** *The local IMA contrast $c_{\mathrm{IMA}}(\mathbf{f}, \mathbf{s})$ defined in* (8) *satisfies:*

*(i)* $c_{\mathrm{IMA}}(\mathbf{f}, \mathbf{s}) \geq 0$*, with equality if and only if all columns $\partial \mathbf{f}/\partial s_i(\mathbf{s})$ of $\mathbf{J_f}(\mathbf{s})$ are orthogonal.*

*(ii)* $c_{\mathrm{IMA}}(\mathbf{f}, \mathbf{s})$ *is invariant to left multiplication of $\mathbf{J_f}(\mathbf{s})$ by an orthogonal matrix and to right multiplication by permutation and diagonal matrices.*

Property *(i)* formalises the geometric interpretation of IMA as an orthogonality condition on the columns of the Jacobian from § 4.1, and property *(ii)* intuitively states that changes of orthonormal basis and permutations or rescalings of the columns of $\mathbf{J_f}$ do not affect their orthogonality. Next, we define a global IMA contrast w.r.t. a source distribution $p_{\mathbf{s}}$ as the expected local IMA contrast.

**Definition 4.5** (Global IMA contrast)**.** The global IMA contrast $C_{\mathrm{IMA}}(\mathbf{f}, p_{\mathbf{s}})$ of $\mathbf{f}$ w.r.t. $p_{\mathbf{s}}$ is given by

$$C_{\mathrm{IMA}}(\mathbf{f}, p_{\mathbf{s}}) = \mathbb{E}_{\mathbf{s} \sim p_{\mathbf{s}}}[c_{\mathrm{IMA}}(\mathbf{f}, \mathbf{s})] = \int c_{\mathrm{IMA}}(\mathbf{f}, \mathbf{s}) p_{\mathbf{s}}(\mathbf{s}) d\mathbf{s}. \tag{9}$$

The global IMA contrast $C_{\mathrm{IMA}}(\mathbf{f}, p_{\mathbf{s}})$ thus quantifies the extent to which the IMA principle is violated for a particular solution $(\mathbf{f}, p_{\mathbf{s}})$ to the nonlinear ICA problem. We summarise its properties as follows.

**Proposition 4.6** (Properties of $C_{\mathrm{IMA}}(\mathbf{f}, p_{\mathbf{s}})$)**.** *The global IMA contrast $C_{\mathrm{IMA}}(\mathbf{f}, p_{\mathbf{s}})$ from* (9) *satisfies:*

*(i)* $C_{\mathrm{IMA}}(\mathbf{f}, p_{\mathbf{s}}) \geq 0$*, with equality iff. $\mathbf{J_f}(\mathbf{s}) = \mathbf{O}(\mathbf{s})\mathbf{D}(\mathbf{s})$ almost surely w.r.t. $p_{\mathbf{s}}$, where $\mathbf{O}(\mathbf{s}), \mathbf{D}(\mathbf{s}) \in \mathbb{R}^{n \times n}$ are orthogonal and diagonal matrices, respectively;*

*(ii)* $C_{\mathrm{IMA}}(\mathbf{f}, p_{\mathbf{s}}) = C_{\mathrm{IMA}}(\tilde{\mathbf{f}}, p_{\tilde{\mathbf{s}}})$ *for any $\tilde{\mathbf{f}} = \mathbf{f} \circ \mathbf{h}^{-1} \circ \mathbf{P}^{-1}$ and $\tilde{\mathbf{s}} = \mathbf{Ph}(\mathbf{s})$, where $\mathbf{P} \in \mathbb{R}^{n \times n}$ is a permutation and $\mathbf{h}(\mathbf{s}) = (h_1(s_1), ..., h_n(s_n))$ an invertible element-wise function.*

Property *(i)* is the distribution-level analogue to *(i)* of Prop. 4.4 and only allows for orthogonality violations on sets of measure zero w.r.t. $p_{\mathbf{s}}$. This means that $C_{\mathrm{IMA}}$ can only be zero if $\mathbf{f}$ is an *orthogonal coordinate transformation* almost everywhere [19, 52, 66], see Fig. 3 for an example. We particularly stress property *(ii)*, as it precisely matches the inherent indeterminacy of nonlinear ICA: $C_{\mathrm{IMA}}$ *is blind to reparametrisation of the sources by permutation and element wise transformation.*

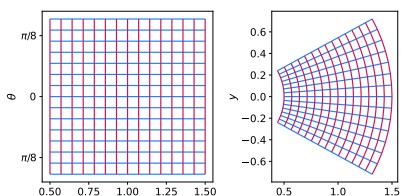

Figure 3: An example of a (non-conformal) orthogonal coordinate transformation from polar (left) to Cartesian (right) coordinates.

### 4.3 Theoretical analysis and justification of $C_{\mathrm{IMA}}$

We now show that, under suitable assumptions on the generative model (1), a large class of spurious solutions—such as those based on the Darmois construction (4) or measure preserving automorphisms such as $\mathbf{a}^{\mathbf{R}}$ from (5) as described in § 2.1—exhibit nonzero IMA contrast. Denote the class of nonlinear ICA models satisfying (7) (IMA) by $\mathcal{M}_{\mathrm{IMA}} = \{(\mathbf{f}, p_{\mathbf{s}}) \in \mathcal{F} \times \mathcal{P} : C_{\mathrm{IMA}}(\mathbf{f}, p_{\mathbf{s}}) = 0\} \subset \mathcal{F} \times \mathcal{P}$. Our first main theoretical result is that, under mild assumptions on the observations, Darmois solutions will have strictly positive $C_{\mathrm{IMA}}$, making them distinguishable from those in $\mathcal{M}_{\mathrm{IMA}}$.

**Theorem 4.7.** *Assume the data generating process in* (1) *and assume that $x_i \not\perp\!\!\!\perp x_j$ for some $i \neq j$. Then any Darmois solution $(\mathbf{f}^D, p_{\mathbf{u}})$ based on $\mathbf{g}^D$ as defined in* (4) *satisfies $C_{\mathrm{IMA}}(\mathbf{f}^D, p_{\mathbf{u}}) > 0$. Thus a solution satisfying $C_{\mathrm{IMA}}(\mathbf{f}, p_{\mathbf{s}}) = 0$ can be distinguished from $(\mathbf{f}^D, p_{\mathbf{u}})$ based on the contrast $C_{\mathrm{IMA}}$.*

The proof is based on the fact that the Jacobian of $\mathbf{g}^D$ is triangular (see Remark 2.4) and on the specific form of (4). A specific example of a mixing process satisfying the IMA assumption is the case where $\mathbf{f}$ is a conformal (angle-preserving) map.

**Definition 4.8** (Conformal map)**.** A smooth map $\mathbf{f} : \mathbb{R}^n \to \mathbb{R}^n$ is conformal if $\mathbf{J_f}(\mathbf{s}) = \mathbf{O}(\mathbf{s})\lambda(\mathbf{s}) \ \forall \mathbf{s}$, where $\lambda : \mathbb{R}^n \to \mathbb{R}$ is a scalar field, and $\mathbf{O} \in O(n)$ is an orthogonal matrix.

**Corollary 4.9.** *Under assumptions of Thm.* 4.7*, if additionally $\mathbf{f}$ is a conformal map, then $(\mathbf{f}, p_{\mathbf{s}}) \in \mathcal{M}_{\mathrm{IMA}}$ for any $p_{\mathbf{s}} \in \mathcal{P}$ due to Prop.* 4.6 *(i), see Defn.* 4.8*. Based on Thm.* 4.7*, $(\mathbf{f}, p_{\mathbf{s}})$ is thus distinguishable from Darmois solutions $(\mathbf{f}^D, p_{\mathbf{u}})$.*

This is consistent with a result that proves identifiability of conformal maps for $n = 2$ and conjectures it in general [39].[12] However, conformal maps are only a small subset of all maps for which $C_{\mathrm{IMA}} = 0$, as is apparent from the more flexible condition of Prop. 4.6 *(i)*, compared to the stricter Defn. 4.8.

---

[12]Note that Corollary 4.9 holds for any dimensionality $n$.

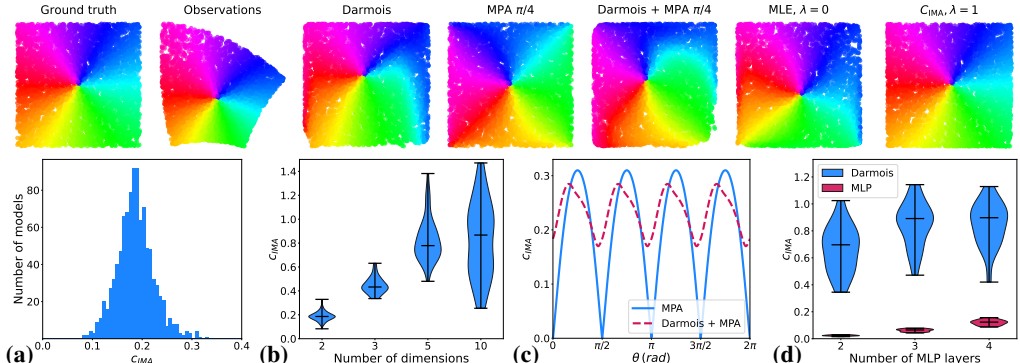

Figure 4: **Top.** Visual comparison of different nonlinear ICA solutions for $n = 2$: *(left to right)* true sources; observed mixtures; Darmois solution; true unmixing, composed with the measure preserving automorphism (MPA) from (5) (with rotation by $\pi/4$); Darmois solution composed with the same MPA; maximum likelihood ($\lambda = 0$); and $C_{\mathrm{IMA}}$-regularised approach ($\lambda = 1$). **Bottom.** Quantitative comparison of $C_{\mathrm{IMA}}$ for different spurious solutions: learnt Darmois solutions for **(a)** $n = 2$, and **(b)** $n \in \{2, 3, 5, 10\}$ dimensions; **(c)** composition of the MPA (5) in $n = 2$ dim. with the true solution (blue) and a Darmois solution (red) for different angles. **(d)** $C_{\mathrm{IMA}}$ distribution for true MLP mixing (red) vs. Darmois solution (blue) for $n = 5$ dim., $L \in \{2, 3, 4\}$ layers.

*Example* 4.10 (Polar to Cartesian coordinate transform). Consider the *non-conformal* transformation from polar to Cartesian coordinates (see Fig. 3), defined as $(x, y) = \mathbf{f}(r, \theta) := (r\cos(\theta), r\sin(\theta))$ with independent sources $\mathbf{s} = (r, \theta)$, with $r \sim U(0, R)$ and $\theta \sim U(0, 2\pi)$.[13] Then, $C_{\mathrm{IMA}}(\mathbf{f}, p_{\mathbf{s}}) = 0$ and $C_{\mathrm{IMA}}(\mathbf{f}^{\mathrm{D}}, p_{\mathbf{u}}) > 0$ for any Darmois solution $(\mathbf{f}^{\mathrm{D}}, p_{\mathbf{u}})$ —see Appendix D for details.

Finally, for the case in which the true mixing is linear, we obtain the following result.

**Corollary 4.11.** *Consider a linear ICA model, $\mathbf{x} = \mathbf{As}$, with $\mathbb{E}[\mathbf{s}^\top \mathbf{s}] = \mathbf{I}$, and $\mathbf{A} \in O(n)$ an orthogonal, non-trivial mixing matrix, i.e., not the product of a diagonal and a permutation matrix $\mathbf{DP}$. If at most one of the $s_i$ is Gaussian, then $C_{\mathrm{IMA}}(\mathbf{A}, p_{\mathbf{s}}) = 0$ and $C_{\mathrm{IMA}}(\mathbf{f}^{\mathrm{D}}, p_{\mathbf{u}}) > 0$.*

In a "blind" setting, we may not know a priori whether the true mixing is linear or not, and thus choose to learn a nonlinear unmixing. Corollary 4.11 shows that, in this case, Darmois solutions are still distinguishable from the true mixing via $C_{\mathrm{IMA}}$. Note that unlike in Corollary 4.9, the assumption that $x_i \not\perp\!\!\!\perp x_j$ for some $i \neq j$ is not required for Corollary 4.11. In fact, due to Theorem 11 of [17], it follows from the assumed linear ICA model with non-Gaussian sources, and the fact that the mixing matrix is not the product of a diagonal and a permutation matrix (see also Appendix A).

Having shown that the IMA principle allows to distinguish a class of models (including, but not limited to conformal maps) from Darmois solutions, we next turn to a second well-known counterexample to identifiability: the "rotated-Gaussian" MPA $\mathbf{a}^{\mathbf{R}}(p_{\mathbf{s}})$ (5) from Defn. 2.5. Our second main theoretical result is that, under suitable assumptions, this class of MPAs can also be ruled out for "non-trivial" $\mathbf{R}$.

**Theorem 4.12.** *Let $(\mathbf{f}, p_{\mathbf{s}}) \in \mathcal{M}_{\mathrm{IMA}}$ and assume that $\mathbf{f}$ is a conformal map. Given $\mathbf{R} \in O(n)$, assume additionally that $\exists$ at least one non-Gaussian $s_i$ whose associated canonical basis vector $\mathbf{e}_i$ is not transformed by $\mathbf{R}^{-1} = \mathbf{R}^\top$ into another canonical basis vector $\mathbf{e}_j$. Then $C_{\mathrm{IMA}}(\mathbf{f} \circ \mathbf{a}^{\mathbf{R}}(p_{\mathbf{s}}), p_{\mathbf{s}}) > 0$.*

Thm. 4.12 states that for conformal maps, applying the $\mathbf{a}^{\mathbf{R}}(p_{\mathbf{s}})$ transformation at the level of the sources leads to an increase in $C_{\mathrm{IMA}}$ except for very specific rotations $\mathbf{R}$ that are "fine-tuned" to $p_{\mathbf{s}}$ in the sense that they permute all non-Gaussian sources $s_i$ with another $s_j$. Interestingly, as for the linear case, non-Gaussianity again plays an important role in the proof of Thm. 4.12.

## 5 Experiments

Our theoretical results from § 4 suggest that $C_{\mathrm{IMA}}$ is a promising contrast function for nonlinear blind source separation. We test this empirically by evaluating the $C_{\mathrm{IMA}}$ of spurious nonlinear ICA solutions (§ 5.1), and using it as a learning objective to recover the true solution (§ 5.2).

We sample the ground truth sources from a uniform distribution in $[0, 1]^n$; the reconstructed sources are also mapped to the uniform hypercube as a reference measure via the CDF transform. Unless

---

[13]For different $p_{\mathbf{s}}$, $(x, y)$ can be made to have independent Gaussian components ([98], II.B), and $C_{\mathrm{IMA}}$-identifiability is lost; this shows that the assumption of Thm. 4.7 that $x_i \not\perp\!\!\!\perp x_j$ for some $i \neq j$ is crucial.

otherwise specified, the ground truth mixing $\mathbf{f}$ is a Möbius transformation [81] (i.e., a conformal map) with randomly sampled parameters, thereby satisfying Principle 4.1. In all of our experiments, we use JAX [12] and Distrax [13]. For additional technical details, equations and plots see Appendix E. The code to reproduce our experiments is available at this link.

## 5.1 Numerical evaluation of the $C_{\mathrm{IMA}}$ contrast for spurious nonlinear ICA solutions

**Learning the Darmois construction.** To learn the Darmois construction from data, we use normalising flows, see [35, 69]. Since Darmois solutions have triangular Jacobian (Remark 2.4), we use an architecture based on residual flows [16] which we constrain such that the Jacobian of the full model is triangular. This yields an expressive model which we train effectively via maximum likelihood.

**$C_{\mathrm{IMA}}$ of Darmois solutions.** To check whether Darmois solutions (learnt from finite data) can be distinguished from the true one, as predicted by Thm. 4.7, we generate 1000 random mixing functions for $n = 2$, compute the $C_{\mathrm{IMA}}$ values of learnt solutions, and find that all values are indeed significantly larger than zero, see Fig. 4 **(a)**. The same holds for higher dimensions, see Fig. 4 **(b)** for results with 50 random mixings for $n \in \{2, 3, 5, 10\}$: with higher dimensionality, both the mean and variance of the $C_{\mathrm{IMA}}$ distribution for the learnt Darmois solutions generally attain higher values.[14] We confirmed these findings for mappings which are not conformal, while still satisfying (7), in Appendix E.5.

**$C_{\mathrm{IMA}}$ of MPAs.** We also investigate the effect on $C_{\mathrm{IMA}}$ of applying an MPA $\mathbf{a}^{\mathbf{R}}(\cdot)$ from (5) to the true solution or a learnt Darmois solution. Results for $n = 2$ dim. for different rotation matrices $\mathbf{R}$ (parametrised by the angle $\theta$) are shown in Fig. 4 **(c)**. As expected, the behavior is periodic in $\theta$, and vanishes for the true solution (blue) at multiples of $\pi/2$, i.e., when $\mathbf{R}$ is a permutation matrix, as predicted by Thm. 4.12. For the learnt Darmois solution (red, dashed) $C_{\mathrm{IMA}}$ remains larger than zero.

**$C_{\mathrm{IMA}}$ values for random MLPs.** Lastly, we study the behavior of spurious solutions based on the Darmois construction under deviations from our assumption of $C_{\mathrm{IMA}} = 0$ for the true mixing function. To this end, we use invertible MLPs with orthogonal weight initalisation and `leaky_tanh` activations [29] as mixing functions; the more layers $L$ are added to the mixing MLP, the larger a deviation from our assumptions is expected. We compare the true mixing and learnt Darmois solutions over 20 realisations for each $L \in \{2, 3, 4\}$, $n = 5$. Results are shown in figure Fig. 4 **(d)**: the $C_{\mathrm{IMA}}$ of the mixing MLPs grows with $L$; still, the one of the Darmois solution is typically higher.

**Summary.** We verify that spurious solutions can be distinguished from the true one based on $C_{\mathrm{IMA}}$.

## 5.2 Learning nonlinear ICA solutions with $C_{\mathrm{IMA}}$-regularised maximum likelihood

**Experimental setup.** To use $C_{\mathrm{IMA}}$ as a learning signal, we consider a regularised maximum-likelihood approach, with the following objective: $\mathcal{L}(\mathbf{g}) = \mathbb{E}_{\mathbf{x}}[\log p_{\mathbf{g}}(\mathbf{x})] - \lambda\, C_{\mathrm{IMA}}(\mathbf{g}^{-1}, p_{\mathbf{y}})$, where $\mathbf{g}$ denotes the learnt unmixing, $\mathbf{y} = \mathbf{g}(\mathbf{x})$ the reconstructed sources, and $\lambda \geq 0$ a Lagrange multiplier. For $\lambda = 0$, this corresponds to standard maximum likelihood estimation, whereas for $\lambda > 0$, $\mathcal{L}$ lower-bounds the likelihood, and recovers it exactly iff. $(\mathbf{g}^{-1}, p_{\mathbf{y}}) \in \mathcal{M}_{\mathrm{IMA}}$. We train a residual flow $\mathbf{g}$ (with full Jacobian) to maximise $\mathcal{L}$. For evaluation, we compute (i) the KL divergence to the true data likelihood, as a measure of goodness of fit for the learnt flow model; and (ii) the mean correlation coefficient (MCC) between ground truth and reconstructed sources [37, 49]. We also introduce (iii) a nonlinear extension of the Amari distance [5] between the true mixing and the learnt unmixing, which is larger than or equal to zero, with equality iff. the learnt model belongs to the BSS equivalence class (Defn. 2.2) of the true solution, see Appendix E.5 for details.

**Results.** In Fig. 4 *(Top)*, we show an example of the distortion induced by different *spurious* solutions for $n = 2$, and contrast it with a solution learnt using our proposed objective *(rightmost plot)*. Visually, we find that the $C_{\mathrm{IMA}}$-regularised solution (with $\lambda = 1$) recovers the true sources most faithfully. Quantitative results for 50 learnt models for each $\lambda \in \{0.0, 0.5, 1.0\}$ and $n \in \{5, 7\}$ are summarised in Fig. 5 (see Appendix E for additional plots) . As indicated by the KL divergence values *(left)*, most trained models achieve a good fit to the data across all values of $\lambda$.[15] We observe that using $C_{\mathrm{IMA}}$ (i.e., $\lambda > 0$) is beneficial for BSS, both in terms of our nonlinear Amari distance *(center, lower is better)* and MCC *(right, higher is better)*, though we do not observe a substantial difference between $\lambda = 0.5$ and $\lambda = 1$.[16]

**Summary:** $C_{\mathrm{IMA}}$ can be a useful learning signal to recover the true solution.

---

[14]the latter possibly due to the increased difficulty of the learning task for larger $n$

[15]models with $n = 7$ have high outlier KL values, seemingly less pronounced for nonzero values of $\lambda$

[16]In Appendix E.5, we also show that our method is superior to a linear ICA baseline, FastICA [36].

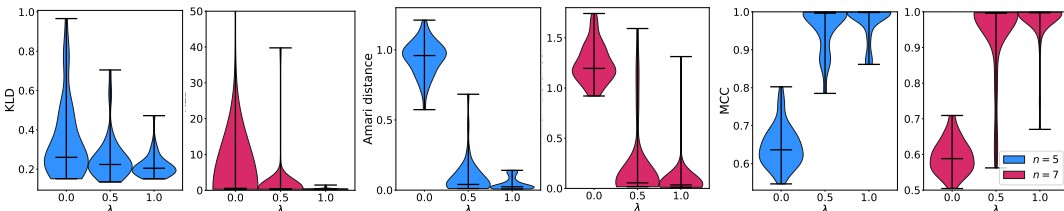

Figure 5: BSS via $C_{\text{IMA}}$-regularised MLE for, side by side, $n = 5$ (blue) and $n = 7$ (red) dim. with $\lambda \in \{0.0, 0.5, 1.0\}$. *(Left)* KL-divergence between ground truth likelihood and learnt model; *(center)* nonlinear Amari distance given true mixing and learnt unmixing; *(right)* MCC between true and reconstructed sources.

# 6   Discussion

**Assumptions on the mixing function.** Instead of relying on weak supervision in the form of auxiliary variables [28, 30, 37, 38, 41, 49], our IMA approach places additional constraints on the functional form of the mixing process. In a similar vein, the *minimal nonlinear distortion principle* [108] proposes to favor solutions that are as close to linear as possible. Another example is the *post-nonlinear model* [98, 109], which assumes an element-wise nonlinearity applied after a linear mixing. IMA is different in that it still allows for strongly nonlinear mixings (see, e.g., Fig. 3) provided that the columns of their Jacobians are (close to) orthogonal. In the related field of disentanglement [8, 58], a line of work that focuses on image generation with adversarial networks [24] similarly proposes to constrain the "generator" function via regularisation of its Jacobian [82] or Hessian [74], though mostly from an empirically-driven, rather than from an identifiability perspective as in the present work.

**Towards identifiability with $C_{\text{IMA}}$.** The IMA principle rules out a large class of spurious solutions to nonlinear ICA. While we do not present a full identifiability result, our experiments show that $C_{\text{IMA}}$ can be used to recover the BSS equivalence class, suggesting that identifiability might indeed hold, possibly under additional assumptions—e.g., for conformal maps [39].

**IMA and independence of cause and mechanism.** While inspired by measures of independence of cause and mechanism as traditionally used for cause-effect inference [18, 45, 46, 110], we view the IMA principle as addressing a different question, in the sense that they evaluate independence between different elements of the causal model. Any nonlinear ICA solution that satisfies the IMA Principle 4.1 can be turned into one with uniform reconstructed sources—thus satisfying IGCI as argued in § 3—through composition with an element-wise transformation which, according to Prop. 4.6 *(ii)*, leaves the $C_{\text{IMA}}$ value unchanged. Both IGCI (6) and IMA (7) can therefore be fulfilled simultaneously, while the former on its own is inconsequential for BSS as shown in Prop. 3.1.

**BSS through algorithmic information.** Algorithmic information theory has previously been proposed as a unifying framework for identifiable approaches to *linear* BSS [67, 68], in the sense that commonly-used contrast functions could, under suitable assumptions, be interpreted as proxies for the total complexity of the mixing and the reconstructed sources. However, to the best of our knowledge, the problem of specifying suitable proxies for the complexity of *nonlinear* mixing functions has not yet been addressed. We conjecture that our framework could be linked to this view, based on the additional assumption of algorithmic independence of causal mechanisms [43], thus potentially representing an approach to *nonlinear* BSS by minimisation of algorithmic complexity.

**ICA for causal inference & causality for ICA.** Past advances in ICA have inspired novel causal discovery methods [50, 64, 92]. The present work constitutes, to the best of our knowledge, the first effort to use ideas from causality (specifically ICM) for BSS. An application of the IMA principle to causal discovery or causal representation learning [88] is an interesting direction for future work.

**Conclusion.** We introduce IMA, a path to nonlinear BSS inspired by concepts from causality. We postulate that the *influences* of different sources on the observed distribution should be approximately independent, and formalise this as an orthogonality condition on the columns of the Jacobian. We prove that this constraint is generally violated by well-known spurious nonlinear ICA solutions, and propose a regularised maximum likelihood approach which we empirically demonstrate to be effective in recovering the true solution. Our IMA principle holds exactly for orthogonal coordinate transformations, and is thus of potential interest for learning spatial representations [33], robot dynamics [63], or physics problems where orthogonal reference frames are common [66].

## Acknowledgements

The authors thank Aapo Hyvärinen, Adrián Javaloy Bornás, Dominik Janzing, Giambattista Parascandolo, Giancarlo Fissore, Nasim Rahaman, Patrick Burauel, Patrik Reizinger, Paul Rubenstein, Shubhangi Ghosh, and the anonymous reviewers for helpful comments and discussions.

## Funding Transparency Statement

This work was supported by the German Federal Ministry of Education and Research (BMBF): Tübingen AI Center, FKZ: 01IS18039B; and by the Machine Learning Cluster of Excellence, EXC number 2064/1 - Project number 390727645.

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
