# APPENDIX

## Overview

- Appendix A contains further elaboration on the notion of identifiability as used in the present work, as well as connections to linear ICA.
- Appendix B contains additional discussion of existing ICM criteria and their relation to IMA.
- Appendix C presents the full proofs for all theoretical results from the main paper.
- Appendix D contains a worked out computation of the value of $C_{\mathrm{IMA}}$ for the mapping from radial to Cartesian coordinates.
- Appendix E contains experimental details and additional results.
- Appendix F contains additional background on conformal maps and Möbius transformations

## A   Additional background on identifiability and linear ICA

In this Appendix, we provide additional background on the notion of identifiability and illustrate it using the example of linear ICA.

### A.1   Identifiability in terms of equivalence relations

Traditionally, identifiability for a class of models $p_\theta$ for observed data $\mathbf{x}$ parametrised by $\theta \in \Theta$ is expressed as the condition that there needs to be a one-to-one mapping between the space of models and the space of parameters, i.e., the model class $p_\theta$ is said to be identifiable if

$$\forall \theta, \theta' \in \Theta: \qquad p_\theta(\mathbf{x}) = p_{\theta'}(\mathbf{x}) \forall \mathbf{x} \quad \implies \quad \theta = \theta'. \tag{10}$$

However, the equality on the RHS of (10) is a very strong condition which makes this type of (strong or unique) identifiability impractical for many settings. For example, in the case of (linear or nonlinear) ICA, the ordering of the sources cannot be determined, so strong identifiability in the sense of (10) is infeasible.

The equality in parameter space on the RHS of the implication in (10) is therefore sometimes replaced by an equivalence relation $\sim$ [49], as is also the case for our Defn. 2.1. An equivalence relation $\sim$ on a set $A$ is a binary relation between pairs of elements of $A$ which satisfies the following three properties:

1. Reflexivity: $a \sim a, \forall a \in A$.
2. Symmetry: $a \sim b \implies b \sim a, \forall a, b \in A$.
3. Transitivity: $(a \sim b) \wedge (b \sim c) \implies a \sim c$.

An equivalence relation on a set $A$ imposes a partition into disjoint subsets. Each such subset corresponds to an equivalence class, i.e., the collection of all elements which are $\sim$-related to each other; for example, $[a] = \{b \in A : a \sim b\}$ denotes the equivalence class containing the element $a$.

A trivial example of an equivalence relation is equality ($=$). More useful examples in the context of ICA are equivalence up to permutation, rescaling, or scalar transformation.

Defining an appropriate equivalence class for the problem at hand therefore allows us to specify exactly the type of indeterminacies which cannot be resolved and up to which the true generative process can be recovered. As argued in § 2, for nonlinear ICA, the desired notion of identifiability—in the sense of the strongest feasible type of identifiable that is possible without further (parametric) assumptions—is captured by $\sim_{\mathrm{BSS}}$ from Defn. 2.2. We give another example for linear ICA in Appendix A.2.

Since the generative process of nonlinear ICA (1) is determined by the choice of mixing function and source distribution, the space $\Theta$ from (10), in this case, corresponds to the product space of the space of mixing functions $\mathcal{F}$ and source distributions $\mathcal{P}$. Moreover, the pushforward density $\mathbf{f}_* p_{\mathbf{s}}$ in Defn. 2.1 corresponds to the density of the observed mixtures $p_{\mathbf{x}}$, or $p_\theta(\mathbf{x})$ in (10).

We deliberately choose to define identifiability and to express the observed distribution in terms of the source distribution and the mixing function—as opposed to in terms of the observed distribution and the unmixing function as in some prior work [37, 38, 41]—because this is aligned with the causal

direction of data generation, and thus more consistent with the causal perspective at nonlinear BSS taken in the present work. We also believe that, in this framework, separate constraints on the space of mixing functions $\mathcal{F}$ and source distributions $\mathcal{P}$ are expressed more naturally.

Next, we illustrate the above ideas for the well-studied case of linear ICA.

## A.2 Identifiability of linear ICA

Linear ICA corresponds to the setting in which a linear mixing is applied to independent sources, i.e.,

$$\mathbf{x} = \mathbf{As}, \tag{11}$$

where $\mathbf{A} \in \mathbb{R}^{n \times n}$ is an invertible mixing matrix. The source variables $\mathbf{s}$ can be assumed to have zero mean without affecting estimation of the mixing matrix, and the ordering and variances of the independent components cannot be determined, so it is customary to assume $\mathbb{E}[s_i^2] = 1$ [40].

Additionally, we can assume w.l.o.g. that the mixing matrix is orthogonal ($\mathbf{AA}^\top = \mathbf{I}$), because we can always *whiten* $\mathbf{x}$ first through an invertible linear transformation and obtain an orthogonal mixing [40], as explained in more detail in Appendix A.3.

Now suppose that the reconstructed sources

$$\mathbf{y} = \mathbf{Bx} = \mathbf{BAs} \tag{12}$$

have independent components for some orthogonal unmixing matrix $\mathbf{B} \in \mathbb{R}^{n \times n}$. Then $\mathbf{C} = \mathbf{BA}$ is also orthogonal and the following type of identifiability holds [17, 21, 93].

**Theorem A.1** (Identifiability of linear ICA; based on Thm. 11 of [17])**.** *Let* $\mathbf{s}$ *be a vector of* $n$ *independent components, of which at most one is Gaussian and whose densities are not reduced to a point mass. Let* $\mathbf{C} \in \mathbb{R}^{n \times n}$ *be an orthogonal matrix. Then* $\mathbf{y} = \mathbf{Cs}$ *has (mutually) independent components iff.* $\mathbf{C} = \mathbf{DP}$*, with* $\mathbf{D}$ *a diagonal matrix and* $\mathbf{P}$ *a permutation matrix.*

Thm. A.1 shows that the two ambiguities deemed unresolvable (scale and ordering of the sources) are, in fact, the only ambiguities, as long as at most one of the $s_i$ is Gaussian. That is, linear ICA is identifiable up to rescaling and permutation of the sources, i.e., linearly transforming the observations $\mathbf{x}$ into independent components is equivalent to separating the sources.

More formally, in terms of an equivalence relation, if we take $\mathcal{F}'$ from (2) as the space of invertible $n \times n$ matrices and $\mathcal{P}'$ as the space of source distributions with at most one Gaussian marginal, then linear ICA is $\sim_{\text{LIN}}$-identifiable on $\mathcal{F}' \times \mathcal{P}'$ where the equivalence relation $\sim_{\text{LIN}}$ on $\mathcal{F}'$ is defined as

$$\mathbf{B} \sim_{\text{LIN}} \mathbf{B}' \iff \exists \mathbf{D}, \mathbf{P} \text{ s.t. } \mathbf{B} = \mathbf{DPB}'.$$

**Beyond non-Gaussianity.** Two other deviations from a Gaussian i.i.d. setting lead to identifiability: nonstationarity [79] and time correlation [80]. A general information-geometric framework links these three different routes to identifiability [15].

## A.3 Whitening in the context of linear ICA

For completeness, we give a brief account of the role of *whitening in linear ICA*, which was mentioned in A.2 and which again plays a role in B.1. The following exposition is partly based on [40], §7.4.2.

A zero-mean random vector, say $\mathbf{y}$, is said to be *white* if its components are uncorrelated and their variances equal unity. In other words, the covariance matrix of $\mathbf{y}$ is equal to the identity matrix:

$$\mathbb{E}\left[\mathbf{yy}^\top\right] = \mathbf{I}.$$

It is always possible to whiten a zero-mean random vector $\mathbf{x}$ through a linear operation,

$$\mathbf{z} = \mathbf{Vx}. \tag{13}$$

As an example, a popular method for whitening uses the eigenvalue decomposition (EVD) of the covariance matrix,

$$\mathbb{E}\left[\mathbf{xx}^\top\right] = \mathbf{EDE}^\top$$

where $\mathbf{E}$ is the orthogonal matrix of eigenvectors of $\mathbb{E}\left[\mathbf{xx}^\top\right]$ and $\mathbf{D}$ is the diagonal matrix of its eigenvalues, $\mathbf{D} = \text{diag}\left(\lambda_1, \ldots, \lambda_n\right)$. Note that the covariance matrix is a symmetric matrix, therefore it is diagonalisable. Whitening can then be performed by substituting in (13) the matrix

$$\mathbf{V} = \mathbf{ED}^{-1/2}\mathbf{E}^\top. \tag{14}$$

so that

$$\mathbb{E}[\mathbf{z}\mathbf{z}^\top] = \mathbf{E}\mathbf{D}^{-1/2}\mathbf{E}^\top\mathbf{E}\mathbf{D}\mathbf{E}^\top\mathbf{E}\mathbf{D}^{-1/2}\mathbf{E}^\top = \mathbf{I}$$

**Whitening is only half ICA.** Assume a linear ICA model,

$$\mathbf{x} = \mathbf{A}\mathbf{s}\,. \tag{15}$$

and suppose that the observed data is whitened, for example, by the matrix $\mathbf{V}$ given in (14). Whitening transforms the mixing matrix into a new one, $\tilde{\mathbf{A}} = \mathbf{V}\mathbf{A}$. We have from (15) and (14)

$$\mathbf{z} = \mathbf{V}\mathbf{A}\mathbf{s} = \tilde{\mathbf{A}}\mathbf{s}$$

Note that whitening does not solve linear ICA, since *uncorrelatedness is weaker than independence*. To see this, consider any orthogonal transformation $\mathbf{U}$ of $\mathbf{z}$:

$$\mathbf{y} = \mathbf{U}\mathbf{z}.$$

Due to the orthogonality of $\mathbf{U}$, we have

$$\mathbb{E}\left[\mathbf{y}\mathbf{y}^\top\right] = \mathbb{E}\left[\mathbf{U}\mathbf{z}\mathbf{z}^\top\mathbf{U}^\top\right] = \mathbf{U}\mathbb{E}\left[\mathbf{z}\mathbf{z}^\top\right]\mathbf{U}^T = \mathbf{U}\mathbf{I}\mathbf{U}^\top = \mathbf{I}\,,$$

so, $\mathbf{y}$ is white as well. Thus, we cannot tell if the independent components are given by $\mathbf{z}$ or $\mathbf{y}$ using the whiteness property alone. Since $\mathbf{y}$ could be any orthogonal transformation of $\mathbf{z}$, whitening gives the independent components only up to an orthogonal transformation.

On the other hand, whitening is useful as a pre-processing step in ICA: its utility resides in the fact that the new mixing matrix $\tilde{\mathbf{A}} = \mathbf{V}\mathbf{A}$ is orthogonal. This can be seen from

$$\mathbb{E}\left[\mathbf{z}\mathbf{z}^\top\right] = \tilde{\mathbf{A}}\mathbb{E}\left[\mathbf{s}\mathbf{s}^\top\right]\tilde{\mathbf{A}}^\top = \tilde{\mathbf{A}}\tilde{\mathbf{A}}^\top = \mathbf{I}.$$

We can thus restrict the search for the (un)mixing matrix to the space of orthogonal matrices. Instead of having to estimate $n^2$ parameters (the elements of the original matrix $\mathbf{A}$), we only need to estimate an orthogonal mixing matrix $\tilde{\mathbf{A}}$ which contains $n(n-1)/2$ degrees of freedom; e.g., in two dimensions, an orthogonal transformation is determined by a single angle parameter. For larger $n$, an orthogonal matrix contains only about half of the number of parameters of an arbitrary matrix.

Whitening thus "solves half of the problem of ICA". Because whitening is a very simple and standard procedure—much simpler than any ICA algorithm—it is a good idea to reduce the complexity of the problem this way. The remaining half of the parameters has to be estimated by some other method.

## B  Existing ICM criteria and their relationship to ICA and IMA

We now provide additional discussion of the ICM principle and its connection to ICA and IMA. First, we introduce a linear ICM criterion and discuss its relation with linear ICA in Appendix B.1.

### B.1  Trace method

As mentioned in § 2.2, besides IGCI, another existing ICM criterion that is closely related to ICA due to also assuming a deterministic relation between cause $\mathbf{c}$ and effect $\mathbf{e}$ is the *trace method* [45, 110]. The trace method assumes a linear relationship,

$$\mathbf{e} = \mathbf{Ac}, \tag{16}$$

and formulates ICM as an "independence" between the covariance matrix $\mathbf{\Sigma}$ of $\mathbf{c}$ and the mechanism $\mathbf{A}$ (which, as for IGCI, we can again think of as a degenerate conditional $p_{\mathbf{e}|\mathbf{c}}$) via the condition

$$\tau(\mathbf{A}\mathbf{\Sigma}\mathbf{A}^\top) = \tau(\mathbf{\Sigma})\tau(\mathbf{A}\mathbf{A}^\top) \tag{17}$$

where $\tau(\cdot)$ denotes the renormalized trace. Intuitively, this condition (17) rules out a fine-tuning of $\mathbf{A}$ to the eigenvectors of $\mathbf{\Sigma}$ which would violate the assumption of no shared information between the cause distribution (specifically, its covariance structure) and the mechanism.

As with IGCI and nonlinear ICA, it can be seen by comparing (16) and (11) that *the trace method assumes the same generative model as linear ICA* (where the cause $\mathbf{c}$ corresponds to the independent sources $\mathbf{s}$ and the effect to the observed mixtures $\mathbf{x}$). While the focus of the present work is on nonlinear ICA, we briefly discuss the usefulness of the trace method as a constraint for achieving identifiability in a linear ICA setting.

As is clear from (17), the trace condition is trivially satisfied if the covariance matrix of the sources (causes) is the identity, $\mathbf{\Sigma} = \mathbf{I}$. However, as explained in Appendix A.3, in the context of linear ICA this can easily be achieved by whitening the data. As with IGCI, the trace method was developed for cause-effect inference where both variables are observed, and thus relies on the observed cause distribution being informative. This renders is unsuitable (on its own) to constrain the unsupervised representation learning problem of linear ICA problem where the sources are unobserved.

Note, however, that this is qualitatively different from the IGCI argument presented in § 3, as whitening on its own does not necessarily lead to independent variables, but only uncorrelated ones, and thus does not solve linear ICA—unlike the Darmois construction in the case of nonlinear ICA which also yields independent components.

### B.2  Information geometric interpretation of the ICM principle

There is a well-established connection between IGCI and the trace method [46]. At the heart of this derivation lies an information-geometric interpretation of the ICM principle for probability distributions, which we sketch in this section. First, we need to review some basic concepts.

**Background on information geometry.**  Information geometry [3, 4] is a discipline in which ideas from differential geometry are applied to probability theory. Probability distributions correspond to points on a Riemannian manifold, known as *statistical manifold*. Equipped with the Kullback-Leibler (KL) divergence, also called the relative entropy distance, as a premetric,[17] one can study the geometrical properties of the statistical manifold. For two probability distributions $P$ and $Q$, we denote their KL divergence by $D_{KL}(P\|Q)$, which is defined for $P$ absolutely continuous with respect to $Q$ as:

$$D_{KL}(P\|Q) = \int dP \log \frac{dP}{dQ}.$$

An interesting property of the KL divergence is its invariance to reparametrisation. Consider an invertible transformation $h$, mapping random variables $X$ and $Y$ to $h(X)$ and $h(Y)$, respectively (the domains and codomains being arbitrary spaces, e.g., discrete or Euclidean of arbitrary dimension). Then the KL divergence between $P_X$ and $P_Y$ is preserved by the pushforward operation implemented by $h$, such that

$$D_{KL}(P_{h(X)}\|P_{h(Y)}) = D_{KL}(P_X\|P_Y). \tag{18}$$

---

[17]A premetric on a set $\mathcal{X}$ is a function $d : \mathcal{X} \times \mathcal{X} \to \mathbb{R}^+ \cup \{0\}$ such that (i) $d(x, y) \geq 0$ for all $x$ and $y$ in $\mathcal{X}$ and (ii) $d(x, x) = 0$ for all $x \in \mathcal{X}$.

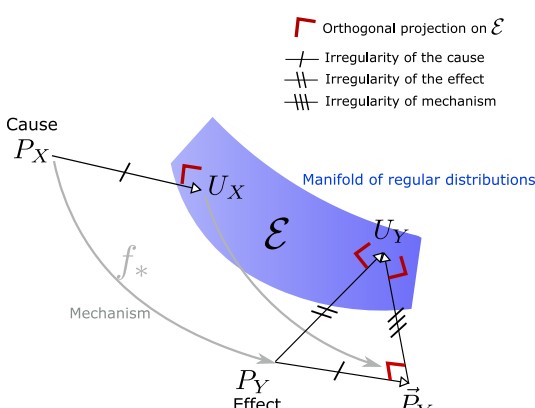

Figure 6: **Interpretation of the ICM principle as an orthogonality principle in information space.** The irregularity of the effect distribution, as measured by $D_{KL}(P_Y \| U_Y)$, can be decomposed into the irregularities of the cause, as measured by $D_{KL}(P_X \| U_X)$, and the irregularity of the mechanism $f$, as measured by $D_{KL}(\vec{P}_Y \| U_Y)$. Here, $U_X$ and $U_Y$ denote the orthogonal projections of $P_X$ and $P_Y$ onto the manifold $\mathcal{E}$ of regular distributions, and $\vec{P}_Y$ denotes the pushforward of the regular distribution $U_X$ via $f$. Note that the KL divergence is invariant to reparametrisation by invertible functions.

**Interpretation of ICM as orthogonality condition in information space.** Consider a deterministic causal relationship of the form $Y := f(X)$, and denote by $P_X$ and $P_Y$ the marginal distributions of the cause $X$ and the effect $Y$, respectively. The "irregularity" of each distribution can be quantified by evaluating their divergence to a reference set $\mathcal{E}$ of "regular" distributions,[18]

$$D_{KL}(P_X \| \mathcal{E}) = \inf_{U \in \mathcal{E}} D_{KL}(P_X \| U), \quad D_{KL}(P_Y \| \mathcal{E}) = \inf_{U \in \mathcal{E}} D_{KL}(P_Y \| U).$$

Let us assume that these infima are reached at a unique point, their projections onto $\mathcal{E}$:

$$U_X = \arg\min_{U \in \mathcal{E}} D_{KL}(P_X \| U), \quad U_Y = \arg\min_{U \in \mathcal{E}} D_{KL}(P_Y \| U).$$

As elaborated in [46, §4], the choice of $\mathcal{E}$ is context-dependent. For example, in the context of the trace method [45], $X$ and $Y$ are assumed to be $n$-dimensional multivariate Gaussian random vectors, and $\mathcal{E}$ is taken as the set of multivariate *isotropic* Gaussian distributions. In contrast, when IGCI is applied in contexts where the considered mechanism is a deterministic non-linear diffeomorphism, the reference distributions are typically uniform distributions [18, 47].

Overall, it can be shown that the independence postulate underlying these approaches leads to the following decomposition of the irregularity of $P_Y$ (see [46, Thm. 2]):

$$D_{KL}(P_Y \| U_Y) = D_{KL}(P_Y \| \vec{P}_Y) + D_{KL}(\vec{P}_Y \| U_Y)$$

where $\vec{P}_Y$ denotes the distribution of $f(U_X)$, i.e., the hypothetical distribution of the effect that would be obtained if the cause $X$ were replaced by the random variable $U_X$ (which corresponds to the closest regularly distributed random variable to $X$).

Since applying the bijection $f^{-1}$ preserves the KL divergences, see (18), we can obtain the equivalent relation

$$D_{KL}(P_Y \| U_Y) = D_{KL}(P_X \| U_X) + D_{KL}(\vec{P}_Y \| U_Y). \tag{19}$$

This relation can be interpreted as an *orthogonality principle* in information space by considering the KL divergences as a generalization of the squared Euclidean norm for the difference vectors $\overrightarrow{P_Y U_Y}$, $\overrightarrow{P_Y \vec{P}_Y}$ and $\overrightarrow{\vec{P}_Y U_Y}$. It can thus be viewed as a Pythagorean theorem in the space of distributions, see Fig. 6 for an illustration.

The orthogonality principle (19) thus captures a decomposition of the irregularity $D_{KL}(P_Y \| U_Y)$ of $P_Y$ on the LHS into the sum of two irregularities on the RHS: the irregularity $D_{KL}(P_X \| U_X)$ of $P_X$,

---

[18]Here "regular" is only meant in an intuitive sense, not implying any further mathematical notion. If $\mathcal{E}$ is the set of Gaussians, for instance, the distance from $\mathcal{E}$ measures non-Gaussianity.

and the term $D_{KL}(\vec{P}_Y \| U_Y)$ which measures the irregularity of the mechanism $f$ indirectly, via the "irregularity" of the distribution resulting from applying $f$ to a regular distribution $U_Y$.

Overall, the decomposition (19) links the postulate of independence between the cause distribution, on the one hand, and the mechanism, on the other hand, to an *orthogonality of their irregularities in information space* (namely the statistical manifold of information geometry). As proposed in [46], this can be intuitively interpreted as a geometric form of independence if we assume that Nature chooses such irregularities independently of each other, and "isotropically" in a high-dimensional subspace of irregularities.

While, to date, we are not aware of similar results in the context of information geometry (i.e., on the statistical manifold), this intuition is supported by concentration of measure results in Euclidean spaces. Indeed, in high-dimensions, it is likely that two vectors are close to orthogonal if they are chosen independently according to a uniform prior [25].

We will take inspiration of the decomposition (19) to justify IMA in the following section.

### B.3 Decoupling of the influences in IMA and comparison with IGCI

In contrast to Appendix B.2, in this section we will, for notational consistency with the main paper, assume that all distributions have a density with respect to the Lebesgue measure, and thus consider, with a slight abuse of notation, that the KL divergence is a distance between two densities on the relevant support, such that

$$D_{KL}(p\|q) = \int p(x) \log \frac{p(x)}{q(x)} dx \,.$$

**Overview.** In line with the information-geometric interpretation of IGCI presented in Appendix B.2, we also consider an interpretation of IMA in information space. We consider the KL-divergence between the observed density $p_{\mathbf{x}}$ of $\mathbf{x} = \mathbf{f}(\mathbf{s})$ and an *interventional* distribution $p_{\widehat{\mathbf{x}}}$ of $\widehat{\mathbf{x}} = \widehat{\mathbf{f}}(\mathbf{s})$, resulting from a soft intervention that replaces the mixing function $\mathbf{f}$ with another mixing $\widehat{\mathbf{f}}$. We take $D_{KL}(p_{\mathbf{x}}\|p_{\widehat{\mathbf{x}}})$ as a measure of the causal effect of the soft intervention (or perturbation) that turns $\mathbf{f}$ into $\widehat{\mathbf{f}}$—similarly to how $D_{KL}(P_Y\|U_Y)$ is used as a measure of the irregularity of the effect distribution in the context of IGCI (Appendix B.2).

As we will show, under suitable assumptions, the functional form imposed on $\mathbf{f}$ by the IMA Principle 4.1 can lead to a decomposition of the *causal effect* of an intervention on the mechanism into a sum of terms, corresponding to the causal effects of separate soft interventions on the mechanisms associated to each source. In contrast, IGCI decomposes *irregularities* of the effect distribution into *two terms, one irregularity of the cause and one irregularity of the mechanism*.

**Soft-interventions on the individual mechanisms.** Assume $\mathbf{f}$ satisfies the IMA principle. We consider interventions performed through the element-wise transformation $\boldsymbol{\sigma}$ such that

$$\boldsymbol{\sigma} : \mathbf{s} \mapsto \begin{bmatrix} \sigma_1(s_1) \\ \vdots \\ \sigma_j(s_j) \\ \vdots \\ \sigma_n(s_n) \end{bmatrix} \,.$$

This can be seen as a composition of $n$ soft interventions $\{\boldsymbol{\sigma}_j\}$ on each individual source component $j$, implemented through univariate smooth diffeomorphisms $\sigma_j$, such that

$$\boldsymbol{\sigma}_j : \mathbf{s} \mapsto \begin{bmatrix} s_1 \\ \vdots \\ \sigma_j(s_j) \\ \vdots \\ s_n \end{bmatrix} \,,$$

and $\boldsymbol{\sigma} = \boldsymbol{\sigma}_n \circ \cdots \circ \boldsymbol{\sigma}_1$ (in arbitrary order, since the individual $\boldsymbol{\sigma}_j$ commute). This soft intervention can be seen as turning the random variable $\mathbf{s}$ into $\widehat{\mathbf{s}}$, yielding the intervened observations $\widehat{\mathbf{x}} = \mathbf{f}(\widehat{\mathbf{s}})$.

Alternatively, the intervention on $\mathbf{x}$ can be implemented by replacing $\mathbf{f}$ by $\widehat{\mathbf{f}} = \mathbf{f} \circ \boldsymbol{\sigma}$—i.e., $\widehat{\mathbf{x}} = \widehat{\mathbf{f}}(\mathbf{s})$. Notably, since $\mathbf{f}$ satisfies the IMA principle, so does $\widehat{\mathbf{f}}$ (due to Prop. 4.6, *(ii)*, since $\boldsymbol{\sigma}$ is an element-wise nonlinearity). Moreover, the partial derivatives of the intervened function are given by

$$\frac{\partial \widehat{\mathbf{f}}}{\partial s_i}(\mathbf{s}) = \frac{\partial \mathbf{f}}{\partial s_i}(\boldsymbol{\sigma}(\mathbf{s})) \left| \frac{d\sigma_i}{ds_i} \right| (s_i).$$

The classical change of variable formula for bijection $\mathbf{f}$ yields the expression of the pushforward density of $\mathbf{x}$ as

$$p_{\mathbf{x}}(\mathbf{x}) = |J_{\mathbf{f}}(\mathbf{f}^{-1}(\mathbf{x}))|^{-1} p_{\mathbf{s}}(\mathbf{f}^{-1}(\mathbf{x})),$$

and for $\widehat{\mathbf{x}}$ we get

$$p_{\widehat{\mathbf{x}}}(\widehat{\mathbf{x}}) = |J_{\widehat{\mathbf{f}}}(\widehat{\mathbf{f}}^{-1}(\widehat{\mathbf{x}}))|^{-1} p_{\mathbf{s}}(\widehat{\mathbf{f}}^{-1}(\widehat{\mathbf{x}})),$$

**Information geometric interpretation of IMA.** Let us now compute the KL divergence between the intervened and observed distribution,

$$D_{KL}(p_{\mathbf{x}} \| p_{\widehat{\mathbf{x}}}) = \int p_{\mathbf{x}}(\mathbf{x}) \log \frac{p_{\mathbf{x}}(\mathbf{x})}{p_{\widehat{\mathbf{x}}}(\mathbf{x})} d\mathbf{x}. \tag{20}$$

Expressing the density of the observed variables as a pushforward of the density of the sources, and without additional assumptions on $\mathbf{f}$ and $\widehat{\mathbf{f}}$ besides smoothness and invertibility, we get,

$$D_{KL}(p_{\mathbf{x}} \| p_{\widehat{\mathbf{x}}}) = \int \left| \mathbf{J}_{\mathbf{f}}(\mathbf{f}^{-1}(\mathbf{x})) \right|^{-1} p_{\mathbf{s}}(\mathbf{f}^{-1}(\mathbf{x})) \log \frac{\left| \mathbf{J}_{\mathbf{f}}(\mathbf{f}^{-1}(\mathbf{x})) \right|^{-1} p_{\mathbf{s}}(\mathbf{f}^{-1}(\mathbf{x}))}{\left| \mathbf{J}_{\widehat{\mathbf{f}}}(\widehat{\mathbf{f}}^{-1}(\mathbf{x})) \right|^{-1} p_{\mathbf{s}}(\widehat{\mathbf{f}}^{-1}(\mathbf{x}))} d\mathbf{x}.$$

We now consider a factorization of $\mathbf{s}$ over a directed acyclic graph (DAG), such that

$$p_{\mathbf{s}}(\mathbf{s}) = \prod_j p_j(s_j | \mathrm{pa}(s_j)),$$

where $\mathrm{pa}(s_j)$ denotes the components associated to the parents of node $j$ in the DAG. Because $\boldsymbol{\sigma}$ is an element-wise transformation the factorization will be the same for $p_{\widehat{\mathbf{s}}}$.

If we now additionally assume that $\mathbf{f}$ and $\widehat{\mathbf{f}}$ satisfy the IMA postulate, we get

$$D_{KL}(p_{\mathbf{x}} \| p_{\widehat{\mathbf{x}}}) = \int \left| \mathbf{J}_{\mathbf{f}}(\mathbf{f}^{-1}(\mathbf{x})) \right|^{-1} p_{\mathbf{s}}(\mathbf{f}^{-1}(\mathbf{x})) \sum_{i=1}^{n} \log \frac{\left\| \frac{\partial \mathbf{f}}{\partial s_i}(\mathbf{f}^{-1}(\mathbf{x})) \right\|^{-1} p_i(\mathbf{f}^{-1}(\mathbf{x})_i | \mathrm{pa}(\mathbf{f}^{-1}(\mathbf{x})_i))}{\left\| \frac{\partial \widehat{\mathbf{f}}}{\partial s_i}(\widehat{\mathbf{f}}^{-1}(\mathbf{x})) \right\|^{-1} p_i(\widehat{\mathbf{f}}^{-1}(\mathbf{x})_i | \mathrm{pa}(\widehat{\mathbf{f}}^{-1}(\mathbf{x})_i))} d\mathbf{x}.$$

By reparameterizing the integral in terms of the source coordinates, we get (using $\widehat{\mathbf{f}}^{-1} = \boldsymbol{\sigma}^{-1} \circ \mathbf{f}^{-1}$)

$$D_{KL}(p_{\mathbf{x}} \| p_{\widehat{\mathbf{x}}}) = \sum_{i=1}^{n} \int p_{\mathbf{s}}(\mathbf{s}) \log \frac{\left\| \frac{\partial \mathbf{f}}{\partial s_i}(\mathbf{s}) \right\|^{-1} p_i(\mathbf{s}_i | \mathrm{pa}(\mathbf{s}_i))}{\left\| \frac{\partial \widehat{\mathbf{f}}}{\partial s_i}(\boldsymbol{\sigma}^{-1}(\mathbf{s})) \right\|^{-1} p_i(\boldsymbol{\sigma}^{-1}(\mathbf{s})_i | \mathrm{pa}(\boldsymbol{\sigma}^{-1}(\mathbf{s})_i))} d\mathbf{s}. \tag{21}$$

such that the $KL$ divergence can be written as a sum of $n$ terms, each associated to the intervention on a mechanism $\frac{\partial \mathbf{f}}{\partial s_i}$. Positivity of these terms would suggest that we can interpret each of them as quantifying the individual contribution of a soft intervention $\boldsymbol{\sigma}_j$ applied to the original sources.

In the following, we propose a justification for the positivity of these terms *in a restricted setting where only the $m$ leaf nodes of the graph are intervened on (with $1 \le m \le n$).*[19] In the special case of independent sources, all nodes are leaves and $m = n$.

Under this assumption, we consider (without loss of generality) an ordering of the nodes such that the $m$ first nodes are the leaf nodes in the DAG. Then we argue that the terms of the right-hand side of (21) associated to leaf nodes ($i \le m$) are positive, as they correspond to the expectations of KL-divergences. Indeed, taking one of the first $m$ terms, denoted $i$, we have the factorization

$$p_{\mathbf{s}}(\mathbf{s}) = p_i(s_i | \mathrm{pa}(s_i)) \prod_{j \ne i} p_j(s_j | \mathrm{pa}(s_j)),$$

---

[19] A leaf node in a DAG is one that does not have any descendants.

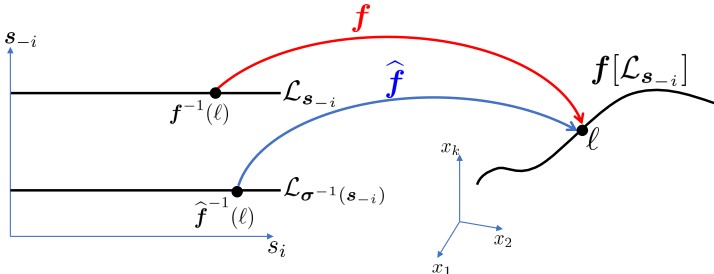

Figure 7: Illustration of the mapping between lines in source space to a curve in observation space. $\mathcal{L}_{\mathbf{s}_{-i}}$ is the line obtained by varying $s_i$ while keeping the value of all other sources fixed to $\mathbf{s}_{-i}$. $\mathcal{L}_{\boldsymbol{\sigma}^{-1}(\mathbf{s}_{-i})}$ is then defined by applying the transformations in $[\boldsymbol{\sigma}^{-1}]_{-i}$ to $\mathcal{L}_{\mathbf{s}_{-i}}$. Both lines are mapped to the same image line $\mathbf{f}[\mathcal{L}_{\mathbf{s}_{-i}}]$.

where $\prod_{j \neq i} p_j(s_j|\mathrm{pa}(s_j))$ does not depend on $s_i$ because node $i$ is a leaf node. Moreover, as non-leaf nodes are not intervened on, the transformation $\boldsymbol{\sigma}$ does not modify the value of any parent variables in these factorizations. As a consequence, the integral can be computed as an iterated integral with respect to $s_i$ and $\mathbf{s}_{-i}$, where $\mathbf{s}_{-i}$ denotes the vector including all source variables but $s_i$, such that

$$\int p_{\mathbf{s}}(\mathbf{s}) \log \frac{\left\| \frac{\partial \mathbf{f}}{\partial s_i}(\mathbf{s}) \right\|^{-1} p_i(\mathbf{s}_i|\mathrm{pa}(\mathbf{s}_i))}{\left\| \frac{\partial \widehat{\mathbf{f}}}{\partial s_i}(\boldsymbol{\sigma}^{-1}(\mathbf{s})) \right\|^{-1} p_i(\boldsymbol{\sigma}^{-1}(\mathbf{s})_i|\mathrm{pa}(\boldsymbol{\sigma}^{-1}(\mathbf{s})_i))} d\mathbf{s}$$

$$= \mathbb{E}_{\mathbf{s}_{-i} \sim \prod_{j \neq i} p_j(s_j|\mathrm{pa}(s_j))} \left[ \int p(s_i|\mathrm{pa}(s_i)) \log \frac{\left\| \frac{\partial \mathbf{f}}{\partial s_i}(s_i, \mathbf{s}_{-i}) \right\|^{-1} p_i(\mathbf{s}_i|\mathrm{pa}(\mathbf{s}_i))}{\left\| \frac{\partial \widehat{\mathbf{f}}}{\partial s_i}(\sigma_i^{-1}(s_i), \boldsymbol{\sigma}^{-1}(\mathbf{s})_{-i}) \right\|^{-1} p_i(\sigma_i^{-1}(s_i)|\mathrm{pa}(s_i))} ds_i \right] .$$

As illustrated in Fig. 7, for a fixed $\mathbf{s}_{-i}$, consider the straight line $\mathcal{L}_{\mathbf{s}_{-i}} = \{(s_i, \mathbf{s}_{-i}) : s_i \in \mathbb{R}\}$ in source space (parallel to the $s_i$ coordinate axis). This line is mapped in observation space to the smooth curve $\mathbf{f}[\mathcal{L}_{\mathbf{s}_{-i}}]$, by $\mathbf{f}$ in a smooth invertible way. Similarly, $\widehat{\mathbf{f}} = \mathbf{f} \circ \boldsymbol{\sigma}$ maps $\mathcal{L}_{\boldsymbol{\sigma}^{-1}(\mathbf{s}_{-i})}$ to the same image curve, since $\widehat{\mathbf{f}}[\mathcal{L}_{\boldsymbol{\sigma}^{-1}(\mathbf{s}_{-i})}] = \mathbf{f} \circ \boldsymbol{\sigma}[\mathcal{L}_{\boldsymbol{\sigma}^{-1}(\mathbf{s}_{-i})}] = \mathbf{f}[\mathcal{L}_{\mathbf{s}_{-i}}]$.

By using the change of variable formula to represent the integral on $\mathbf{f}[\mathcal{L}_{\mathbf{s}_{-i}}]$ indexed by the curvilinear coordinate $\ell$, we get the expression of the pushfoward distribution $\mathbf{f}_* p_i(\,.\,|\mathrm{pa}(s_i))$ on the curve $\mathbf{f}[\mathcal{L}_{\mathbf{s}_{-i}}]$

$$\left[ \mathbf{f}_* p_i(\,.\,|\mathrm{pa}(s_i)) \right](\ell) = \left\| \frac{\partial \mathbf{f}}{\partial s_i} \left( \mathbf{f}^{-1}(\ell), \mathbf{s}_{-i} \right) \right\|^{-1} p_i \left( \mathbf{f}^{-1}(\ell)|\mathrm{pa}(s_i) \right) .$$

where, to simplify notation, $\mathbf{f}^{-1}(\ell)$ denotes in this context the coordinate $s_i$ on $\mathcal{L}_{\mathbf{s}_{-i}}$ in bijection with the curvilinear coordinate $\ell$ on $\mathbf{f}[\mathcal{L}_{\mathbf{s}_{-i}}]$.

Similarly, we get the expression of the pushfoward distribution $\widehat{\mathbf{f}}_* p_i(\,.\,|\boldsymbol{\sigma}^{-1}(\mathrm{pa}(s_i)))$ from $\mathcal{L}_{\boldsymbol{\sigma}^{-1}(\mathbf{s}_{-i})}$ to the curve $\mathbf{f}[\mathcal{L}_{\mathbf{s}_{-i}}]$ (using again the fact that parent variables are not intervened on, and thus left unchanged by $\boldsymbol{\sigma}$)

$$\left[ \widehat{\mathbf{f}}_* p_i(\,.\,|\boldsymbol{\sigma}^{-1}(\mathrm{pa}(s_i))) \right](\ell) = \left\| \frac{\partial \widehat{\mathbf{f}}}{\partial s_i} \left( \widehat{\mathbf{f}}^{-1}(\ell), \boldsymbol{\sigma}^{-1}(\mathbf{s})_{-i} \right) \right\|^{-1} p_i \left( \widehat{\mathbf{f}}^{-1}(\ell)|\mathrm{pa}(s_i) \right) .$$

These terms appear when rewriting the $i$-th term (for a leaf variable) in (21) as a curvilinear integral:

$$\int p_{\mathbf{s}}(\mathbf{s}) \log \frac{\left\| \frac{\partial \mathbf{f}}{\partial s_i}(\mathbf{s}) \right\|^{-1} p_i(\mathbf{s}_i | \mathrm{pa}(\mathbf{s}_i))}{\left\| \frac{\partial \widehat{\mathbf{f}}}{\partial s_i}(\boldsymbol{\sigma}^{-1}(\mathbf{s})) \right\|^{-1} p_i(\boldsymbol{\sigma}^{-1}(\mathbf{s})_i | \mathrm{pa}(\boldsymbol{\sigma}^{-1}(\mathbf{s})_i))} d\mathbf{s}$$

$$= \mathbb{E}_{\mathbf{s}_{-i} \sim \prod_{j \neq i} p_j(s_j | \mathrm{pa}(s_j))} \left[ \int \left\| \frac{\partial \mathbf{f}}{\partial s_i}(\mathbf{f}^{-1}(\ell), \mathbf{s}_{-i}) \right\|^{-1} p_i(\mathbf{f}^{-1}(\ell) \mid \mathrm{pa}(s_i)) \right.$$

$$\left. \log \frac{\left\| \frac{\partial \mathbf{f}}{\partial s_i}(\mathbf{f}^{-1}(\ell), \mathbf{s}_{-i}) \right\|^{-1} p_i(\mathbf{f}^{-1}(\ell) \mid \mathrm{pa}(s_i))}{\left\| \frac{\partial \widehat{\mathbf{f}}}{\partial s_i}(\widehat{\mathbf{f}}^{-1}(\ell), \boldsymbol{\sigma}^{-1}(\mathbf{s})_{-i}) \right\|^{-1} p_i\left(\widehat{\mathbf{f}}^{-1}(\ell) \mid \mathrm{pa}(s_i)\right)} d\ell \right].$$

The inner integral term can thus be interpreted as the KL divergence between two pushforward measures defined on $\mathbf{f}_*[\mathcal{L}_{\mathbf{s}_{-i}}]$ by $\mathbf{f}$ and $\widehat{\mathbf{f}}$, that we can denote by

$$D_{KL}\left(\mathbf{f}_* p_i\left(\,.\,| \mathrm{pa}(s_i)\right) \| \widehat{\mathbf{f}}_* p\left(\,.\,| \boldsymbol{\sigma}^{-1}(\mathrm{pa}(\mathbf{s}_i))\right)\right).$$

To conclude, this implies that the causal effect of the soft intervention $\mathbf{f} \to \widehat{\mathbf{f}}$ can be decomposed as the following sum of $m$ positive terms associated to interventions on each leaf variable, plus an additional term for the remaining non-leaf variables, which further simplifies (in comparison to (21)) due to the assumption that those variables are unintervened.

$$D_{KL}(p_{\mathbf{x}} \| p_{\widehat{\mathbf{x}}}) = \sum_{i=1}^{m} \mathbb{E}_{\mathbf{s}_{-i} \sim \prod_{j \neq i} p_j(s_j | \mathrm{pa}(s_j))} \left[ D_{KL}\left(\mathbf{f}_* p\left(\,.\,| \mathrm{pa}(s_i) \| \widehat{\mathbf{f}}_* p\left(\,.\,| \boldsymbol{\sigma}^{-1}(\mathrm{pa}(s_i))\right)\right)\right)\right]$$

$$+ \sum_{i > m} \int p_{\mathbf{s}}(\mathbf{s}) \log \frac{\left\| \frac{\partial \mathbf{f}}{\partial s_i}(\mathbf{s}) \right\|^{-1}}{\left\| \frac{\partial \widehat{\mathbf{f}}}{\partial s_i}(\boldsymbol{\sigma}^{-1}(\mathbf{s})) \right\|^{-1}} d\mathbf{s}. \quad (22)$$

This expression suggests that the KL-divergences appearing in the first $m$ terms each reflect the causal effect of an intervention on the mechanism at the level of one single source coordinate $i$, turning $\frac{\partial \mathbf{f}}{\partial s_i}$ into $\frac{\partial \widehat{\mathbf{f}}}{\partial s_i}$. When the sources are jointly independent, we have $m = n$ and the right hand side of (22) contains only positive terms. An interesting direction for future work would be to analyse the remaining term in the case of non unconditionally independent sources.

In contrast to the decomposition (19) in the context of IGCI, the IMA decomposition (22) involves $m$ (expectations of) KL-divergence terms instead of two, each related to the intervention on the part of the mechanism $\frac{\partial \mathbf{f}}{\partial s_i}$ that reflects the influence of a single source.

### B.4 Independence of cause and mechanism and IMA

We now discuss an example in which a formalisation of the principle of independence of cause and mechanism [45] is violated, and one in which the IMA principle is violated.

#### B.4.1 Violations of independence of cause and mechanism

In the context of the Trace method [45], used in causal discovery, a technical example of fine-tuning can be constructed by taking a vector of i.i.d. random variables with arbitrary (not diagonal) covariance matrix $\Sigma$ as the cause, and by constructing the mechanism as a whitening matrix, turning the cause variables into uncorrelated (effect) variables. By doing so, the singular values and singular vectors of the matrix (the mechanism) are fine-tuned to the input covariance matrix (a property of the cause distribution), and such fine-tuning can be quantified via the Trace method (see [45], Section 1).

#### B.4.2 Violations of the IMA principle

**Technical example.** As mentioned in § 3, an example of a mixing function $\mathbf{f}$ which is non-generic according to the IMA principle is an autoregressive function, for example an autoregressive normalising flow [69], where the $k$-th component of the observations only depends on the $k$-th

sources: intuitively, this would correspond to the unlikely cocktail-party setting where the $k$-th microphone only picks up the voices of the first speakers. More precisely, as we show in Lemma C.1, this leads to positive $C_{\text{IMA}}$ value for such mixing.

**Pictorial example: Violations of the IMA principle in a cocktail party.**   A cocktail party (Fig. 1, left) may violate our IMA principle when the locations of several speakers and the room acoustics have been fine tuned to one another. This is for example the case in concert halls where the acoustics of the room have been fine-tuned to the position and configuration of multiple locations on the stage, where the sources (i.e., the voices of the actors or singers) are emitted—in order to make the listening experience as homogeneous as possible across the spectators (that is, the influence of each of the sources on the different listeners should not differ too much). This would lead to an increase in collinearity between the columns of the mixing's Jacobian, thus violating the IMA principle.

Additionally, we recall that the ICM principle is often informally introduced by referencing the fine-tuning and non-generic viewpoints giving rise to certain visual illusions, such as the Beuchet chair (see [78], Section 2); in a similar vein, we can imagine that violations of the IMA principle in the cocktail party setting may be related to illusions in binaural hearing such as for example the Franssen effect, where the listener is tricked into incorrectly localizing a sound [89].

# C Proofs

We now provide the proofs of all our theoretical results from the main paper.

## C.1 Proof of Prop. 4.4

Before giving the proof, it is useful to rewrite the local IMA constrast (8) as follows:

$$
\begin{aligned}
c_{\mathrm{IMA}}(\mathbf{f}, \mathbf{s}) &= \sum_{i=1}^{n} \log \left\| \frac{\partial \mathbf{f}}{\partial s_i}(\mathbf{s}) \right\| - \log |\mathbf{J_f}(\mathbf{s})| \\
&= \frac{1}{2} \left( \log \left| \mathrm{diag} \left( \mathbf{J_f^\top}(\mathbf{s}) \mathbf{J_f}(\mathbf{s}) \right) \right| - \log \left| \mathbf{J_f^\top}(\mathbf{s}) \mathbf{J_f}(\mathbf{s}) \right| \right) \\
&= \frac{1}{2} D_{KL}^{\mathrm{left}} \left( \mathbf{J_f^\top}(\mathbf{s}) \mathbf{J_f}(\mathbf{s}) \right) ,
\end{aligned}
\tag{23}
$$

where the quantity in (23) is called the left KL measure of diagonality of the matrix $\mathbf{J_f^\top}(\mathbf{s})\mathbf{J_f}(\mathbf{s})$ [2] (see Remark 4.3):

$$
\begin{aligned}
D_{KL}^{\mathrm{left}}(\mathbf{A}) &= - \log \left| (\mathrm{diag}(\mathbf{A}))^{-\frac{1}{2}} \mathbf{A} (\mathrm{diag}(\mathbf{A}))^{-\frac{1}{2}} \right| \\
&= \log |\mathrm{diag}(\mathbf{A})| - \log |\mathbf{A}| .
\end{aligned}
$$

From (23), it can be seen that $c_{\mathrm{IMA}}(\mathbf{f}, \mathbf{s})$ is a function of $\mathbf{J_f}(\mathbf{s})$ only through $\mathbf{J_f^\top}(\mathbf{s})\mathbf{J_f}(\mathbf{s})$.

**Proposition 4.4** (Properties of $c_{\mathrm{IMA}}(\mathbf{f}, \mathbf{s})$). *The local IMA contrast $c_{\mathrm{IMA}}(\mathbf{f}, \mathbf{s})$ defined in (8) satisfies:*

*(i) $c_{\mathrm{IMA}}(\mathbf{f}, \mathbf{s}) \geq 0$, with equality if and only if all columns $\partial \mathbf{f}/\partial s_i(\mathbf{s})$ of $\mathbf{J_f}(\mathbf{s})$ are orthogonal.*

*(ii) $c_{\mathrm{IMA}}(\mathbf{f}, \mathbf{s})$ is invariant to left multiplication of $\mathbf{J_f}(\mathbf{s})$ by an orthogonal matrix and to right multiplication by permutation and diagonal matrices.*

*Proof.* For ease of exposition, we denote the value of the Jacobian of $\mathbf{f}$ evaluated at the point $\mathbf{s}$ by $\mathbf{J_f}(\mathbf{s}) = \mathbf{W}$. The two properties can then be proved as follows:

(i) This is a consequence of Hadamard's inequality, applied to the expression on the RHS of (8), which states that, for a matrix $\mathbf{W}$ with columns $\mathbf{w}_i$, $\sum_{i=1}^n \log \|\mathbf{w}_i\| \geq \log |\mathbf{W}|$; equality in Hadamard's inequality is achieved iff. the vectors $\mathbf{w}_i$ are orthogonal.

(ii) We split the proof in three parts.

    a. *Invariance to left multiplication by an orthogonal matrix:*
    Let $\tilde{\mathbf{W}} = \mathbf{OW}$, with $\mathbf{O}$ an orthogonal matrix, i.e., $\mathbf{OO^\top} = \mathbf{I}$. Then the property follows from writing $c_{\mathrm{IMA}}(\mathbf{f}, \mathbf{s})$ as in (23):

$$
\frac{1}{2} D_{KL}^{\mathrm{left}}(\tilde{\mathbf{W}}^\top \tilde{\mathbf{W}}) = \frac{1}{2} D_{KL}^{\mathrm{left}}(\mathbf{W}^\top \mathbf{O}^\top \mathbf{O} \mathbf{W}) = \frac{1}{2} D_{KL}^{\mathrm{left}}(\mathbf{W}^\top \mathbf{I} \mathbf{W}) = \frac{1}{2} D_{KL}^{\mathrm{left}}(\mathbf{W}^\top \mathbf{W})
$$

    b. *Invariance to right multiplication by a permutation matrix:*
    Let $\tilde{\mathbf{W}} = \mathbf{WP}$, with $\mathbf{P}$ a permutation matrix. Then $\tilde{\mathbf{W}}$ is just $\mathbf{W}$ with permuted columns. Clearly, the sum of the log-column-norms does not change by changing the order of the summands. Further, $\log |\tilde{\mathbf{W}}| = \log |\mathbf{W}| + \log |\mathbf{P}| = \log |\mathbf{W}|$, because the absolute value of the determinant of a permutation matrix is one.

    c. *Invariance to right multiplication by a diagonal matrix*:
    Let $\tilde{\mathbf{W}} = \mathbf{WD}$, with $\mathbf{D}$ a diagonal matrix. Consider the two terms on the RHS of (8). For the first term, we know that the columns of $\tilde{\mathbf{W}}$ are scaled versions of the columns of $\mathbf{W}$, that is $\tilde{\mathbf{w}}_i = d_i \mathbf{w}_i$, where $d_i$ denotes the $i^{\mathrm{th}}$ diagonal element of $\mathbf{D}$. Then $\|\tilde{\mathbf{w}}_i\| = |d_i| \|\mathbf{w}_i\|$. For the second term, we use the decomposition of the determinant:

$$
\log |\tilde{\mathbf{W}}| = \log |\mathbf{W}| + \log |\mathbf{D}| = \log |\mathbf{W}| + \sum_{i=1}^{n} \log |d_i|.
$$

Taken together, we obtain

$$\sum_{i=1}^{n} \log \|\tilde{\mathbf{w}}_i\| - \log |\tilde{\mathbf{W}}| = \sum_{i=1}^{n} \log \left(|d_i| \, \|\mathbf{w}_i\|\right) - \left(\log |\mathbf{W}| + \sum_{i=1}^{n} \log |d_i|\right)$$

$$= \sum_{i=1}^{n} \log \|\mathbf{w}_i\| + \sum_{i=1}^{n} \log |d_i| - \log |\mathbf{W}| - \sum_{i=1}^{n} \log |d_i|$$

$$= \sum_{i=1}^{n} \log \|\mathbf{w}_i\| - \log |\mathbf{W}|$$

$$\square$$

## C.2 Proof of Prop. 4.6

**Proposition 4.6** (Properties of $C_{\text{IMA}}(\mathbf{f}, p_{\mathbf{s}})$)**.** *The global IMA contrast $C_{\text{IMA}}(\mathbf{f}, p_{\mathbf{s}})$ from* (9) *satisfies:*

*(i)* $C_{\text{IMA}}(\mathbf{f}, p_{\mathbf{s}}) \geq 0$, *with equality iff.* $\mathbf{J_f}(\mathbf{s}) = \mathbf{O}(\mathbf{s})\mathbf{D}(\mathbf{s})$ *almost surely w.r.t.* $p_{\mathbf{s}}$, *where* $\mathbf{O}(\mathbf{s}), \mathbf{D}(\mathbf{s}) \in \mathbb{R}^{n \times n}$ *are orthogonal and diagonal matrices, respectively;*

*(ii)* $C_{\text{IMA}}(\mathbf{f}, p_{\mathbf{s}}) = C_{\text{IMA}}(\tilde{\mathbf{f}}, p_{\tilde{\mathbf{s}}})$ *for any* $\tilde{\mathbf{f}} = \mathbf{f} \circ \mathbf{h}^{-1} \circ \mathbf{P}^{-1}$ *and* $\tilde{\mathbf{s}} = \mathbf{P}\mathbf{h}(\mathbf{s})$, *where* $\mathbf{P} \in \mathbb{R}^{n \times n}$ *is a permutation and* $\mathbf{h}(\mathbf{s}) = (h_1(s_1), ..., h_n(s_n))$ *an invertible element-wise function.*

*Proof.* The properties can be proved as follows:

(i) From property *(i)* of Prop. 4.4, we know that $c_{\text{IMA}}(\mathbf{f}, \mathbf{s}) \geq 0$. Hence, $C_{\text{IMA}}(\mathbf{f}, p(\mathbf{s})) \geq 0$ follows as a direct consequence of integrating the non-negative quantity $c_{\text{IMA}}(\mathbf{f}, \mathbf{s})$.

Equality is attained iff. $c_{\text{IMA}}(\mathbf{f}, \mathbf{s}) = 0$ almost surely w.r.t. $p_{\mathbf{s}}$, which according to property *(i)* of Prop. 4.4 occurs iff. the columns of $\mathbf{J_f}(\mathbf{s})$ are orthogonal almost surely w.r.t. $p_{\mathbf{s}}$.

It remains to show that this is the case iff. $\mathbf{J_f}(\mathbf{s})$ can be written as $\mathbf{O}(\mathbf{s})\mathbf{D}(\mathbf{s})$, with $\mathbf{O}(\mathbf{s})$ and $\mathbf{D}(\mathbf{s})$ orthogonal and diagonal matrices, respectively. (To avoid confusion, note that *orthogonal columns* need not have unit norm, whereas an *orthogonal matrix* $\mathbf{O}$ satisfies $\mathbf{O}\mathbf{O}^\top = \mathbf{I}$.)

The *if* is clear since right multiplication by a diagonal matrix merely re-scales the columns, and hence does not affect their orthogonality.

For the *only if*, let $\mathbf{J_f}(\mathbf{s})$ be any matrix with orthogonal columns $\mathbf{j}_i(\mathbf{s})$, $\mathbf{j}_i(\mathbf{s})^\top \mathbf{j}_j(\mathbf{s}) = 0, \forall i \neq j$, and denote the column norms by $d_i(\mathbf{s}) = \|\mathbf{j}_i(\mathbf{s})\|$. Further denote the normalised columns of $\mathbf{J_f}(\mathbf{s})$ by $\mathbf{o}_i(\mathbf{s}) = \mathbf{j}_i(\mathbf{s})/d_i(\mathbf{s})$ and let $\mathbf{O}(\mathbf{s})$ and $\mathbf{D}(\mathbf{s})$ be the orthogonal and diagonal matrices with columns $\mathbf{o}_i(\mathbf{s})$ and diagonal elements $d_i(\mathbf{s})$, respectively. Then $\mathbf{J_f}(\mathbf{s}) = \mathbf{O}(\mathbf{s})\mathbf{D}(\mathbf{s})$.

(ii) Let $\tilde{\mathbf{f}} = \mathbf{f} \circ \mathbf{h}^{-1} \circ \mathbf{P}^{-1}$ and $\tilde{\mathbf{s}} = \mathbf{P}\mathbf{h}(\mathbf{s})$, where $\mathbf{P} \in \mathbb{R}^{n \times n}$ is a permutation matrix and $\mathbf{h}(\mathbf{s}) = (h_1(s_1), ..., h_n(s_n))$ is an invertible element-wise function. Then

$$C_{\text{IMA}}(\tilde{\mathbf{f}}, p_{\tilde{\mathbf{s}}}) = \int c_{\text{IMA}}(\tilde{\mathbf{f}}, \tilde{\mathbf{s}}) p_{\tilde{\mathbf{s}}}(\tilde{\mathbf{s}}) d\tilde{\mathbf{s}} = \int c_{\text{IMA}}(\tilde{\mathbf{f}}, \tilde{\mathbf{s}}) p_{\mathbf{s}}(\mathbf{s}) d\mathbf{s} \tag{24}$$

where, for the second equality, we have used the fact that

$$p_{\tilde{\mathbf{s}}}(\tilde{\mathbf{s}}) d\tilde{\mathbf{s}} = p_{\mathbf{s}}(\mathbf{s}) d\mathbf{s} \,.$$

since $\mathbf{P} \circ \mathbf{h}$ is an invertible tranformation (see, e.g., [83]). It thus suffices to show that

$$c_{\text{IMA}}(\tilde{\mathbf{f}}, \tilde{\mathbf{s}}) = c_{\text{IMA}}(\mathbf{f}, \mathbf{s}). \tag{25}$$

at any point $\tilde{\mathbf{s}} = \mathbf{P}\mathbf{h}(\mathbf{s})$. To show this, we write

$$\mathbf{J}_{\tilde{\mathbf{f}}}(\tilde{\mathbf{s}}) = \mathbf{J}_{\mathbf{f} \circ \mathbf{h}^{-1} \circ \mathbf{P}^{-1}}(\mathbf{P}\mathbf{h}(\mathbf{s}))$$

$$= \mathbf{J}_{\mathbf{f} \circ \mathbf{h}^{-1}}\left(\mathbf{P}^{-1}\mathbf{P}\mathbf{h}(\mathbf{s})\right) \mathbf{J}_{\mathbf{P}^{-1}}\left(\mathbf{P}\mathbf{h}(\mathbf{s})\right)$$

$$= \mathbf{J}_{\mathbf{f} \circ \mathbf{h}^{-1}}(\mathbf{h}(\mathbf{s})) \mathbf{J}_{\mathbf{P}^{-1}}(\mathbf{P}\mathbf{h}(\mathbf{s}))$$

$$= \mathbf{J}_{\mathbf{f}}(\mathbf{h}^{-1} \circ \mathbf{h}(\mathbf{s})) \mathbf{J}_{\mathbf{h}^{-1}}(\mathbf{h}(\mathbf{s})) \mathbf{J}_{\mathbf{P}^{-1}}(\mathbf{P}\mathbf{h}(\mathbf{s}))$$

$$= \mathbf{J}_{\mathbf{f}}(\mathbf{s}) \, \mathbf{D}(\mathbf{s})\mathbf{P}^{-1} \tag{26}$$

where we have repeatedly used the chain rule for Jacobians, as well as that $\mathbf{P}^{-1}\mathbf{P} = \mathbf{I}$; that permutation is a linear operation, so $\mathbf{J_P}(\mathbf{s}) = \mathbf{P}$ for any $\mathbf{s}$; and that $\mathbf{h}$ (and thus $\mathbf{h}^{-1}$) is an element-wise transformation, so the Jacobian $\mathbf{J_{h^{-1}}}$ is a diagonal matrix $\mathbf{D}(\mathbf{s})$.

The equality in (25) then follows from (26) by applying property *(ii)* of Prop. 4.4, according to which $c_{\mathrm{IMA}}$ is invariant to right multiplication of the Jacobian $\mathbf{J_f}(\mathbf{s})$ by diagonal and permutation matrices.

Substituting (25) into the RHS of (24), we finally obtain

$$C_{\mathrm{IMA}}(\tilde{\mathbf{f}}, p_{\tilde{\mathbf{s}}}) = C_{\mathrm{IMA}}(\mathbf{f}, p_{\mathbf{s}}).$$

$\square$

### C.3 Remark on a similar condition to IMA, expressed in terms of the rows of the Jacobian

We remark that the condition imposed by the IMA Principle 4.1 needs to be expressed in terms of the columns of the Jacobian, and would not lead to a criterion with desirable properties for BSS if it were instead expressed in terms of its rows (which correspond to gradients of the $f_i(\mathbf{s})$). One way to justify this is that, for the same condition expressed on the rows of the Jacobian, that is

$$\sum_{i=1}^{n} \log \|\nabla f_i(\mathbf{s})\| - \log |\mathbf{J_f}(\mathbf{s})| = 0 \,,$$

property *(ii)* of Prop. 4.4 would not hold (because invariance would hold w.r.t. right, not left, multiplication with a diagonal matrix). As a consequence, the resulting global contrast would not be blind to reparametrisation of the source variables by permutation and element-wise invertible transformations, thereby not being a good contrast in the context of blind source separation.

### C.4 Proof of Thm. 4.7

Before proving the main theorem, we first introduce some additional details on the Jacobian of the Darmois construction [39] which will be important for the proof.

**Jacobian of the Darmois construction for $n = 2$.** Consider the Darmois construction for $n = 2$,

$$y_1 = g_1^{\mathrm{D}}(x_1) := F_{X_1}(x_1) = \mathbb{P}_{X_1}(X_1 \le x_1)$$
$$y_2 = g_2^{\mathrm{D}}(y_1, x_2) := F_{X_2|Y_1=y_1}(x_2) = \mathbb{P}_{X_2|Y_1=y_1}(X_2 \le x_2|Y_1 = y_1)$$

Its Jacobian takes the form

$$\mathbf{J_{g^D}}(\mathbf{x}) = \begin{pmatrix} p(x_1) & 0 \\ c_{21}(\mathbf{x}) & p(x_2|x_1) \end{pmatrix} \,, \tag{27}$$

where

$$c_{21}(\mathbf{x}) = \frac{\partial}{\partial x_1} \int_{-\infty}^{x_2} p(x_2'|x_1) dx_2' \,.$$

**Jacobian of the Darmois construction: general case.** In the general case, the Jacobian of the Darmois construction will be

$$\mathbf{J_{g^D}}(\mathbf{x}) = \begin{pmatrix} p(x_1) & \cdots & 0 \\ & \ddots & \vdots \\ \mathbf{C}(\mathbf{x}) & & p(x_n|x_1,\dots,x_{n-1}) \end{pmatrix} \tag{28}$$

where the components $c_{ji}(\mathbf{x}_{1:j})$ of $\mathbf{C}(\mathbf{x})$ for all $i < j$ are defined by

$$c_{ji}(\mathbf{x}_{1:j}) = \frac{\partial}{\partial x_i} \int_{-\infty}^{x_j} p(x_j'|\mathbf{x}_{1:j-1}) dx_j' \,.$$

It is additionally useful to introduce the following lemmas.

**Lemma C.1.** *A function $\mathbf{f}$ with triangular Jacobian has $C_{\mathrm{IMA}}(\mathbf{f}, p_{\mathbf{s}}) = 0$ iff. its Jacobian is diagonal almost everywhere. Otherwise, $C_{\mathrm{IMA}}(\mathbf{f}, p_{\mathbf{s}}) > 0$.*

*Proof.* Let $\mathbf{f}$ have lower triangular Jacobian at $\mathbf{s}$, and denote $\mathbf{J_f}(\mathbf{s}) = \mathbf{W}$. Then we have

$$c_{\mathrm{IMA}}(\mathbf{f}, \mathbf{s}) = \sum_{i=1}^{n} \log\left(\sqrt{\sum_{j=i}^{n} w_{ji}^2}\right) - \sum_{i=1}^{n} \log|w_{ii}|\,,$$

where $w_{ji} = [\mathbf{W}]_{ji}$. Since the logarithm is a strictly monotonically increasing function and since

$$\sqrt{\sum_{j=1}^{n} w_{ji}^2} \geq |w_{ii}|\,,$$

with equality iff. $w_{ji} = 0, \forall j \neq i$ (i.e., iff. $\mathbf{W}$ is a diagonal matrix), we must have $c_{\mathrm{IMA}}(\mathbf{f}, \mathbf{s}) = 0$ iff. $\mathbf{W}$ is diagonal.

$C_{\mathrm{IMA}}(\mathbf{f}, p_{\mathbf{s}})$ is therefore equal to zero iff. $\mathbf{f}$ has diagonal Jacobian almost everywhere, and it is strictly larger than zero otherwise. $\qquad \square$

**Lemma C.2.** *A smooth function $\mathbf{f} : \mathbb{R}^n \to \mathbb{R}^n$ whose Jacobian is diagonal everywhere is an element-wise function, $\mathbf{f}(\mathbf{s}) = (f_1(s_1), ..., f_n(s_n))$.*

*Proof.* Let $\mathbf{f}$ be a smooth function with diagonal Jacobian everywhere.

Consider the function $f_i(\mathbf{s})$ for any $i \in \{1, ..., n\}$. Suppose *for a contradiction* that $f_i$ depends on $s_j$ for some $j \neq i$. Then there must be at least one point $\mathbf{s}^*$ such that $\partial f_i / \partial s_j(\mathbf{s}^*) \neq 0$. However, this contradicts the assumption that $\mathbf{J_f}$ is diagonal everywhere (since $\partial f_i / \partial s_j$ is an off-diagonal element for $i \neq j$). Hence, $f_i$ can only depend on $s_i$ for all $i$, i.e., $\mathbf{f}$ is an element wise function. $\qquad \square$

We can now restate and prove Thm. 4.7.

**Theorem 4.7.** *Assume the data generating process in* (1) *and assume that $x_i \not\perp\!\!\!\perp x_j$ for some $i \neq j$. Then any Darmois solution $(\mathbf{f}^D, p_{\mathbf{u}})$ based on $\mathbf{g}^D$ as defined in* (4) *satisfies $C_{\mathrm{IMA}}(\mathbf{f}^D, p_{\mathbf{u}}) > 0$. Thus a solution satisfying $C_{\mathrm{IMA}}(\mathbf{f}, p_{\mathbf{s}}) = 0$ can be distinguished from $(\mathbf{f}^D, p_{\mathbf{u}})$ based on the contrast $C_{\mathrm{IMA}}$.*

*Proof.* First, the Jacobian $\mathbf{J}_{\mathbf{g}^D}(\mathbf{x})$ of the Darmois construction $\mathbf{g}^D$ is lower triangular $\forall \mathbf{x}$, see (28).

Because CDFs are monotonic functions (strictly monotonically increasing given our assumptions on $\mathbf{f}$ and $p_{\mathbf{s}}$), $\mathbf{g}^D$ is invertible.

We can thus apply the inverse function theorem (with $\mathbf{f}^D = (\mathbf{g}^D)^{-1}$) to write

$$\mathbf{J}_{\mathbf{f}^D}(\mathbf{y}) = \left(\mathbf{J}_{\mathbf{g}^D}(\mathbf{x})\right)^{-1}$$

Since the inverse of a lower triangular matrix is lower triangular, we conclude that $\mathbf{J}_{\mathbf{f}^D}(\mathbf{y})$ is lower triangular for all $\mathbf{y} = \mathbf{g}^D(\mathbf{x})$.

Now, according to Lemma C.1, we have $C_{\mathrm{IMA}}(\mathbf{f}^D, p_{\mathbf{u}}) > 0$, unless $\mathbf{J}_{\mathbf{f}^D}$ is diagonal almost everywhere.

Suppose *for a contradiction* that $\mathbf{J}_{\mathbf{f}^D}$ is diagonal almost everywhere.

Since $\mathbf{f}$ and $p_{\mathbf{s}}$ are smooth by assumption, so is the push-forward $p_{\mathbf{x}} = \mathbf{f}_* p_{\mathbf{s}}$, and thus also $\mathbf{g}^D$ (CDF of a smooth density) and its inverse $\mathbf{f}^D$. Hence, the partial derivatives $\partial f_i^D / \partial y_j$, i.e., the elements of $\mathbf{J}_{\mathbf{f}^D}$ are continuous.

Consider an off-diagonal element $\partial f_i^D / \partial y_j$ for $i \neq j$. Since these are zero almost everywhere, and because continuous functions which are zero almost everywhere must be zero everywhere, we conclude that $\partial f_i^D / \partial y_j = 0$ everywhere for $i \neq j$, i.e., the Jacobian $\mathbf{J}_{\mathbf{f}^D}$ is *diagonal everywhere*.

Hence, we conclude from Lemma C.2 that $\mathbf{f}^D$ must be an element-wise function, $\mathbf{f}^D(\mathbf{y}) = (f_1^D(y_1), ..., f_1^D(y_n))$.

Since $\mathbf{y}$ has independent components by construction, it follows that $x_i = f_i^D(y_i)$ and $x_j = f_j^D(y_j)$ are independent for any $i \neq j$.

However, this constitutes a contradiction to the assumption that $x_i \not\perp\!\!\!\perp x_j$ for some $x_j$.

We conclude that $\mathbf{J}_{\mathbf{f}^D}$ cannot be diagonal almost everywhere, and hence, by Lemma C.1, we must have $C_{\mathrm{IMA}}(\mathbf{f}^D, p_{\mathbf{u}}) > 0$. $\qquad \square$

### C.5 Proof of Corollary 4.9

**Corollary 4.9.** *Under assumptions of Thm. 4.7, if additionally* $\mathbf{f}$ *is a conformal map, then* $(\mathbf{f}, p_{\mathbf{s}}) \in \mathcal{M}_{\mathrm{IMA}}$ *for any* $p_{\mathbf{s}} \in \mathcal{P}$ *due to Prop. 4.6 (i), see Defn. 4.8. Based on Thm. 4.7,* $(\mathbf{f}, p_{\mathbf{s}})$ *is thus distinguishable from Darmois solutions* $(\mathbf{f}^D, p_{\mathbf{u}})$.

*Proof.* The proof follows from property *(i)* of Prop. 4.6: by definition, the Jacobian of conformal maps at any point $\mathbf{s}$ can be written as $\mathbf{O}(\mathbf{s})\lambda(\mathbf{s})$, with $\lambda : \mathbb{R}^n \to \mathbb{R}$, which is a special case of $\mathbf{O}(\mathbf{s})\mathbf{D}(\mathbf{s})$, with $\mathbf{D}(\mathbf{s}) = \lambda(\mathbf{s})\mathbf{I}$. $\qquad \square$

### C.6 Proof of Corollary 4.11

**Corollary 4.11.** *Consider a linear ICA model,* $\mathbf{x} = \mathbf{A}\mathbf{s}$*, with* $\mathbb{E}[\mathbf{s}^\top \mathbf{s}] = \mathbf{I}$*, and* $\mathbf{A} \in O(n)$ *an orthogonal, non-trivial mixing matrix, i.e., not the product of a diagonal and a permutation matrix* $\mathbf{DP}$*. If at most one of the* $s_i$ *is Gaussian, then* $C_{\mathrm{IMA}}(\mathbf{A}, p_{\mathbf{s}}) = 0$ *and* $C_{\mathrm{IMA}}(\mathbf{f}^D, p_{\mathbf{u}}) > 0$.

*Proof.* Since, by assumption, the mixing matrix is non-trivial (i.e., not the product of a diagonal and permutation matrix), and at most one of the $s_i$ is Gaussian, according to Thm. A.1 there must be at least one pair $x_i, x_j$, with $i \neq j$, such that $x_i \not\!\perp\!\!\!\perp x_j$.

We can then use the same argument as in the proof of Thm. 4.7 to show that the Darmois construction has nonzero $C_{\mathrm{IMA}}$, whereas the linear orthogonal transformation $\mathbf{A}$ has orthogonal Jacobian, and thus $C_{\mathrm{IMA}} = 0$ by property *(i)* of Prop. 4.6. $\qquad \square$

### C.7 Proof of Thm. 4.12

**Theorem 4.12.** *Let* $(\mathbf{f}, p_{\mathbf{s}}) \in \mathcal{M}_{\mathrm{IMA}}$ *and assume that* $\mathbf{f}$ *is a conformal map. Given* $\mathbf{R} \in O(n)$*, assume additionally that* $\exists$ *at least one non-Gaussian* $s_i$ *whose associated canonical basis vector* $\mathbf{e}_i$ *is not transformed by* $\mathbf{R}^{-1} = \mathbf{R}^\top$ *into another canonical basis vector* $\mathbf{e}_j$*. Then* $C_{\mathrm{IMA}}(\mathbf{f} \circ \mathbf{a}^{\mathbf{R}}(p_{\mathbf{s}}), p_{\mathbf{s}}) > 0$.

*Proof.* Recall the definition

$$\mathbf{a}^{\mathbf{R}}(p_{\mathbf{s}}) = \mathbf{F}_{\mathbf{s}}^{-1} \circ \mathbf{\Phi} \circ \mathbf{R} \circ \mathbf{\Phi}^{-1} \circ \mathbf{F}_{\mathbf{s}}.$$

For notational convenience, we denote $\boldsymbol{\sigma} = \mathbf{\Phi}^{-1} \circ \mathbf{F}_{\mathbf{s}}$ and write

$$\mathbf{a}^{\mathbf{R}}(p_{\mathbf{s}}) = \boldsymbol{\sigma}^{-1} \circ \mathbf{R} \circ \boldsymbol{\sigma}.$$

Note that, since both $\mathbf{F}_{\mathbf{s}}$ and $\mathbf{\Phi}$ are element-wise transformations, so is $\boldsymbol{\sigma}$.

First, by using property *(ii)* of Prop. 4.6 (invariance of $C_{\mathrm{IMA}}$ to element-wise transformation), we obtain

$$C_{\mathrm{IMA}}(\mathbf{f} \circ \mathbf{a}^{\mathbf{R}}(p_{\mathbf{s}}), p_{\mathbf{s}}) = C_{\mathrm{IMA}}(\mathbf{f} \circ \boldsymbol{\sigma}^{-1} \circ \mathbf{R} \circ \boldsymbol{\sigma}, p_{\mathbf{s}}) = C_{\mathrm{IMA}}(\mathbf{f} \circ \boldsymbol{\sigma}^{-1} \circ \mathbf{R}, p_{\mathbf{z}}),$$

with $\mathbf{z} = \boldsymbol{\sigma}(\mathbf{s})$ such that $p_{\mathbf{z}}$ is an isotropic Gaussian distribution.

Suppose *for a contradiction* that $C_{\mathrm{IMA}}(\mathbf{f} \circ \boldsymbol{\sigma}^{-1} \circ \mathbf{R}, p_{\mathbf{z}}) = 0$.

According to property *(i)* of Prop. 4.6, this entails that the matrix

$$\mathbf{J}_{\mathbf{f} \circ \boldsymbol{\sigma}^{-1} \circ \mathbf{R}}(\mathbf{z})^\top \mathbf{J}_{\mathbf{f} \circ \boldsymbol{\sigma}^{-1} \circ \mathbf{R}}(\mathbf{z}) = \mathbf{R}^\top \mathbf{J}_{\boldsymbol{\sigma}^{-1}}(\mathbf{z})^\top \mathbf{J}_{\mathbf{f}}(\boldsymbol{\sigma}^{-1}(\mathbf{z}))^\top \mathbf{J}_{\mathbf{f}}(\boldsymbol{\sigma}^{-1}(\mathbf{z})) \mathbf{J}_{\boldsymbol{\sigma}^{-1}}(\mathbf{z}) \mathbf{R} \qquad (29)$$

is diagonal almost surely w.r.t. $p_{\mathbf{z}}$. Moreover, smoothness of $p_{\mathbf{s}}$ and $\mathbf{f}$ implies the matrix expression of (29) is a continuous function of $\mathbf{z}$. Thus (29) actually needs to be diagonal for all $\mathbf{z} \in \mathbb{R}^n$, i.e., *everywhere* (c.f., the argument used in the proof of Thm. 4.7, l.1008–1013).

Since $(\mathbf{f}, p_{\mathbf{s}}) \in \mathcal{M}_{\mathrm{IMA}}$ by assumption, by property *(i)* of Prop. 4.6, the inner term on the RHS of (29),

$$\mathbf{J}_{\mathbf{f}}(\boldsymbol{\sigma}^{-1}(\mathbf{z}))^\top \mathbf{J}_{\mathbf{f}}(\boldsymbol{\sigma}^{-1}(\mathbf{z})),$$

is diagonal. Moreover, since $\boldsymbol{\sigma}$ is an element-wise transformation, $\mathbf{J}_{\boldsymbol{\sigma}^{-1}}(\mathbf{z})^\top$ and $\mathbf{J}_{\boldsymbol{\sigma}^{-1}}(\mathbf{z})$ are also diagonal. Taken together, this implies that

$$\mathbf{J}_{\boldsymbol{\sigma}^{-1}}(\mathbf{z}) \mathbf{J}_{\mathbf{f}}(\boldsymbol{\sigma}^{-1}(\mathbf{z}))^\top \mathbf{J}_{\mathbf{f}}(\boldsymbol{\sigma}^{-1}(\mathbf{z})) \mathbf{J}_{\boldsymbol{\sigma}^{-1}}(\mathbf{z}) \qquad (30)$$

is diagonal (i.e., (29) is of the form $\mathbf{R}^\top \mathbf{D}(\mathbf{z})\mathbf{R}$ for some diagonal matrix $\mathbf{D}(\mathbf{z})$).

Without loss of generality, we assume the first component $s_1$ of $\mathbf{s}$ is non-Gaussian and satisfies the assumptions stated relative to $\mathbf{R}$ (axis not invariant nor sent to another canonical axis).

Now, since both the Gaussian CDF $\Phi$ and the CDF $\mathbf{F_s}$ are smooth (the latter by the assumption that of $p_\mathbf{s}$ is a smooth density), $\boldsymbol{\sigma}$ is a smooth function, and thus has continuous partial derivatives.

By continuity of the partial derivative, the first diagonal element $\frac{\partial \sigma_1^{-1}}{\partial z_1}$ of $\mathbf{J}_{\boldsymbol{\sigma}^{-1}}$ must be strictly monotonic in a neighborhood of some $z_1^0$ (otherwise $\sigma_1$ would be an affine transformation, which would contradict non-Gaussianity of $s_1$).

On the other hand, our assumptions relative to $\mathbf{R}$ entail that there are at least two non-vanishing coefficients in the first row of $\mathbf{R}$ (i.e., first column of $\mathbf{R}^\top$).[20] Let us call $i \neq j$ such pair of coordinates, i.e., $r_{1j} \neq 0$ and $r_{1i} \neq 0$.

Now consider the off-diagonal term $(i, j)$ of (29), which we assumed (for a contradiction) must be zero almost surely w.r.t. $p_\mathbf{z}$. Since the term in (30) is diagonal, this off-diagonal term is given by:

$$\sum_{k=1}^{n} \left( \frac{d\sigma_k^{-1}}{dz_k}(z_k) \right)^2 \left\| \frac{\partial \mathbf{f}}{\partial s_k} \circ \boldsymbol{\sigma}^{-1}(\mathbf{z}) \right\|^2 r_{ki} r_{kj} = \sum_{k=1}^{n} \left( \frac{d\sigma_k^{-1}}{dz_k}(z_k) \right)^2 \lambda(\boldsymbol{\sigma}^{-1}(\mathbf{z}))^2 r_{ki} r_{kj} = 0 \,.$$

where for the first equality we have used the fact that $\mathbf{f}$ is a conformal map with conformal factor $\lambda(\mathbf{s})$ (by assumption), and where the second equality must hold almost surely w.r.t. $p_\mathbf{z}$.

Since $\mathbf{f}$ is invertible, it has non vanishing Jacobian determinant. Hence, the conformal factor $\lambda$ must be a strictly positive function, so

$$\lambda(\boldsymbol{\sigma}^{-1}(\mathbf{z}))^2 > 0, \, \forall \mathbf{z} .$$

Thus, for almost all $\mathbf{z}$, we must have:

$$\sum_{k=1}^{n} \left( \frac{d\sigma_k^{-1}}{dz_k}(z_k) \right)^2 r_{ki} r_{kj} = 0 \,. \tag{31}$$

Now consider the first term $\left( \frac{d\sigma_1^{-1}}{dz_1}(z_1) \right)^2 r_{1i} r_{1j}$ in the sum.

Recall that $r_{1i} r_{1j} \neq 0$, and that $\frac{d\sigma_1^{-1}}{dz_1}(z_1)$ is strictly monotonic on a neighborhood of $z_1^0$.

As a consequence, $\left( \frac{d\sigma_1^{-1}}{dz_1}(z_1) \right)^2 r_{1i} r_{1j}$ is also strictly monotonic with respect to $z_1$ on a neighborhood of $z_1^0$ (where the other variables $(z_2, ..., z_n)$ are left constant), while the other terms in the sum in (31) are left constant because $\boldsymbol{\sigma}$ is an element-wise transformation.

This leads to a contradiction as (31) (which should be satisfied for all $\mathbf{z}$) cannot stay constantly zero as $z_1$ varies within the neighbourhood of $z_1^0$.

Hence our assumption that $C_{\mathrm{IMA}}(\mathbf{f} \circ \mathbf{a}^\mathbf{R}(p_\mathbf{s}), p_\mathbf{s}) = 0$ cannot hold.

We conclude that $C_{\mathrm{IMA}}(\mathbf{f} \circ \mathbf{a}^\mathbf{R}(p_\mathbf{s}), p_\mathbf{s}) > 0$. □

---

[20]In short, if this were not the case, this column would have a single non-vanishing coefficient, which would need to be one due to the unit norm of the rows of this orthogonal matrix. Such structure of the matrix $\mathbf{R}$ would entail that the associated canonical basis vector $\mathbf{e}_1$ is transformed by $\mathbf{R}^{-1} = \mathbf{R}^\top$ into a canonical basis vector $\mathbf{e}_j$ which contradicts the assumptions.

## D Worked out example

*Example* D.1 (Polar to Cartesian coordinates). Consider the following example of a nonlinear ICA model which represents a change of basis from polar to Cartesian coordinates:

$$\mathbf{x} = \begin{pmatrix} x_1 \\ x_2 \end{pmatrix} = \mathbf{f}(\mathbf{s}) = \begin{pmatrix} f_1(\mathbf{s}) \\ f_2(\mathbf{s}) \end{pmatrix} = \begin{pmatrix} r\cos(\theta) \\ r\sin(\theta) \end{pmatrix}$$

with sources

$$\mathbf{s} = \begin{pmatrix} s_1 \\ s_2 \end{pmatrix} = \begin{pmatrix} r \\ \theta \end{pmatrix}, \qquad r \sim U[0, R], \qquad \theta \sim U[0, 2\pi],$$

First, we consider the Jacobian of the true mixing $\mathbf{f}$ which is given by:

$$\mathbf{J_f}(\mathbf{s}) = \mathbf{J_f}(r, \theta) = \begin{pmatrix} \cos(\theta) & -r\sin(\theta) \\ \sin(\theta) & r\cos(\theta) \end{pmatrix},$$

and its determinant and column norms are given by

$$|\det \mathbf{J_f}(\mathbf{s})| = r\left(\cos^2(\theta) + \sin^2(\theta)\right) = r$$

$$\left\|\frac{\partial \mathbf{f}}{\partial s_1}(\mathbf{s})\right\| = \left\|\frac{\partial \mathbf{f}}{\partial r}(r, \theta)\right\| = \cos^2(\theta) + \sin^2(\theta) = 1$$

$$\left\|\frac{\partial \mathbf{f}}{\partial s_2}(\mathbf{s})\right\| = \left\|\frac{\partial \mathbf{f}}{\partial \theta}(r, \theta)\right\| = r\left(\cos^2(\theta) + \sin^2(\theta)\right) = r$$

In other words, the columns of $\mathbf{J_f}(\mathbf{s})$ are orthogonal for all $\mathbf{s}$, so that $C_{\text{IMA}} = 0$ for the true solution.

Next, we apply the Darmois construction.

First, we write the joint density of $(x_1, x_2)$ using the change of variable formula:

$$p(x_1, x_2) = |\det \mathbf{J_f}(r, \theta)|^{-1} p(r, \theta) = r^{-1}\frac{1}{2\pi R} = \frac{1}{\sqrt{x_1^2 + x_2^2}}\frac{1}{2\pi R}.$$

Next, we compute the marginal density $p(x_1)$. Note that the observations $\mathbf{x}$ live on the disk of radius $R$, $\|\mathbf{x}\| \leq R$, so $p(x_1, x_2) = 0$ whenever $x_1^2 + x_2^2 > R^2$.

$$p(x_1) = \int_{-\sqrt{R^2-x_1^2}}^{\sqrt{R^2-x_1^2}} p(x_1, x_2)dx_2 = \frac{1}{2\pi R}\int_{-\sqrt{R^2-x_1^2}}^{\sqrt{R^2-x_1^2}} \frac{dx_2}{\sqrt{x_1^2 + x_2^2}} = \frac{1}{2\pi R}\int_{-\sqrt{R^2-x_1^2}}^{\sqrt{R^2-x_1^2}} \frac{dx_2}{x_1\sqrt{1 + (\frac{x_2}{x_1})^2}}$$

Applying the change of variable $t = \frac{x_2}{x_1}$ with $dt = \frac{dx_2}{x_1}$, and using the integral $\int (1 + t^2)^{-\frac{1}{2}}dt = \text{arcsinh}(t) + C$, as well as the fact that $\text{arcsinh}$ is an odd function, we obtain

$$p(x_1) = \frac{1}{2\pi R}\int_{-\sqrt{\left(\frac{R}{x_1}\right)^2-1}}^{\sqrt{\left(\frac{R}{x_1}\right)^2-1}} \frac{dt}{\sqrt{1 + t^2}} = \frac{1}{\pi R}\text{arcsinh}\left(\sqrt{\left(\frac{R}{x_1}\right)^2 - 1}\right)$$

Next, we compute the conditional density $p(x_2|x_1)$:

$$p(x_2|x_1) = \frac{p(x_1, x_2)}{p(x_1)} = \frac{(2\pi R)^{-1}\left(x_1^2 + x_2^2\right)^{-1}}{(\pi R)^{-1}\text{arcsinh}\left(\sqrt{\left(\frac{R}{x_1}\right)^2 - 1}\right)} = \left(2\sqrt{x_1^2 + x_2^2}\,\text{arcsinh}\left(\sqrt{\left(\frac{R}{x_1}\right)^2 - 1}\right)\right)^{-1}$$

Finally, we compute the off-diagonal term in the general form of the inverse Jacobian for Damois-style solutions in (27):

$$c_{21}(\mathbf{x}) = \frac{\partial}{\partial x_1} \int_{-\infty}^{x_2} p(x_2|x_1)dx_2 = \frac{\partial}{\partial x_1} \int_{-\sqrt{R^2-x_1^2}}^{x_2} \frac{dx_2}{2\sqrt{x_1^2+x_2^2}\,\operatorname{arcsinh}\left(\sqrt{\left(\frac{R}{x_1}\right)^2-1}\right)}$$

$$= \frac{1}{2}\frac{\partial}{\partial x_1}\left(\operatorname{arcsinh}\left(\sqrt{\left(\frac{R}{x_1}\right)^2-1}\right)^{-1} \int_{-\sqrt{R^2-x_1^2}}^{x_2} \frac{dx_2}{\sqrt{x_1^2+x_2^2}}\right)$$

$$= \frac{1}{2}\frac{\partial}{\partial x_1}\left(\operatorname{arcsinh}\left(\sqrt{\left(\frac{R}{x_1}\right)^2-1}\right)^{-1}\left(\operatorname{arcsinh}(x_2) - \operatorname{arcsinh}\left(-\sqrt{\left(\frac{R}{x_1}\right)^2-1}\right)\right)\right)$$

$$= \frac{1}{2}\frac{\partial}{\partial x_1}\left(\operatorname{arcsinh}\left(\sqrt{\left(\frac{R}{x_1}\right)^2-1}\right)^{-1}\left(\operatorname{arcsinh}(x_2) + \operatorname{arcsinh}\left(\sqrt{\left(\frac{R}{x_1}\right)^2-1}\right)\right)\right)$$

$$= \frac{1}{2}\frac{\partial}{\partial x_1}\left(1 + \frac{\operatorname{arcsinh}(x_2)}{\operatorname{arcsinh}\left(\sqrt{\left(\frac{R}{x_1}\right)^2-1}\right)}\right)$$

$$= \frac{1}{2}\operatorname{arcsinh}(x_2)\frac{\partial}{\partial x_1}\left(\operatorname{arcsinh}\left(\sqrt{\left(\frac{R}{x_1}\right)^2-1}\right)^{-1}\right)$$

$$= -\frac{1}{2}\operatorname{arcsinh}(x_2)\operatorname{arcsinh}\left(\sqrt{\left(\frac{R}{x_1}\right)^2-1}\right)^{-2}\frac{\partial}{\partial x_1}\operatorname{arcsinh}\left(\sqrt{\left(\frac{R}{x_1}\right)^2-1}\right)$$

Using the derivative $\frac{\partial}{\partial t}\operatorname{arcsinh}(t) = (t^2+1)^{-\frac{1}{2}}$ and repeatedly applying the chain rule, we obtain:

$$c_{21}(\mathbf{x}) = -\frac{1}{2}\operatorname{arcsinh}(x_2)\operatorname{arcsinh}\left(\sqrt{\left(\frac{R}{x_1}\right)^2-1}\right)^{-2}\frac{x_1}{R}\frac{\partial}{\partial x_1}\left(\sqrt{\left(\frac{R}{x_1}\right)^2-1}\right)$$

$$= -\frac{1}{2}\operatorname{arcsinh}(x_2)\operatorname{arcsinh}\left(\sqrt{\left(\frac{R}{x_1}\right)^2-1}\right)^{-2}\frac{x_1}{R}\frac{1}{2}\frac{1}{\sqrt{\left(\frac{R}{x_1}\right)^2-1}}(-2)R^2 x_1^{-3}$$

$$= \frac{R}{2x_1\sqrt{R^2-x_1^2}}\operatorname{arcsinh}(x_2)\operatorname{arcsinh}\left(\sqrt{\left(\frac{R}{x_1}\right)^2-1}\right)^{-2}$$

Again, recall that this only holds inside the disk of radius $R$, otherwise $c_{12} = 0$ (as the CDF will be zero or one, irrespective of $x_1$).

The $C_{\text{IMA}}$ for the Darmois solution thus takes the form:

$$C_{\text{IMA}}^{\text{Darmois}} = \int \frac{1}{2} \log \left( p(x_1)^{-2} + c_{21}(\mathbf{x})^2 p(x_1, x_2)^{-2} \right) + \log \left( p(x_2|x_1)^{-1} \right) - \log \left( p(x_1, x_2)^{-1} \right) d\mathbf{s}$$

$$= \int \frac{1}{2} \log \left[ \left( \frac{1}{\pi R} \operatorname{arcsinh} \left( \sqrt{\left( \frac{R}{x_1} \right)^2 - 1} \right) \right)^{-2} \right.$$

$$+ \left( \frac{R}{2x_1 \sqrt{R^2 - x_1^2}} \operatorname{arcsinh}(x_2) \operatorname{arcsinh} \left( \sqrt{\left( \frac{R}{x_1} \right)^2 - 1} \right)^{-2} \right)^2 \left( \frac{1}{\sqrt{x_1^2 + x_2^2}} \frac{1}{2\pi R} \right)^{-2} \right]$$

$$+ \log \left( 2 \sqrt{x_1^2 + x_2^2} \operatorname{arcsinh} \left( \sqrt{\left( \frac{R}{x_1} \right)^2 - 1} \right) \right)$$

$$- \log (2\pi R) - \frac{1}{2} \log \left( x_1^2 + x_2^2 \right) d\mathbf{s}$$

$$= \int \frac{1}{2} \log \left[ \pi^2 R^2 \operatorname{arcsinh} \left( \sqrt{\left( \frac{R}{x_1} \right)^2 - 1} \right)^{-2} \right.$$

$$\left. + \frac{R^2}{4x_1^2(R^2 - x_1^2)} \operatorname{arcsinh}(x_2)^2 \operatorname{arcsinh} \left( \sqrt{\left( \frac{R}{x_1} \right)^2 - 1} \right)^{-4} (x_1^2 + x_2^2) 4\pi^2 R^2 \right]$$

$$+ \log(2) + \frac{1}{2} \log(x_1^2 + x_2^2) + \log \left( \operatorname{arcsinh} \left( \sqrt{\left( \frac{R}{x_1} \right)^2 - 1} \right) \right)$$

$$- \log(2) - \log (\pi R) - \frac{1}{2} \log \left( x_1^2 + x_2^2 \right) d\mathbf{s}$$

$$= \int \frac{1}{2} \log \left[ \pi^2 R^2 \operatorname{arcsinh} \left( \sqrt{\left( \frac{R}{x_1} \right)^2 - 1} \right)^{-2} \right.$$

$$\left. + \frac{\pi^2 R^4 (x_1^2 + x_2^2)}{x_1^2(R^2 - x_1^2)} \operatorname{arcsinh}(x_2)^2 \operatorname{arcsinh} \left( \sqrt{\left( \frac{R}{x_1} \right)^2 - 1} \right)^{-4} \right]$$

$$+ \log \left( \operatorname{arcsinh} \left( \sqrt{\left( \frac{R}{x_1} \right)^2 - 1} \right) \right) - \log (\pi R) d\mathbf{s}$$

$$= \int \frac{1}{2} \log \left( 1 + \frac{R^2 (x_1^2 + x_2^2) \operatorname{arcsinh}(x_2)^2}{x_1^2 (R^2 - x_1^2) \operatorname{arcsinh} \left( \sqrt{\left( \frac{R}{x_1} \right)^2 - 1} \right)^2} \right) d\mathbf{s} > 0$$

where the strict inequality in the last step follows from the fact that the fraction inside the logarithm, and hence the entire integrand, is strictly positive within the disk of integration.

We have thus shown that for the example of an orthogonal coordinate transformation from polar to Cartesian coordinates, which is not a conformal map, the $C_{\text{IMA}}$ os the true solution is zero and that of the Darmois construction is strictly greater than zero, hence the two can be distinguished based on the value of the $C_{\text{IMA}}$ contrast.

# E  Experiments

The code for our experiments (enclosed in the supplemental material) is in Python; we use Jax [12], Distrax [13] and Haiku [32] to implement our models; the Jacobian and $C_{\text{IMA}}$ computation and optimisation are performed with the automatic differentiation tools provided in Jax.

## E.1  Sampling random Möbius transformations.

In order to generate mixing functions with $C_{\text{IMA}} = 0$, we use Möbius transformations (see Appendix F and in particular Thm. F.2, for additional details on this kind of functions) with randomly sampled parameters, as specified below. A Möbius transformation $\mathbf{f}^{\text{M}} : \mathbb{R}^n \to \mathbb{R}^n$ is given by

$$\mathbf{f}^{\text{M}}(\mathbf{s}) = \mathbf{t} + \frac{r\mathbf{A}(\mathbf{s} - \mathbf{b})}{\|\mathbf{s} - \mathbf{b}\|^{\epsilon}}, \tag{32}$$

with parameters $\mathbf{b}, \mathbf{t} \in \mathbb{R}^n$, $r \in \mathbb{R}$, $\mathbf{A}$ is an orthogonal matrix and $\epsilon \in \{0, 2\}$. The flow models we train have an diagonal affine layer at the top with fixed shift and scale set to the mean and standard deviation of the training data, thereby normalizing the inputs. Hence, without loss of generality, we can set the $\mathbf{t}$ parameter to zero and $r$ to one. Since $\epsilon = 0$ corresponds to a linear transformation, we generally set $\epsilon = 2$ in our experiments unless otherwise specified. We sample the orthogonal matrix through the `ortho_group` function in `scipy.stats` [101]. To avoid singularities given by a vanishing denominator in the second term on the RHS of (32), which would yield observed distributions with strong outliers and therefore hard to fit for our models, we restrict $\mathbf{b}$ to lie outside the unit square $\mathbf{s}$ is sampled from. We achieve this by sampling $\mathbf{b}$ from a normal distribution and reject the sample until it is located outside of the unit square.

## E.2  How to implement the Darmois construction

In the following, we describe how the Darmois construction can be implemented based on normalising flow models [69]. The key idea is that the components $g_i^{\text{D}}$ of the Darmois construction (4) are conditional (cumulative) density functions corresponding to a given factorisation $p(\mathbf{x}) = \prod_{i=1}^{n} p(x_i|\mathbf{x}_{1:i-1})$ of the likelihood. A flow model with triangular Jacobian can be used to maximise the likelihood of the observations under a change of variable respecting said factorisation, and learning to map the observed variables onto a given (factorised) base distribution. After training, and provided that the model is expressive enough, the CDF of each component of the reconstructed sources should match that of the base distribution. By further transforming each reconstructed variable through said CDF, we achieve a global mapping of the observations onto a Uniform distribution on the $n$-dimensional hypercube, with a triangular Jacobian, matching the transformation operated by the Darmois construction (see also see [69], section 2.2). Note that, for the purpose of computing the $C_{\text{IMA}}$ of the Darmois construction, this final step can be omitted due to Prop. 4.6, *(ii)*, stating that the contrast is blind to element-wise reparametrisations of the sources.

We remark that, while the possibility of using normalising flows to "learn" the Darmois construction is mentioned in [35, 69], where a similar construction is mentioned in a theoretical argument to prove "universal approximation capacity for densities" for normalising flow models with triangular Jacobian, it has to the best of our knowledge not been tested empirically, since autoregressive modules with triangular Jacobian are typically used in combination with permutation, shuffling or linear layers which overall lead to architectures with a non-triangular Jacobian.

**Expressive normalising flow with triangular Jacobian.**  To obtain an expressive normalizing flow with triagular Jacobian, we modify the residual flow model [16].[21] A residual flow is a residual network which is made invertible through spectral normalization. Each layer is given by

$$\mathbf{z}' = \mathbf{z} + \mathbf{g}(\mathbf{z}), \tag{33}$$

where $\mathbf{z}', \mathbf{z} \in \mathbb{R}^n$ and $\mathbf{g} : \mathbb{R}^n \to \mathbb{R}^n$ is a small neural network. Due to the chain rule, for the Jacobian of the overall flow model to be triangular, a sufficient condition is that all the layers have triangular Jacobian. Since the Jacobian of $\mathbf{f}(\mathbf{z}) = \mathbf{z}$ is the identity matrix, we can restrict our attention to the neural network $\mathbf{g}$. In our experiments, this is going to be a fully connected network. If it has $l$ layers and $h \geq n$ hidden units, it is given by

$$\mathbf{g}(\mathbf{z}) = \mathbf{b}_1 + \mathbf{W}_1\phi(\mathbf{b}_2 + \mathbf{W}_2\phi(\mathbf{b}_3 + \mathbf{W}_3 \cdots \phi(\mathbf{b}_l + \mathbf{W}_l\mathbf{z})\cdots)), \tag{34}$$

---

[21]We describe how to implement a function with upper triangular Jacobian, but the reasoning can be extended to implement functions whose Jacobian is lower triangular.

where $\phi : \mathbb{R}^n \to \mathbb{R}^n$ is an element-wise nonlinearity, $\mathbf{b}_1 \in \mathbb{R}^n$, $\mathbf{b}_2, ..., \mathbf{b}_l \in \mathbb{R}^h$ are the biases, and $\mathbf{W}_1 \in \mathbb{R}^{n \times h}$, $\mathbf{W}_2, ..., \mathbf{W}_{l-1} \in \mathbb{R}^{h \times h}$, $\mathbf{W}_l \in \mathbb{R}^{h \times n}$ are the weight matrices. In order for the Jacobian of $\mathbf{g}$ to be triangular, $g_n(\mathbf{z})$ should only depend on $z_n$, $g_{n-1}(\mathbf{z})$ should only depend on $z_n$ and $z_{n-1}$, and so on. To achieve this, we make the weight matrices block triangular as indicated in (35), (36), and (37).

$$\mathbf{W}_1 = \begin{pmatrix} * & * & & * \\ \vdots & \vdots & & \vdots \\ * & * & & * \\ 0 & * & & * \\ \vdots & \vdots & & \vdots \\ 0 & * & & * \\ & & \ddots & \\ 0 & 0 & & * \\ \vdots & \vdots & & \vdots \\ 0 & 0 & & * \end{pmatrix} \begin{matrix} \left.\vphantom{\begin{matrix}*\\\vdots\\ *\end{matrix}}\right\} h_1 \\ \left.\vphantom{\begin{matrix}*\\\vdots\\ *\end{matrix}}\right\} h_2 \\ \\ \left.\vphantom{\begin{matrix}*\\\vdots\\ *\end{matrix}}\right\} h_n \end{matrix} \tag{35}$$

$$\mathbf{W}_l = \begin{pmatrix} * & \cdots & * & * & \cdots & * & & * & \cdots & * \\ 0 & \cdots & 0 & * & \cdots & * & & * & \cdots & * \\ & & & & & & \ddots & & & \\ 0 & \cdots & 0 & 0 & \cdots & 0 & & * & \cdots & * \end{pmatrix} \tag{36}$$
$$\underbrace{\hphantom{* \cdots *}}_{h_1} \underbrace{\hphantom{* \cdots *}}_{h_2} \quad \underbrace{\hphantom{* \cdots *}}_{h_n}$$

$$\mathbf{W}_i = \begin{pmatrix} * & \cdots & * & * & \cdots & * & & * & \cdots & * \\ \vdots & \ddots & \vdots & \vdots & \ddots & \vdots & & \vdots & \ddots & \vdots \\ * & \cdots & * & * & \cdots & * & & * & \cdots & * \\ 0 & \cdots & 0 & * & \cdots & * & & * & \cdots & * \\ \vdots & \ddots & \vdots & \vdots & \ddots & \vdots & & \vdots & \ddots & \vdots \\ 0 & \cdots & 0 & * & \cdots & * & & * & \cdots & * \\ & & & & & & \ddots & & & \\ 0 & \cdots & 0 & 0 & \cdots & 0 & & * & \cdots & * \\ \vdots & \ddots & \vdots & \vdots & \ddots & \vdots & & \vdots & \ddots & \vdots \\ 0 & \cdots & 0 & 0 & \cdots & 0 & & * & \cdots & * \end{pmatrix} \begin{matrix} \left.\vphantom{\begin{matrix}*\\\vdots\\ *\end{matrix}}\right\} h_1 \\ \\ \left.\vphantom{\begin{matrix}*\\\vdots\\ *\end{matrix}}\right\} h_2 \\ \\ \left.\vphantom{\begin{matrix}*\\\vdots\\ *\end{matrix}}\right\} h_n \end{matrix} \quad \text{for } i \in \{2, ..., l-1\} \tag{37}$$
$$\underbrace{\hphantom{* \cdots *}}_{h_1} \underbrace{\hphantom{* \cdots *}}_{h_2} \quad \underbrace{\hphantom{* \cdots *}}_{h_n}$$

Here, $h_i$ is the number of hidden units dedicated to transforming $\mathbf{z}_i$ with the constraint $\sum_{i=1}^n h_i = h$. We perform an even split such that the $h_i$ and $h_j$ differ by at most 1 for $i, j \in \{1, ..., n\}$. The weight matrices are restricted to be block triangular during optimization by setting the respective matrix elements to zero after each iteration of the optimizer. The model can simply be made and kept invertible using the same spectral normalization as is used for dense residual flows [16]. We train our model to map onto a standard Normal base distribution.

### E.3 Generating random MLP mixing functions

In order to generate random MLP mixing functions, we adopt the same initalisation as in [29]: we initialise the square weight matrices to be orthogonal,[22] and use the `leaky_tanh` invertible nonlinearity.

---

[22]Note that orthogonality of the weight matrices in a MLP does not guarantee satisfying Principle 4.1, due to the element-wise nonlinearities between the layers, which overall lead to a Jacobian whose columns are in general not orthogonal.

### E.4 Maximum likelihood with low $C_{\text{IMA}}$

The modified maximum likelihood objective described in § 5.2 can be written as follows:[23]

$$
\begin{aligned}
\mathcal{L}(\mathbf{g}; \mathbf{x}) &= \log p(\mathbf{x}) - \lambda \cdot c_{\text{IMA}}(\mathbf{g}^{-1}, p_{\mathbf{y}}) \\
&= \sum_{i=1}^{n} \log p_{y_i}(\mathbf{g}^i(\mathbf{x})) + \log |\mathbf{J}_{\mathbf{g}}(\mathbf{x})| - \lambda \cdot \left( \sum_{i=1}^{n} \log \left\| [\mathbf{J}_{\mathbf{g}^{-1}}(\mathbf{g}(\mathbf{x}))]_i \right\| - \log |\mathbf{J}_{\mathbf{g}^{-1}}(\mathbf{g}(\mathbf{x}))| \right) \\
&= \sum_{i=1}^{n} \log p_{y_i}(\mathbf{g}^i(\mathbf{x})) + \log |\mathbf{J}_{\mathbf{g}}(\mathbf{x})| - \lambda \cdot \left( \sum_{i=1}^{n} \log \left\| [\mathbf{J}_{\mathbf{g}}^{-1}(\mathbf{x})]_i \right\| + \log |\mathbf{J}_{\mathbf{g}}(\mathbf{x})| \right) \\
&= \sum_{i=1}^{n} \log p_{y_i}(\mathbf{g}^i(\mathbf{x})) + (1 - \lambda) \log |\mathbf{J}_{\mathbf{g}}(\mathbf{x})| - \lambda \sum_{i} \log \left\| [\mathbf{J}_{\mathbf{g}}^{-1}(\mathbf{x})]_i \right\| ,
\end{aligned}
\tag{38}
$$

where $[\mathbf{J}_{\mathbf{g}}^{-1}(\mathbf{x})]_i$ represents the $i$-th column of the inverse of the Jacobian of $\mathbf{g}$ computed at $\mathbf{x}$.

We use the same model as the one described in Appendix E.2, but without the constraint that the Jacobian should be triangular, and train with a Logistic base distribution.

Note that the computational efficiency of optimising objective (38) is cubic in the input size $n$, due to a number of operations (matrix inversion, Jacobian and determinant computation via automatic differentiation, etc.) which are $\mathcal{O}(n^3)$. However, similarly to what already observed in [30], we found that for data of moderate dimensionality computing and optimising objective (38) with automatic differentiation is feasible. For example, training a residual flow with 64 layers for $10^5$ iterations takes roughly 5.3 hours for $n = 2$, 5.7 hours for $n = 5$, and 6.3 hours for $n = 7$ on the same hardware (see section E.5). An interesting direction for future work would be to find computationally efficient ways of optimising (38).

When computing the $C_{\text{IMA}}$ of the Darmois solutions of randomly generated functions, we restricted ourselves to Möbius transformations, i.e. conformal maps. However, there are also nonconformal maps satisfying $C_{\text{IMA}} = 0$, e.g. the transformation of Cartesian to Polar coordinates, see Appendix D. To test whether the $C_{\text{IMA}}$ of the Darmois solutions is actually bigger than 0, we gener

### E.5 Evaluation

**Mean correlation coefficient.** To evaluate the performance of our method, we compute the mean correlation coefficient (MCC) between the original sources and the corresponding latents, see for example [49]. We first compute the matrix of correlation coefficients between all pairs of ground truth and reconstructed sources. Then, we solve a linear sum assignment problem (e.g. using the Hungarian algorithm) to match each reconstructed source to the ground truth one which has the highest correlation with it. The MCC matrix contains the Spearman rank-order correlations between the ground truth and reconstructed sources, a measure which is blind to nonlinear invertible reparametrisations of the sources.

**Nonlinear Amari metric.** While the MCC metric evaluates BSS by comparing ground truth and reconstructed sources, we propose an additional evaluation directly based on comparing the (Jacobians of the) true mixing and the learned unmixing. We take inspiration from an evaluation metric used in the context of linear ICA, the Amari distance [5]: Given a learned unmixing $\mathbf{W}$ and the true mixing $\mathbf{A}$, and defining the matrix $\mathbf{R} = \mathbf{A}\mathbf{W}$, the Amari distance is defined as

$$
d^{\text{Amari}}(\mathbf{R}) = \sum_{i=1}^{n} \left( \sum_{j=1}^{n} \frac{[\mathbf{R}]_{ij}^2}{\max_l [\mathbf{R}]_{il}^2} - 1 \right) + \sum_{i=1}^{n} \left( \sum_{j=1}^{n} \frac{[\mathbf{R}]_{ji}^2}{\max_l [\mathbf{R}]_{lj}^2} - 1 \right) ,
\tag{39}
$$

and is greater than or equal to zero, canceling if and only if $\mathbf{R}$ is a scale and permutation matrix, that is when the learned unmixing is matching the unresolvable ambiguities of linear ICA.

We extend this idea to the nonlinear setting: Given a true mixing $\mathbf{f}$ and a learned unmixing $\mathbf{g}$, we define our nonlinear Amari distance as

$$
d^{\text{n-Amari}}(\mathbf{g}, \mathbf{f}) = \mathbb{E}_{\mathbf{x} \sim p_{\mathbf{x}}} \left[ d^{\text{Amari}} \left( \mathbf{J}_{\mathbf{g}}(\mathbf{x}) \mathbf{J}_{\mathbf{f}}(\mathbf{f}^{-1}(\mathbf{x})) \right) \right] .
\tag{40}
$$

---

[23]while the objective in § 5.2 involves an expectation over $p_{\mathbf{x}}$, we consider the loss for a single point $\mathbf{x}$ here, $\mathcal{L}(\mathbf{g}; \mathbf{x})$.

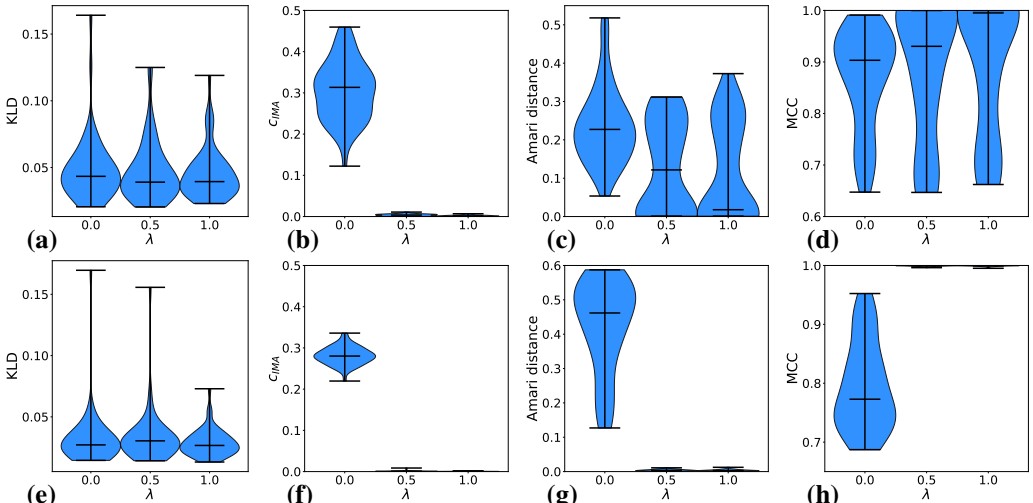

Figure 8: BSS via $C_{\text{IMA}}$-regularised MLE for $n = 2$ dimensions with $\lambda \in \{0.0, 0.5, 1.0\}$. The true mixing function is a randomly generated Möbius transformation, nonlinear (with $\epsilon = 2$) in (a)–(d) and linear (with $\epsilon = 0$) transformation for (e)–(h). For each type of transformation and $\lambda$, seeded runs are done. (a), (e) KL-divergence between ground truth likelihood and learnt model; (b), (f) $C_{\text{IMA}}$ of the learnt models; (c), (g) nonlinear Amari distance given true mixing and learnt unmixing; (d), (h) MCC between true and reconstructed sources.

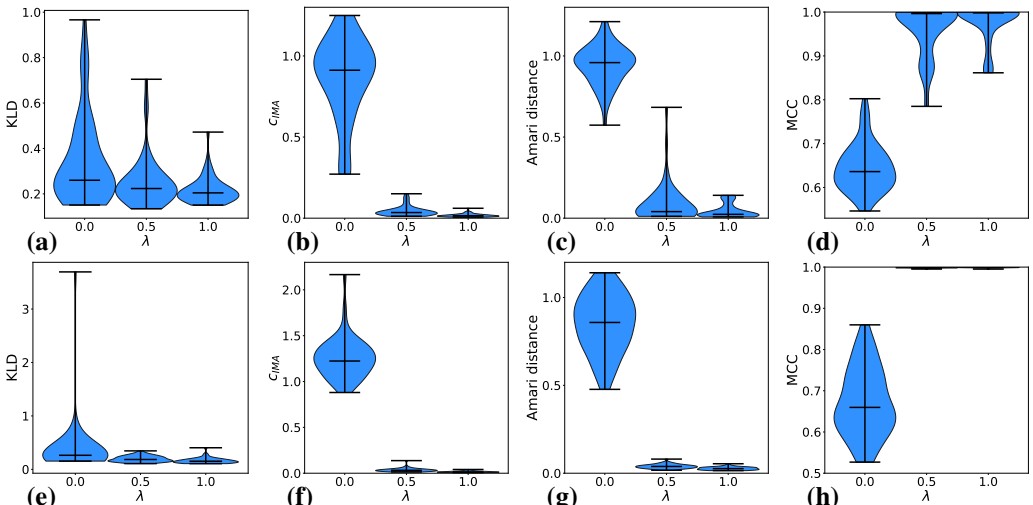

Figure 9: BSS via $C_{\text{IMA}}$-regularised MLE for $n = 5$ dimensions with $\lambda \in \{0.0, 0.5, 1.0\}$. The true mixing function is a randomly generated Möbius transformation, nonlinear (with $\epsilon = 2$) in (a)–(d) and linear (with $\epsilon = 0$) transformation for (e)–(h). For each type of transformation and $\lambda$, seeded runs are done. (a), (e) KL-divergence between ground truth likelihood and learnt model; (b), (f) $C_{\text{IMA}}$ of the learnt models; (c), (g) nonlinear Amari distance given true mixing and learnt unmixing; (d), (h) MCC between true and reconstructed sources.

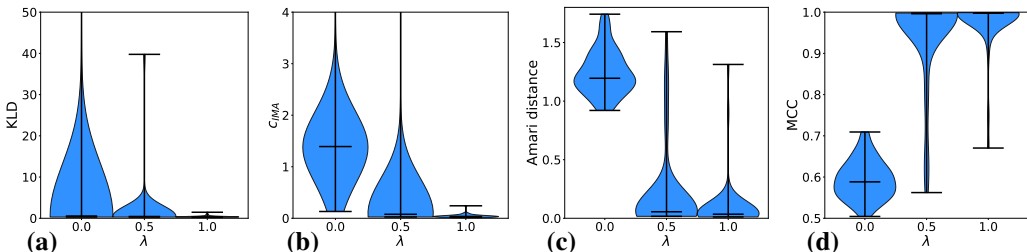

Figure 10: BSS via $C_{\text{IMA}}$-regularised MLE for $n = 7$ dimensions with $\lambda \in \{0.0, 0.5, 1.0\}$. The true mixing function is a randomly generated Möbius transformation (with $\epsilon = 2$). For each $\lambda$, seeded runs are done. **(a)** KL-divergence between ground truth likelihood and learnt model; **(b)** $C_{\text{IMA}}$ of the learnt models; **(c)** nonlinear Amari distance given true mixing and learnt unmixing; **(d)** MCC between true and reconstructed sources.

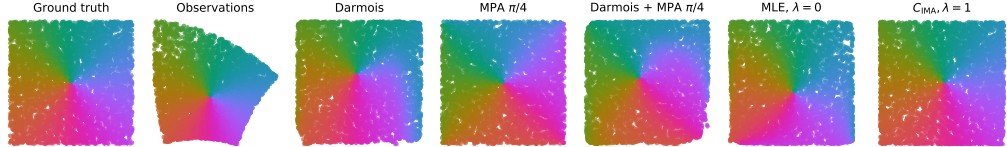

Figure 11: Visual comparison of different nonlinear ICA solutions for $n = 2$: *(left to right)* true sources; observed mixtures; Darmois solution; true unmixing, composed with the measure preserving automorphism (MPA) from (5) (with rotation by $\pi/4$); Darmois solution composed with the same MPA; maximum likelihood ($\lambda = 0$); and $C_{\text{IMA}}$-regularised approach ($\lambda = 1$).

Then, according to the definition of Amari distance (39), if the smooth function $\mathbf{g} \circ \mathbf{f}$ is a permutation composed with a scalar function, thus precisely matching the BSS equivalence class defined in Defn. 2.2, this would result in its Jacobian (that is, the product of the Jacobians $\mathbf{J_g}(\mathbf{x})\mathbf{J_f}(\mathbf{f}^{-1}(\mathbf{x}))$) equalling the product of a diagonal matrix and a permutation matrix at every point $\mathbf{x}$: the quantity $d^{\text{m-Amari}}(\mathbf{g}, \mathbf{f})$ would therefore be equal to zero.

This metric can be of independent interest and potentially useful in contexts where the reconstructed sources might be a noisy version of the true ones, but the true unmixing is nevertheless identifiable. Our implementation is based on the one for the (linear) Amari distance provided in the code for [1].

$C_{\text{IMA}}$ **of Darmois solutions for nonconformal maps satisfying the IMA principle.** When computing the $C_{\text{IMA}}$ of the Darmois solutions of randomly generated functions, we restricted ourselves to Möbius transformations which are conformal maps. However, there are also nonconformal maps satisfying $C_{\text{IMA}} = 0$, e.g., the transformation from polar to Cartesian coordinates with $n = 2$, see Appendix D. To test whether the $C_{\text{IMA}}$ of the Darmois solutions is actually bigger than 0, we generate random radial transformations by imposing a random scale and shift before applying the radial transformation, compute the Darmois solution as we have done in § 5.1, and calculate its $C_{\text{IMA}}$ on the test set. We did 50 runs and the results are shown in Fig. 12.

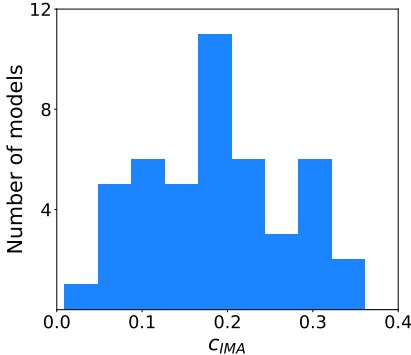

Figure 12: Histogram of the $C_{\text{IMA}}$ values of the Darmois solutions of 50 randomly generated radial transformations.

Similar to Fig. 4 **(a)** we can clearly see that all $C_{\text{IMA}}$ values of the final models are larger than 0, with the smallest value being 0.01. This confirms the result we have already shown theoretically.

**Additional plots for § 5.2.** We show additional plots for the quantitative experiments involving training with the objective described in (38), see Fig. 8, Fig. 9 and Fig. 10.

For $\epsilon = 0$ (that is, ground truth mixing linear), there appears to be an almost perfect recovery of the ground truth sources (resp. unmixing function) for $\lambda \in \{0.5, 1.0\}$, as can be seen by the high (resp. low) values of the MCC (resp. nonlinear Amari distance) evaluations ; this is in stark contrast with the distribution of the MCC (resp. nonlinear Amari distance) values for models trained with $\lambda = 0$, which are typically much higher (resp. lower), indicating that the learned solutions do not achieve blind source separation (see $n = 2$, Fig. 8 **(g), (h)**; $n = 5$, Fig. 9 **(g), (h)**). All models achieve a comparably good fit, reflected in the KL-divergence values ($n = 2$, Fig. 8 **(e)**; $n = 5$, Fig. 9 **(e)**).

The trend is confirmed when the true mixing is nonlinear ($\epsilon = 2$), with slightly lower (resp. higher) values achieved with $C_{\text{IMA}}$ regularisation for the MCC (resp. nonlinear Amari) metrics; this possibly due to the increased difficulty of fitting observations generated by a nonlinear mixing, as can be seen from the higher values of the KL-divergence ($n = 2$, Fig. 8 **(a)**; $n = 5$, Fig. 9 **(a)**; $n = 7$, Fig. 10 **(a)**);[24] still, the beneficial effect of $\lambda \in \{0.5, 1.0\}$ with respect to models trained with $\lambda = 0$ is clear, and is apparently stronger for $\lambda = 1.0$ and with higher data dimensionality $n$ ($n = 2$, Fig. 8 **(c), (d)**; $n = 5$, Fig. 9 **(c), (d)**; $n = 7$, Fig. 10 **(c), (d)**).

We additionally plot the $C_{\text{IMA}}$ values for the all trained models, for all values of $\lambda$. It can be seen that solutions found by unregularised maximum likelihood estimation typically learn functions with relatively high values of $C_{\text{IMA}}$, while as expected the regularised version achieves low values ($n = 2$, Fig. 8 **(b), (f)**; $n = 5$, Fig. 9 **(b), (f)**; $n = 7$, Fig. 10 **(b)**).

Finally, in figure 11, we report the same plot as in 4, top row, but with a perceptually uniform colormap.

**Comparison to FastICA.** We compared the performance of our proposed regularised maximum likelihood procedure to a state of the art method for linear ICA, FastICA [36], in the implementation from the Scikit-learn package [73], over 50 repetitions. Our experiments show that our regularised method ($\lambda = 0.5$, and particularly $\lambda = 1.0$; $\lambda = 0.0$ provides the unregularised nonlinear baseline) is superior in learning the true unmixing and reconstructing the sources. This indicates that the linearity assumption of FastICA does not allow enough flexibility to solve blind source separation in our setting, whereas our criterion does (see Fig. 13, Fig. 14 and Fig. 15).[25] While the spread in the distributions of MCC and Amari distance can be largely attributed to the brittleness of neural networks, the median values for the MCC (resp. nonlinear Amari distance) are consistently higher (resp. lower) for our regularised method than for FastICA. In contrast, the performance of FastICA is consistently better than the unregularised baseline.

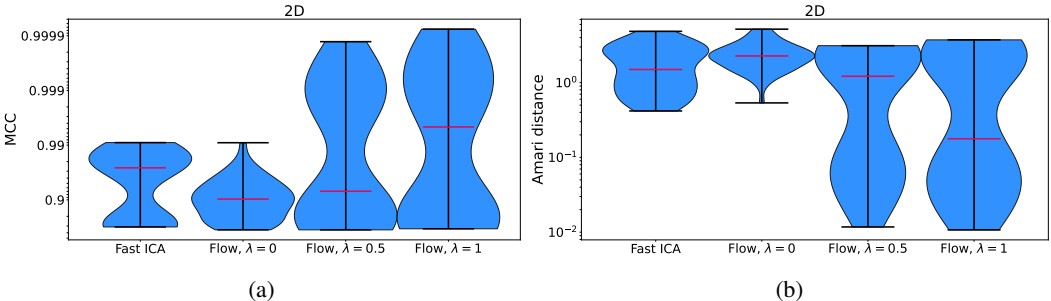

(a)                                                       (b)

Figure 13: Comparison between FastICA and our normalising flow method with $\lambda \in \{0.0, 0.5, 1.0\}$, $n = 2$. (a) MCC; (b) Amari distance.

---

[24]The distribution of the KL values contains outliers, and seemingly more strongly for lower values of $\lambda$.

[25]the experimental setting and the plots for the normalising flow models correspond to those already shown in the paper, but here we modified the $y$-axis scale to facilitate the comparison of all methods

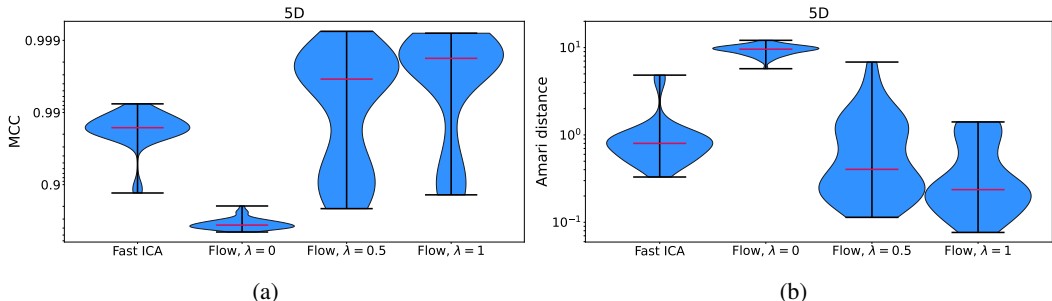

Figure 14: Comparison between FastICA and our normalising flow method with $\lambda \in \{0.0, 0.5, 1.0\}$, $n = 5$. (a) MCC; (b) Amari distance.

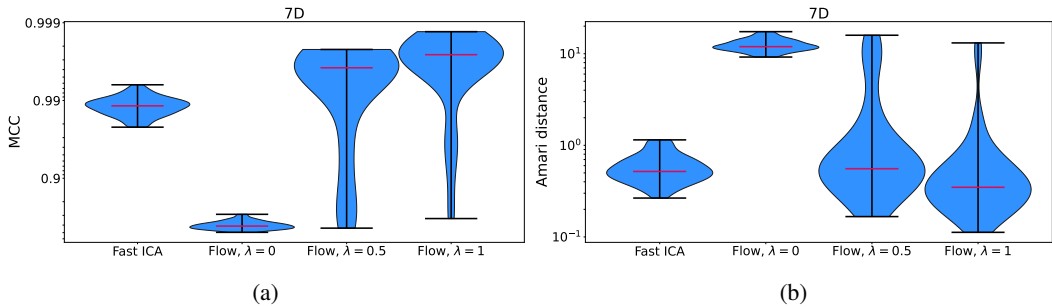

Figure 15: Comparison between FastICA and our normalising flow method with $\lambda \in \{0.0, 0.5, 1.0\}$, $n = 7$. (a) MCC; (b) Amari distance.

**Details on resources used.** All models were trained on compute instances with 16 Intel Xeon E5-2698 CPUs and a Nvidia Geforce GTX980 GPU. The cluster we used has 204 thereof. Training the models took between 4 and 16 hours depending mainly on the dimensionality $n$ and number of samples in the dataset, and on the number of iterations used for training. Overall, we trained around 2000 models, amounting to roughly 18000 GPU hours.

# F   Additional background on conformal maps and Möbius transformations

**Similarities.**   A *similarity* of a Euclidean space is a bijection $\mathbf{f}$ from the space onto itself that multiplies all distances by the same positive real number $r$, so that for any two points $\mathbf{x}$ and $\mathbf{y}$ we have

$$d(\mathbf{f}(\mathbf{x}), \mathbf{f}(\mathbf{y})) = rd(\mathbf{x}, \mathbf{y}),$$

where $d(\mathbf{x}, \mathbf{y})$ is the Euclidean distance from $\mathbf{x}$ to $\mathbf{y}$ [94]. The scalar $r$ is sometimes termed the ratio of similarity, the stretching factor and the similarity coefficient. When $r = 1$ a similarity is called an isometry (rigid transformation). Two sets are called similar if one is the image of the other under a similarity.

As a map $\mathbf{f} : \mathbb{R}^n \to \mathbb{R}^n$, a similarity of ratio $r$ takes the form

$$\mathbf{f}(\mathbf{x}) = r\mathbf{A}\mathbf{x} + \mathbf{t},$$

where $\mathbf{A}$ is a orthogonal matrixn $n \times n$ and $\mathbf{t} \in \mathbb{R}^n$ is a translation vector.

Note that such a similarity $\mathbf{f}$ has Jacobian $\mathbf{J_f}(\mathbf{x}) = r\mathbf{A}$ for any $\mathbf{x}$.

**Conformal maps.**   Conformal maps are angle preserving transformation, and in this sense, are a generalization of similarities. In short, let $U$ be an open subset of $\mathbb{R}^n$, $\varphi : U \to \mathbb{R}^n$ is a conformal map if, for two arbitrary curves $\gamma_1(t)$ and $\gamma_2(t)$ on $\mathbb{R}^n$, where these curves intersect each other with angle $\theta$ in point $\mathbf{p} \in U$, then $\varphi \circ \gamma_1(t)$ and $\varphi \circ \gamma_2(t)$ intersect each other with the same angle $\theta$ in the point $\varphi(\mathbf{p})$.

A characterisation of conformal maps directly related to orthogonal coordinate systems is the following.

**Proposition F.1** (See e.g. [95])**.** *Let $U$ be an open subset of $\mathbb{R}^n$ with a $C^1$-function $\varphi : U \to \mathbb{R}^n$. Then $\varphi$ is conformal iff there exists a scalar function $\lambda : U \to \mathbb{R}$ such that $\lambda(\mathbf{x})^{-1}\mathbf{J}_\varphi(\mathbf{x})$ is an orthogonal matrix for all $\mathbf{x}$ in $U$. We call $\lambda$ the scale factor of $\varphi$.*

While it can be shown that *linear* conformal maps are similarities, an interesting class of *nonlinear* conformal maps are the unit radius sphere inversion (restriction to unit radius is only to avoid unnecessary notational complexity):

$$I_{\mathbf{b}} : \mathbb{R}^n \setminus \{0\} \to \mathbb{R}^n \setminus \{0\}$$

$$\mathbf{x} \mapsto \frac{\mathbf{x} - \mathbf{b}}{\|\mathbf{x} - \mathbf{b}\|^2} + \mathbf{b}$$

We can notice that such transformation leaves the hypersphere of center $\mathbf{b}$ and radius 1 invariant, while the points outside of the unit ball are mapped to the interior of the unit ball, and vice-versa.

Interestingly, conformal maps in Euclidean spaces of dimension superior or equal to 3 can be restricted to two kinds according to the following result from Liouville.

**Theorem F.2** (see e.g. [100])**.** *Let $f : U \to \mathbb{R}^n$ be a conformal map defined on a connected open subset of Euclidean space $\mathbb{R}^n$ of dimension $n \geq 3$. Then $f = L_{|U}$ can be written either as the restriction of a similarity $L$ to $U$, or as the composition $f = I \circ L_{|U}$ of such a map with an inversion with respect to a hypersphere of unit radius, centered at the origin.*

The class of function described in Thm. F.2 corresponds exactly to the Möbius transformations described in (32). These transformation can as well be defined in dimension 2, with the specificity that they are only a subset of the class conformal maps in this dimension.

**Properties of sphere inversion.**   We characterize the properties of the unit sphere centered at zero, that we denote $I$

$$I : \mathbb{R}^n \setminus \{0\} \to \mathbb{R}^n \setminus \{0\}$$

$$\mathbf{x} \mapsto \frac{\mathbf{x}}{\|\mathbf{x}\|^2}$$

Now let us derive the Jacobian of $I$. A straightforward computation leads to

$$\mathbf{J}_I(\mathbf{x}) = \frac{1}{\|\mathbf{x}\|^2} \left( \mathbf{I}_n - 2\frac{\mathbf{x}\mathbf{x}^\top}{\|\mathbf{x}\|^2} \right)$$

where $\mathbf{I}_n$ denote the identity matrix.

By noticing that $\frac{\mathbf{x}\mathbf{x}^\top}{\|\mathbf{x}\|^2}$ is rank one symmetric with eigenvalue 1 associated with unit norm eigenvector $\frac{\mathbf{x}}{\|\mathbf{x}\|}$, we can diagonalize this matrix in any (space dependent) orthogonal basis that has $\frac{\mathbf{x}}{\|\mathbf{x}\|}$ as the first basis vector.

Let us thus pick the unit vectors associated to the hyperspherical coordinates (which satisfy this condition by definition), and consider the orthogonal matrix $\mathbf{B}(\frac{\mathbf{x}}{\|\mathbf{x}\|})$ gathering these basis vectors as its columns (it is parameterized by the unit vector $\frac{\mathbf{x}}{\|\mathbf{x}\|}$, as this basis is radially invariant. Then we can write

$$\frac{\mathbf{x}\mathbf{x}^\top}{\|\mathbf{x}\|^2} = \mathbf{B}\left(\frac{\mathbf{x}}{\|\mathbf{x}\|}\right)\mathbf{D}\mathbf{B}\left(\frac{\mathbf{x}}{\|\mathbf{x}\|}\right)^\top$$

and thus

$$\mathbf{J}_I(\mathbf{x}) = \frac{1}{\|\mathbf{x}\|^2}\left(\mathbf{I}_n - 2\mathbf{B}\left(\frac{\mathbf{x}}{\|\mathbf{x}\|}\right)\mathbf{D}\mathbf{B}\left(\frac{\mathbf{x}}{\|\mathbf{x}\|}\right)^\top\right) = \frac{1}{\|\mathbf{x}\|^2}\mathbf{B}\left(\frac{\mathbf{x}}{\|\mathbf{x}\|}\right)(\mathbf{I}_n - 2\mathbf{D})\mathbf{B}\left(\frac{\mathbf{x}}{\|\mathbf{x}\|}\right)^\top$$

with $\mathbf{D}$ a diagonal matrix with diagonal elements $[1, 0, \ldots, 0]$. This leads to

$$\mathbf{J}_I(\mathbf{x}) = \frac{1}{\|\mathbf{x}\|^2}\mathbf{B}\left(\frac{\mathbf{x}}{\|\mathbf{x}\|}\right)\mathbf{D}_I\mathbf{B}\left(\frac{\mathbf{x}}{\|\mathbf{x}\|}\right)^\top$$

with $\mathbf{D}_I = \mathbf{I}_n - 2\mathbf{D}$ a diagonal matrix with diagonal elements $[-1, 1, \ldots, 1]$. The Jacobian thus takes the form predicted by the above proposition for conformal maps

$$\mathbf{J}_I(\mathbf{x}) = \lambda(\mathbf{x})\mathbf{O}\left(\frac{\mathbf{x}}{\|\mathbf{x}\|}\right)$$

with scale factor $\lambda(\mathbf{x}) = \frac{1}{\|\mathbf{x}\|^2}$ and $\mathbf{O}(\frac{\mathbf{x}}{\|\mathbf{x}\|}) = \mathbf{B}\left(\frac{\mathbf{x}}{\|\mathbf{x}\|}\right)\mathbf{D}_I\mathbf{B}\left(\frac{\mathbf{x}}{\|\mathbf{x}\|}\right)^\top$ a space dependent orthogonal matrix, which has the additional property to be radially invariant for the specific case of sphere inversions.