# OpenReview forum: "Independent mechanism analysis, a new concept?"
_NeurIPS.cc/2021/Conference — NeurIPS 2021 Poster_

### Official Review · Reviewer_989y · 2021-07-14

**Rating:** 7
**Confidence:** 3

**Summary:**

The paper aims at the identifiability of nonlinear ICA. It proposes a principle as a constraint of mixture functions based on the modified independent causal mechanism principle in the causal discovery literature; investigates the identifiability of nonlinear ICA under such an assumption; then applies it to nonlinear ICA. It also shows that the identifiability results can cover some non-identifiable cases in the existed literature.

**Limitations And Societal Impact:**

Yes.

**Main Review:**

1. The idea of the work is interesting: it adjusted the ICM principle in the related field of causal discovery for nonlinear ICA.
2. The paper is in general well written and properly introduces the method. The theoretical analysis and examples are appreciated.

My minor concern is how restrictive is the proposed IMA in line 199 which imposes constraints on the determinant of the Jacobin matrix. Although the information-theoretic interpretation and geometric interpretation were trying to further explain IMA, I didn't get help from the given information without further details and explanation.

**Time Spent Reviewing:**

2

---

> ### Author Response · Authors · 2021-08-10
> **Response to reviewer 989y**
>
> We thank you for your positive evaluation of our work, and we will try to provide answers to the concerns you have raised.
>
> * **How restrictive is the proposed IMA principle?** This should be judged relative to other settings seeking identifiability. We stress again that our results concern the fully unsupervised setting of blind source separation, where no auxiliary variable is available. Whenever auxiliary variables are available, and under suitable technical assumptions, identifiability results hold even when the mixing is allowed to be a generic, albeit smooth and invertible, function [34, 35, 38, 44, 27]. There is, however, a tradeoff between this flexibility and the strong assumptions that (i) an additional auxiliary variable, typically termed $\mathbf{u}$, is observed; and (ii) stringent technical assumptions need to hold for the conditional probability of the sources given $\mathbf{u}$, or the observations are required to follow a specific time-structure [28].
> In contrast, in the fully unsupervised setting we consider, nonlinear blind source separation represents a challenging and long standing problem, expected to require stronger conditions for the mixing. We remark that **many of the state of the art methods for identifiable, unsupervised independent component analysis either rely on a linearity assumption [33, 36, 16] or on small deviations from it, such as the minimal nonlinear distortion principle [93]**. As a test for the restrictiveness of our assumption, we have shown in Section 5.1 that even in cases where the IMA principle is not exactly satisfied (e.g., the case where the mixing is given by a random MLP, a standard benchmark in nonlinear ICA literature [34, 35, 38, 44]) our proposed $C_{\operatorname{IMA}}$ criterion can provide useful learning signal in distinguishing some “spurious” solutions from the “good” ones achieving blind source separation.
> Moreover, we ran some additional experiments, comparing our proposed regularised maximum likelihood procedure to a state of the art method for linear ICA (FastICA) over $50$ repetitions. Our experiments show that our regularised method ($\lambda=0.5$, and particularly $\lambda=1$; $\lambda=0$ provides the unregularised nonlinear baseline) is superior in learning the true unmixing and reconstructing the sources. This indicates that the linearity assumption does not allow enough flexibility to solve blind source separation in our setting, whereas our criterion does (see plots at the following link for the MCC evaluation: https://ibb.co/YtVBrWf, and this link for the nonlinear Amari distance: https://ibb.co/6bQK8QY). While the spread in the distributions of MCC and Amari distance can be largely attributed to the brittleness of neural networks, the median values for the MCC (resp. nonlinear Amari distance) are consistently higher (resp. lower).
>
> * **The information-theoretic and geometric interpretation did not provide help.** We will try to provide more elaboration, both in technical and in more intuitive terms, as detailed below.
>   * **Information geometric interpretation.** We see our IMA principle as closely related to IGCI formulation in the context of cause-effect pair inference --- where the dependence between cause and mechanism is viewed as a fine tuning of the derivative of the mechanism to the input density, as illustrated in Figure. 2a.  In our considered problem setting---that is, ICA and blind source separation---the IMA principle defines a complementary, non-statistical measure of independence between the influences of the individual sources on the vector of observations, which can be expressed as partial derivatives $\partial \mathbf{f}/\partial s_i$. This connection is described in technical terms in appendix B.3. In short, both the IGCI and IMA postulates have an information geometric interpretation related to the influence of “non-statistically” independent factors on the observations. While for IGCI independent factors are the cause and mechanism, for IMA these are the influences of each source component on the observations. Specifically, the effect of component-wise soft interventions on the sources, as measured by the KL divergence between the original and the intervened distributions, decomposes into the sum of the individual KL-divergences, see equation (21).
>   * **Intuitive explanation.** (see also our response to `ovUQ`) As a more intuitive explanation of our principle, it can be useful to provide a pictorial example of a cocktail party-like situation violating the IMA principle. A cocktail party may violate our IMA principle when the speakers' positions and the room acoustics have been fine tuned to one another. This is for example the case in a concert halls where the acoustics of the room has been fine-tuned to the position and configuration of multiple locations on the stage, where the sources (i.e. the voices of the actors or singers) are emitted---in order to make the listening experience as homogeneous as possible across the spectators (that is, the influence of each of the sources on the different listeners should not differ too much). This would lead to an increase in collinearity between the columns of the mixing’s Jacobian, thus violating the IMA principle.

---

> > ### Comment · Reviewer_989y · 2021-08-31
> > **Response**
> >
> > Thank you for the further explanation about the restrictiveness of the assumption and the interpretation.

---

### Official Review · Reviewer_hbzz · 2021-07-15

**Rating:** 8
**Confidence:** 5

**Summary:**

The authors propose to borrow concepts from causality, in particular "independent causal mechanisms" , to propose a novel framework termed "independent mechanism analysis" (IMA), which can provide non-spurious solutions to nonlinear blind source separation. IMA can be seen as a restricted nonlinear ICA model, where the contributions of the latent variables $z_i$ to the nonlinear mixing $\mathbf{f}$ are independent, i.e. the columns of the Jacobian $J_\mathbf{f}$ are orthogonal.


**Limitations And Societal Impact:**

The limitations of the paper are well explained and discussed in section 6.

**Main Review:**

The proposed IMA framework is novel, and provides an alternative solution to the nonlinear ICA (equivalently nonlinear BSS) problem. It is by now known that nonlinear ICA is not possible when dealing with general nonlinear mixing functions by only assuming independence of the latent components. Recent approaches usually involve using inductive biases in the form of auxiliary information to condition the latent distribution, without constraining the mixing. The authors propose to take a different path, by constraining the mixing instead.
The paper is very well written, and was a joy to read. Section 2 does a very good job of laying the ground for the IMA framework. Readers without knowledge about nonlinear ICA or the ICM framework will find this section very helpful.
The theoretical results are sound, and the assumptions are well justified.
The new IMA contrast functions is very useful, and can be used in practice to rule out a lot of spurious solutions to the nonlinear ICA problem.

Overall, this work is original, and of great significance. Learning identifiable representations is an important topic in unsupervised learning, and the provided framework is an important step in this direction.
I don't have much to criticize about the paper. I would have liked to see a full identifiability theory, but I believe it will follow in future work, and I am satisfied with the quality and quantity of the current contribution.

**Time Spent Reviewing:**

10 hours

---

> ### Author Response · Authors · 2021-08-10
> **Response to reviewer hbzz**
>
> We warmly thank you for your words of praise and appreciation of our work, as well as  for the amount of time dedicated to reviewing it. We concur that a full identifiability proof for this unsupervised setting would be an extremely important result: it is arguably a difficult task, where, to the best of our knowledge, very few positive results have been attained in the last twenty years. We hope that our work can provide a stepping stone in this direction.

---

### Official Review · Reviewer_tDZV · 2021-07-16

**Rating:** 6
**Confidence:** 4

**Summary:**

This work proposes a new regularization scheme to improve identifiability in the _nonlinear blind source separation problem_ (nonlinear ICA) inspired by the notion of _independence of mechanism_ coming from causal inference. The idea is to consider only mixing functions which have a Jacobian matrix with orthogonal columns everywhere. They show how this extra assumption allows to exclude two classical types of degeneracies in nonlinear ICA, although without showing identifiability. The theoretical claims are validated by synthetic experiments.


**Limitations And Societal Impact:**

Yes.

**Main Review:**

Originality:

Recent works on nonlinear ICA have focused on imposing structure on the distribution of the source via auxiliary variables. Instead, this work proposes to put restrictions on the mixing function f. Although different assumptions on f have been explored in the past (cited by the authors), their independence of mechanism assumption (IMA) hasn’t (to the best of my knowledge).

Quality:

I have not reviewed the proofs, however the various theoretical claims seem reasonable and are clearly presented. I was happy to read about new findings in that interesting direction. However, I have some concerns with the overall framing of the paper.

The inspiration for the authors' approach is the principle of independent causal mechanism (ICM). However, it is not clear to me what is the link between ICM and the orthogonality of the columns of the Jacobian of the mixing function. The authors seem to suggest a connection, but it felt somewhat vague. ICM is about algorithmic independence of mechanisms in a structural causal model (SCM), and here we regularize for orthogonality of the columns of the Jacobian of the mixing function.

In causality, the term “mechanism” is typically reserved to the conditional distribution of a variable given its direct causal parents (or its associated structural equation). In this work, “mechanism” is used to denote a partial derivative of f w.r.t. some source s_i. This is an unnecessary and confusing clash of terminology in my opinion. Consequently, I do not think independent mechanism analysis is a good name for this contribution.

The authors also seem to suggest a connection between the orthogonality of partial derivatives and their algorithmic independence (lines 192-200), but again, this is a bit vague and unjustified (or maybe I missed an important point).

The above points can be summarized as follows: I am not convinced that ICM and algorithmic independence is the correct language to describe the authors’ contribution, unless more justification is added to the paper.

Section 3 shows how “regularizing” for algorithmic independence between p(s) and f via a proxy (IGCI) is not sufficient for identifiability. It is always interesting to have negative results, but, following up on the above paragraph, the connection to ICM feels a bit forced.

Instead of showing identifiability for the model class they consider, the authors show how it excludes two types of degeneracies, namely the Darmois construction and the “rotated-Gaussian measuring preserving automorphisms”. This is not sufficient to show the model class is identifiable, although it is a good step in that direction. The authors acknowledge this fact at the very end of the paper (in the discussion section), I believe this should’ve been stressed earlier in the paper as well as near the theorem statements, since I believe it is not a typical thing to do in this literature (usually, there’s a theorem showing identifiability).

In corollary 4.11, it would be nice to have an intuition for why we need non-Gaussianity, can’t we just apply corollary 4.9 directly (since A is a conformal map)? What am I missing?
It would be good to have experiments with non-conformal maps satisfying the IMA assumption (I didn’t see any).

Minor:
Would be great to have an intuition for why the Darmois construction yields independent sources.

The presentation of information geometric causal inference is very fast and I could not understand its meaning.

Line 332: Would be good to have a brief description of the MCC and the Amari metrics.

Clarity:

I enjoyed reading the paper overall.

Significance:

I believe this work has average significance, given how strong the conditions are on the mixing function and the fact that we do not know if it holds in practice (even partially) due to lack of real world experiments.

That being said, I believe imposing constraints on the mixing function to improve identifiability is an interesting direction with potential significance.

-------
After reading the author's rebuttal, I decided to raise my score from 5 to 6. See my comment above.




**Time Spent Reviewing:**

5

---

> ### Author Response · Authors · 2021-08-10
> **Response to reviewer tDZV**
>
>
> Thank you for a thoughtful and thorough review of our paper, and for raising various important points. Your main concerns appear to be related to the interpretation of our theoretical framework and how it relates to the ICM principle: as also mentioned in the response to all reviewers, this might have been caused by an overemphasis on the algorithmic complexity formulation of ICM in the main text, particularly in Sections 1 and 4, whereas the IGCI formulation is more technically related to our IMA principle. We detail how we will remedy this below, and answer your concerns one by one.
>
> * **The connection between ICM and the orthogonality of the Jacobian columns presented by the authors is not convincing.**
> The ICM principle (Principle 2.6, l.139-140) in its original form is quite general and not directly linked to Kolmogorov complexity; rather, **it is meant to encode, in non-technical terms, an assumption about the structure and modularity of the physical world, which may underlie many aspects of causal learning and reasoning.** This informal statement can be linked to a number of technical methods and formalisations (most notably in causal discovery), based on specific modelling assumptions. In the context of causal discovery, IGCI and the Trace method [41] are possible formalisations; algorithmic complexity provides another one---and it was also proposed as a unifying framework for ICM ([43] and [Lemeire and Janzing, Replacing Causal Faithfulness with Algorithmic Independence of Conditionals, 2013]). However, drawing a formal link between these formulations is nontrivial. Interestingly, you express this when you state: *“the negative result on IGCI in section 3 is interesting, but the connection to ICM feels a bit forced”*. We note that the connection between IGCI and ICM is widely discussed in the literature and predates our paper---see e.g. [70], Section 4.1.7. Therefore, it is not specific to our work---which, in turn, proposes a formalisation of ICM in a novel context, and whose technical formulation is inspired by IGCI as discussed in Appendix B.2 and B.3.
> Summarising, we concede that we might have put excessive emphasis on the algorithmic complexity interpretation of ICM. This is due to the special status it attained in the literature, as a unifying principle for both causal inference (aforementioned references) and blind source separation [61, 62]. We will de-emphasize this connection, and emphasize the presentation of our **IMA principle as a specific instantiation of the principle of independence of mechanisms in the context of blind source separation, with IGCI as a closest technical formalisation**---although in a different context.
>
>
> * **In causality, the term “mechanism” is typically reserved to the conditional distribution of a variable given its direct causal parents (or its associated structural equation).** We acknowledge and are aware that the term “mechanism” has been used in several works, especially in recent years, to denote a causal Markov kernel $p(X_i|PA_i)$ (or the corresponding structural equation).
> However, the term has also been widely used in the causality literature without referring to a specific mathematical object: for example, we refer to Table 1 in [Mahoney, Review: Beyond Correlational Analysis: Recent Innovations in Theory and Method, Sociological Forum 2001] for a long list of definitions from the literature that warrants not being constrained to one very specific definition. To highlight just two definitions, [Slamon, Scientific Explanation and the Causal Structure of the World. Princeton University 1984] states: *“Causal processes, causal interactions, and causal laws provide the mechanisms by which the world works; to understand why certain things happen, we need to see how they are produced by these mechanisms''*; further, [Tilly, Historical analysis of political processes. Handbook of Sociological Theory 2020] states: *“Mechanisms are events that alter relations among some specified set of elements”*.
> Following this perspective, we argue that **a causal mechanism can more generally denote any process that describes the way in which causes influence their effects**. In this context, the partial derivative $\partial \mathbf{f} / \partial s_i$ reflects a causal mechanism (as stated in our Principle 4.1), as it describes the infinitesimal changes in the observations $\mathbf{x}$, when an infinitesimal perturbation is applied to $s_i$. For a related interpretation of the IMA principle in terms of a decomposition of the effects of soft interventions on the sources, we refer to Appendix B.3, in particular l.885-887.
> We will provide the following clarification next to this Principle in the main text:
>
> > While recent work based on the ICM principle has mostly used the term "mechanism" to refer to causal Markov kernels $p(X_i|PA_i)$ or structural equations, our use of the term is meant in a broader sense, in line with many definitions of the concept in the philosophical literature [Mahoney, 2001]. The partial derivative $\partial \mathbf{f} / \partial s_i$ reflects a causal mechanism (as stated in our Principle 4.1) in the sense that it describes the infinitesimal changes in the observations $\mathbf{x}$, when an infinitesimal perturbation is applied to $s_i$.
>
> * **I do not think independent mechanism analysis is a good name for this contribution.** We respect your opinion: the title reflects our proposed interpretation of the ICM principle in the context of blind source separation, as explained above. We remain open to other suggestions.
> * **It should have been stressed earlier in the paper that the authors don’t provide a full identifiability result.** We note that both reviewers `ovUQ` and `hbzz` explicitly mentioned that this limitation of our work is clearly stated and discussed in the paper: we will, however, follow your suggestion and make an effort to clarify it further and earlier on in the paper.
> * **In corollary 4.11, it would be nice to have an intuition for why we need non-Gaussianity. Also, why can we not apply corollary 4.9?**
> Thanks for raising this point: this misunderstanding will be resolved by explicitly stating that Corollary 4.9 relies on the same assumptions as Theorem 4.7, which include that $x_i {\large \not⫫} x_j$ for some $i \neq j$. This assumption is instead not required for Corollary 4.11: in fact, it follows from the assumed linear ICA model with non-Gaussian sources, and the fact that the mixing matrix $\mathbf{A}$ is not the product of a diagonal and a permutation matrix, due to Theorem 11 of [16] (see also Theorem A.1 in the Appendix of our paper). We will clarify this in the main text.
> * **Experiments on non-conformal maps satisfying IMA.** We remark that the MLP experiment involves non-conformal maps which, while not precisely satisfying the IMA assumption, still reflect the differences between true and spurious solutions predicted by our theory, thus indicating that our proposed theory is also useful for non-conformal maps, and possibly even when the IMA principle is not exactly satisfied. Let us also remark that our experiments on Moebius transformations in dimensionality $>2$ extend previous work on conformal maps in nonlinear ICA, e.g. [37], which only considered dimension $n=2$ (see footnote 8 and lines 265-266 in the paper). Nevertheless, the experiments you suggested can be added to the paper, by replicating part of the results in section 5 with a non-conformal mixing satisfying IMA, e.g. the radial map described in section 3, possibly applied before a Moebius transformation (the composed map still satisfies IMA). We will replicate these and add them for a camera ready version of the paper.
>
> Other minor points:
> * **Would be great to have an intuition for why the Darmois construction yields independent sources.** This is a good suggestion, and something which can be easily added to the paper, for example based on the argument in Section 2.2 of [63]: briefly, by applying a change of variables, we can see (eq. (11) in the aforementioned reference) that the transformed variables are uniformly distributed in the open unit cube, thereby corresponding to independent components. We will add this intuition to the paper.
> * **The presentation of information geometric causal inference is very fast and I could not understand its meaning.** We will remedy this, providing a more detailed presentation either in the main text, exploiting the additional space granted for the camera ready, or in the Appendix.
> * **Line 332: Would be good to have a brief description of the MCC and the Amari metrics.** This is also a good suggestion. We relegated most of the details to the appendices (see e.g. Appendix E.5); wherever possible, we will summarise or move them into the main text.

---

> > ### Comment · Reviewer_tDZV · 2021-08-23
> > **Response**
> >
> > I thank the authors for considering seriously the point I raised in my review. I believe the authors understood my concerns and proposed satisfying solutions to adress them. I will adjust my score to 6 to take into account the future modifications. I won't raise my score further because I believe this work still has issues regarding its framing even by focusing on the information-theoretic interpretation of lines 201-204 and appendix B.3. It is still unclear to me how (8) maps to the ICM principle which states that:
> > "The causal generative process of a system’s variables is composed of autonomous modules that do not inform or influence each other." I do not understand how the orthogonality of the partial derivatives of the mixing can be thought of as some form of "autonomy" or the fact that the partial derivatives "do not inform or influence each other". This connection remains vague. I understand that the ICM principle is meant to be informal, but it should be very clear in which way the formal IMA principle connects to it, and at the moment it is not.

---

> > > ### Author Response · Authors · 2021-08-27
> > > **Thanks and addressing final point**
> > >
> > > Thank you for adjusting your score based on our proposed solutions.
> > >
> > > For completeness, in order to answer your remaining point, we will add a connection between the orthogonality in the IMA principle and ICM in the main text in the “Geometric interpretation” paragraph, after line 212, as follows:
> > > > In the high dimensional setting ($n$ large), this orthogonality can be intuitively interpreted from the ICM perspective as _Nature choosing the direction of the influence of each source component in the observation space independently and from an isotropic prior_. Indeed, it can be shown that the scalar product between two independent isotropic random vectors vanishes as the dimensionality $n$ increases (equivalently: two isotropic vectors of high dimension are typically orthogonal). This property was previously exploited in other linear ICM-based criteria (see [Janzing and Schölkopf, Detecting Confounding in Multivariate Linear Models via Spectral Analysis, Lemma 5] and, in a more general setting, [Janzing et al., Telling cause from effect
> > > based on high-dimensional observations, Lemma 1 and Theorem 1]). Importantly, this has also been used as a _“leading intuition”_ [sic] to interpret IGCI in [Information-geometric approach to inferring causal directions, Janzing et al., Artificial Intelligence 2012]. By enforcing orthogonality between columns of the Jacobian of nonlinear functions at all points in observation space, the IMA principle can be seen as a constraint on the function space that approximates the high-dimensional behavior described above.

---

### Official Review · Reviewer_VVZG · 2021-07-16

**Rating:** 7
**Confidence:** 3

**Summary:**

The paper proposes applying the theory of independent causal mechanisms (ICM), from causal discovery, to nonlinear ICA approaches to the blind source separation problem (BSS).

ICM is based on the idea that variables in a system are algorithmically independent or do not share information with each other. The paper shows, however, that ICM is not sufficient for BSS as spurious solutions exist which are not equivalent to the truth.

To address these difficulties, the paper proposes a new condition for identifiability which places an orthogonality condition on the columns of the Jacobian and provides information theoretic and geometric interpretations. The paper then shows theoretically that a large class of spurious solutions are not admitted under the IMA formulation.

Experiments on toy examples demonstrate that the model is identifiable under these spurious solutions.

**Limitations And Societal Impact:**

The paper discusses limitations

**Main Review:**

In general the paper is well written (though a bit difficult to parse for a non-specialist) and the theoretical contributions are rigorous. I did not check proofs.

I am familiar with the related causal inference literature but less so the ICA literature so it is difficult for me to assess the significance of the class of spurious solutions references in the paper and whether these correspond to real world obstacles or are more of general theoretical interest. If the authors could clarify this, it may influence my final score as it seems this is sort of the crux of whether the paper constitutes a significant advance since IMA is not compared to other ICA approaches in any other way.

I found the experimental results section to be particularly weak as no baselines are included, only the spurious cases are investigated and there are no results on real data. This leaves the reader unclear on several points: (i) whether IMA works similarly in practice to existing ICA approaches (despite the theoretical results), (ii) whether IMA is competitive enough with other ICA approaches on the non-spurious cases to justify the support for the spurious cases, (iii) whether the spurious cases are of practical significance in any real world datasets.

The paper would be significantly improved if other ICA baselines were included as well as experiments on the non spurious cases.

===

I increased my review to 7 based on the discussions and other reviews. See discussion below.

**Time Spent Reviewing:**

5

---

> ### Author Response · Authors · 2021-08-10
> **Response to reviewer VVZG**
>
> Thank you for your time and for reviewing our work. We believe that your main points of criticism might be tied to a potential misunderstanding on the non-identifiability issues arising in nonlinear ICA, which we tried to convey as clearly as we could in Section 2.1. We will try to clarify this in our response, as requested, and will elaborate further on these points in the main paper.
>
> * **Significance of the class of spurious solutions; unclear whether the spurious cases are of practical significance.** The objective of nonlinear blind source separation is to find an unsupervised (“blind”) way to distinguish “good” solutions, which “separate the sources” (the desideratum is formally stated in Definition 2.2), from other solutions that, despite yielding independent components, do not reconstruct the sources up to the allowed ambiguities. There are many solutions of the latter kind: we loosely refer to them as “spurious solutions”. In the literature, existence of spurious solutions is proved constructively: Definitions 2.3 and 2.5 define two classes of spurious solutions which play an important role in such constructive proofs, and feature prominently in the literature [37, 52, 44], also beyond the field of nonlinear blind source separation:  for example, the construction from Definition 2.5 is used to illustrate non-identifiability of fully unsupervised causal discovery in the i.i.d. setting with arbitrary nonlinear relationships, see, e.g., [58], Section 2.1; and [45], Section 3. Furthermore, the same construction is also used to show that autoregressive normalising flows can approximate any density [63, 32].
> Despite their prominent role in the literature, however, these two classes do not necessarily contain _all_ possible spurious solutions: we are in fact not aware of an analytical characterisation of all possible solutions yielding independent components---though see [Taleb and Jutten, Source Separation in Post-Nonlinear Mixtures, 1999], eq. (7), for the differential functional equation they need to satisfy. This makes it particularly hard to derive theoretical statements for the general case, whereas the classes from Definitions 2.3 and 2.5 nicely support theoretical investigation due to their closed form.
> It is hard to assess whether spurious solutions _correspond to_ real world obstacles. We are not aware of empirical studies comparing spurious solutions obtained by unsupervised learning to those in Definitions 2.3 and 2.5, but there are strong indications that they are _predictive of_ real world obstacles: if a model allows such “spurious” solutions, it can typically be expected to be unable to solve blind source separation in practice. For example, in the related field of disentanglement, for example, large-scale empirical study [55] has shown that if Definition 2.5 corresponds to one possible minimum of the learning objective, neural networks trained with said objective actually fail to identify the true latent sources.
> We also note that, beyond these two classes of solutions, our work also _directly_ investigates solutions learned by a class of neural networks (residual flows) in practice. Since these are much harder to describe analytically, we resort to a numerical investigation. In Section 5.2, we show that solutions learned with a classic unsupervised learning objective ($\lambda=0$) fail to solve BSS (are “spurious”); whereas the same neural networks, when trained with a regularisation term enforcing our IMA principle, typically provide “good” solutions.
>
> * **Experimental results section is weak; only the spurious cases are investigated.** We remark that this opinion is not shared by other reviewers e.g., reviewer `ovUQ` found our experiments *“well-designed to support the claims in the paper”*. We believe this may be related to the misunderstanding discussed above (and/or to the fact that the reviewer is, as he/she states, not from the ICA field). The statement *“only the spurious cases are investigated”* is somewhat misleading in the context of nonlinear ICA and blind source separation since **the aim of blind source separation is precisely to distinguish the “good” solutions from the “spurious” ones**. In the first part of our Experiments (section 5.1), we verify our theoretical predictions on the classes of spurious solutions in Definitions 2.3 and 2.5 (also investigating to what extent those predictions hold upon deviations from our assumptions, see the experiment on random MLPs). In the second part (section 5.2), we compare solutions learned by unregularised maximum likelihood estimation ($\lambda=0$), to those learned with a regularised objective based on our theory ($\lambda>0$) ---showing that, typically, the former fail to separate the sources, while the latter lead to “good” BSS solutions.
>
> * **IMA is not compared to other ICA approaches in any other way; no baselines are included.** In ICA, unlike in other unsupervised settings, identifiability is typically considered a prerequisite for developing estimation procedures, see for example [40], Section 1: *“The essential difference [between ICA and] most methods for unsupervised representation learning is that the approach starts by defining a generative model in which the original latent variables can be recovered, i.e., the model is identifiable by design”*.
> As already remarked, nonlinear ICA is unidentifiable: on the one hand, recent identifiability results and the related estimation procedures require additional observed variables or specific time structure in the data [34, 35, 38, 44, 27, 28], and are not applicable to our problem setting. On the other hand, large scale empirical investigations of methods for independent component _estimation_ (that is, methods which transform the observations into independent components, regardless of whether the model class is identifiable or not), verifying that they cannot reconstruct the ground truth sources in an unsupervised way, have already been conducted (e.g. [55] and [Eastwood and Williams, A Framework for the Quantitative Evaluation of Disentangled Representations, ICLR 2018]), and lie outside the scope of our work.
> It is, however, a misunderstanding that we do not include baselines: in fact, models trained to estimate independent components with $\lambda=0$ (qualitatively shown in Figure 4, top, second from the left; and quantitatively characterised in Section 5.2 and Figure 5) _constitute the default baseline_ of non-identifiable, nonlinear independent component estimation methods, and we show that they perform poorly compared to methods enforcing our principle ($\lambda>0$). We nevertheless ran some additional experiments, comparing our proposed regularised maximum likelihood procedure to a state of the art method for linear ICA (FastICA) over $50$ repetitions. Our experiments show that our regularised method ($\lambda=0.5$, and particularly $\lambda=1$; $\lambda=0$ provides the unregularised nonlinear baseline) is superior in learning the true unmixing and reconstructing the sources. This indicates that the linearity assumption does not allow enough flexibility to solve blind source separation in our setting, whereas our criterion does (see plots at the following link for the MCC evaluation: https://ibb.co/YtVBrWf, and this link for the nonlinear Amari distance: https://ibb.co/6bQK8QY). While the spread in the distributions of MCC and Amari distance can be largely attributed to the brittleness of neural networks, the median values for the MCC (resp. nonlinear Amari distance) are consistently higher (resp. lower).
>
> Summarising, **statements such as “only the spurious cases are investigated” are misleading, since distinguishing “spurious” solutions from “good” ones is precisely the aim of nonlinear blind source separation.** The set of all “spurious” solutions has, to the best of our knowledge, not been characterised analytically and in closed form (in the sense that a general form for the unmixing function can not be given): **we therefore develop our main theoretical results based on two prominent classes of spurious solutions which have an analytical form.** Moreover, **we empirically investigate the class of spurious solutions learned by a class of models in Section 5.2, verifying that they don’t solve blind source separation, and contrast them with those learned with a regularisation based on our criterion, which is shown to lead to blind source separation.**

---

> > ### Comment · Reviewer_VVZG · 2021-08-31
> > **Raising score based on other reviews and discussions, still questions**
> >
> > To clarify my previous point about the experiments - I did find that they were well designed to verify the theoretical predictions about the classes of spurious solutions. What I remain unclear about, however, is whether the proposed method represents a significant practical improvement over SotA ICA methods on real problems. This is where I think the experiments are lacking and am still unconvinced. That said, as previously stated I am not familiar with the ICA literature and from reading the other reviews and discussions, the theoretical results seem to be well received by those more familiar so I am raising my review to a 7 given my lack of confidence in my original rating.
> >
> > As the authors suggest in their reply, my concern may be based on the fact that I am not from the ICA field, but I think the concern should still be considered when the authors make their revisions so the paper has more appeal to and presents less confusion to non-experts in this specific subfield. In other areas of ML, it is not uncommon for a paper to propose a new solution with better theoretical properties than the SotA, but when applied to a wide range of practical problems performs comparably or even worse. When I read the paper I did not find any results to convince me this was not the case and because I am a non-expert in ICA it was difficult for me to judge the significance of the theoretical results. If the authors could clarify and make this question more accessible in their revisions, I think the paper would be improved and appeal to a larger audience.

---

> > > ### Author Response · Authors · 2021-09-02
> > > **Thank you and answering your final questions**
> > >
> > > Thank you for taking the other reviews and the discussion into account in your revised evaluation. We will do our best to incorporate your feedback for the revised manuscript, and address the concerns you raised.
> > >
> > > In our view, there are two aspects in the ICA problem, that is: (1) identifiability and (2) estimation. **Practical performance critically depends on the estimation algorithm**, and **once a setting is shown to be identifiable, different estimation methods can be developed and meaningfully compared**. For example, general identifiability for nonlinear ICA with auxiliary variables was initially formulated in [Hyvarinen et al., Nonlinear ICA Using Auxiliary Variables and Generalized Contrastive Learning, 2018], and tied to an estimation procedure based on contrastive learning. Subsequently, a related identifiability proof was given for a VAE based estimation procedure [Khemakhem et al., Variational Autoencoders and Nonlinear ICA: A Unifying Framework, 2020] and, more recently, identifiable energy-based models [Khemakhem et al., ICE-BeeM: Identifiable Conditional Energy-Based Deep Models, 2020] were shown to be competitive or better than previous methods on some tasks.
> > >
> > > Inspired by your review, we will include experiments (described in our rebuttal) comparing our proposed estimation procedure with a SoTA linear ICA method (FastICA), and showing that our method is superior when the assumptions are met. While said synthetic setting allows for a direct comparison, which would not be as straightforward in a practical scenario with unknown ground truth, we agree that superior performance in a controlled synthetic setting does not directly imply that the same would hold in a non-synthetic task: this is an important question, additionally requiring the design of reliable and application-dependent evaluation metrics or downstream tasks.

---

### Official Review · Reviewer_ovUQ · 2021-07-22

**Rating:** 7
**Confidence:** 3

**Summary:**

The paper proposes a criterion for non-linear ICA as applied to Blind Source Separation inspired by Independence of Causal Mechanisms in the causal discovery literature. The general idea is to constrain the gradients of the output with respect to the input sources to be orthogonal to each other; this is linked to the notion that the mixing mechanism is in some sense independent (or not "too fine-tuned") to the input sources. The authors show that this constraint eliminates a number of known counterexamples to identifiability in non-linear ICA, and test it experimentally to show that helps to characterize true solutions (both as an evaluation metric, and as a regularizer). The authors conclude with a discussion putting their contributions into context.

**Ethical Concerns:**

None.

**Limitations And Societal Impact:**

The authors do a good job of noting exactly what they establish regarding IMA in the paper---that their measure eliminates certain counterexamples---and what they do not---full identifiability under the IMA restriction.

**Main Review:**

Overall, the paper is clearly written and the technical properties and limitations of the method are well-discussed. The experiments are well-designed to support the claims in the paper.

My main concern with the paper is that the discussion of ICM as motivation for IMA is a bit long and imprecise. While I do appreciate the review of ICM ideas, the notion of "algorithmic independence" is still rather vague (this is, IMO, often a problem in a lot of the ICM literature). Although two different measurements of the concept are given (in terms of Kolmogorov complexity and IGCI), a concrete counterexample of what it would mean to have "fine-tuning" between the cause and the conditional would be helpful. In particular, is there an example that could be given (e.g., a modification of the cocktail party example) where this independence would not hold? Perhaps, an operator might have changed some properties of the recording to better capture certain parts of the input sources?

A concrete example of this type would also give a better sense of when this functional restriction may or may not be appropriate in practice. When specifying identifying assumptions, this kind of grounding is generally useful because, by definition, there is no way to test these assumptions in practice.

**Time Spent Reviewing:**

2 hours

---

> ### Author Response · Authors · 2021-08-10
> **Response to reviewer ovUQ**
>
> We thank you for your words of praise, and for your overall positive evaluation of our paper. We will answer to the points you raised and, following your suggestions, provide further explanation on the ICM and IMA principles, in terms of both technical examples and intuitive explanations.
>
> * **The discussion of ICM as motivation for IMA is a bit long and imprecise.**
> As stated in the answer to all reviewers, we acknowledge that the algorithmic independence formalisation of ICM might feature too prominently in the paper: our IMA principle can be more precisely linked to the IGCI formulation in the context of cause-effect pair inference---where the dependence between cause and mechanism can be conceived as a fine tuning between the derivative of the mechanism and the input density, as illustrated in Figure 2(a) ---and we will stress this more in the paper. The IMA principle leads to a complementary, non-statistical measure of independence between the influences of the individual sources on the vector of observations (defined as the partial derivatives $\partial \mathbf{f}/\partial s_i$) that is useful in the context of nonlinear blind source separation. This connection is described in technical terms in Appendix B.3. In short, **both the IGCI and IMA postulates have an information geometric interpretation related to the influence of “non-statistically” independent factors on the observations**. While for IGCI independent factors are the cause and the mechanism mapping the cause to the effect, for IMA these are the influences of each source component on the observations in an interventional setting. Specifically, the effect of component-wise soft interventions on each source, as measured by the KL divergence between the original and the intervened distributions, decomposes into the sum of the individual KL-divergences, see equation (21).
>
> * **A concrete counterexample of what it would mean to have "fine-tuning" between the cause and the conditional would be helpful.** We will provide two different counterexamples, distinguishing what it means to have “fine-tuning” according to ICM formulations from causal discovery and according to our IMA principle.
> 1. In the context of the Trace method [41], used in causal discovery, a technical example of fine-tuning can be constructed by taking a vector of i.i.d. random variables with arbitrary (not diagonal) covariance matrix $\Sigma$ as the cause, and by constructing the mechanism as a whitening matrix, turning the cause variables into uncorrelated (effect) variables. By doing so, the singular values and singular vectors of the matrix (the mechanism) are fine-tuned to the input covariance matrix (a property of the cause distribution) in order to decorrelate the signals, and such fine-tuning can be quantified via the Trace method (see [41], Section 1). However, as explained in Appendix B.1, this criterion (and related ones, such as the one discussed in Section 3) is insufficient for nonlinear blind source separation, where the cause is not observed---which motivates our introduction of the IMA principle.
> 2. An example of a mixing function which is non-generic according to the IMA principle is an autoregressive function, for example an autoregressive normalising flow, where the $k$-th component of the observations only depends on the $1:k$-th sources: intuitively, this would correspond to the unlikely cocktail-party setting where the $k$-th microphone only picks up the voices of the first $k$ speakers. More precisely, as we show in the Appendix, Lemma C.1, this leads to positive $C_{\operatorname{IMA}}$ value for such mixing.
> * **Modification of the cocktail party example where the IMA principle would not hold.** A cocktail party may violate our IMA principle when the location of several speakers' and the room acoustics have been fine tuned to one another. This is for example the case in concert halls where the acoustics of the room have been fine-tuned to the position and configuration of multiple locations on the stage, where the sources (i.e., the voices of the actors or singers) are emitted---in order to make the listening experience as homogeneous as possible across the spectators (that is, the influence of each of the sources on the different listeners should not differ too much). This would lead to an increase in collinearity between the columns of the mixing’s Jacobian, thus violating the IMA principle. Additionally, we recall that the ICM principle is often informally introduced by referencing the fine-tuning and non-generic viewpoints giving rise to certain visual illusions, such as the Beuchet chair (see [70], Section 2); in a similar vein, we can imagine that violations of the IMA principle in the cocktail party setting may be related to illusions in binaural hearing such as for example the Franssen effect, where the listener is tricked into incorrectly localizing a sound [Schroeder, "Listening with two ears”, Music perception,1993].
>
> As a minor comment, we point out a typo in your Summary. The sentence *“the general idea is to constrain the gradients of the output with respect to the input sources to be orthogonal to each other”* should read: *“the general idea is to constrain the vectors of partial derivatives of the mixing function $\mathbf{f}$ with respect to the each source $s_i$ to be orthogonal”* (our condition is over the Jacobian’s columns, whereas the gradients are the rows).

---

> > ### Comment · Reviewer_ovUQ · 2021-08-30
> > **Thanks for the response!**
> >
> > I appreciate the response to my review. I particularly like the concrete examples of the fine-tuning "corner cases" eliminated by the IMA principle in the context of the cocktail party example. I think these could really clarify some of the early motivation in the paper. My rating remains the same, and high.
> >
> > I also appreciate the correction to my summary!

---

### Author Response · Authors · 2021-08-10
**Response to all reviewers**

We thank all reviewers for their thoughtful comments and suggestions.

We are pleased to read that the reviewers found the paper *“original and of great significance”* (`hbzz`); that  *“the theoretical contributions are rigorous”* (`VVZG`), and *“the method is properly introduced [and] the theoretical analysis and examples are appreciated”* (`989y`); that it presents *“new findings in [an] interesting direction”* (`tDZV`); that *“its technical properties and limitations are well-discussed”* and *“the experiments are well-designed to support the claims in the paper”* (`ovUQ`); and that overall it *“was a joy to read”* (`hbzz`).


A concern shared by some reviewers (`tDZV`, `VVZG`, `ovUQ`) relates to the connection between existing formalisations of the principle of independent mechanisms---in particular the one based on algorithmic independence---and our own IMA principle. This is a subtle and important point, concerning the interpretation of the IMA principle and, more broadly, of the ICM principle. In the paper, we often referred to the formulation of ICM in terms of algorithmic independence, particularly in the Introduction and in Sections 2.1 and 4.1: following the suggestions of reviewers `ovUQ` and `tDZW`, we will instead emphasize more the connection to Information Geometric Causal Inference (IGCI) ---whose link to our own IMA principle can be made precise and expressed in technical terms, as also discussed in Appendix B.3.

At the same time, the originality and technical soundness of our work were praised, with reviewer `hbzz` stating that *“learning identifiable representations is an important topic in unsupervised learning, and the provided framework is an important step in this direction”*, and indicating that our work can be a fruitful stepping stone for further cross-pollination between the independent component analysis and causal inference communities---as we also detail at lines 375-378 in the paper.

In an effort to add more baselines and comparison, and answering questions from reviewers `VVZG` and `989y`, we performed experiments to compare our proposed regularised maximum likelihood procedure to a state of the art method for linear ICA (FastICA), showing that our regularised method ($\lambda=0.5$, and particularly $\lambda=1$; $\lambda=0$ provides the unregularised nonlinear baseline) is superior in learning the true unmixing and reconstructing the sources, and thus indicating that the linearity assumption does not allow enough flexibility to solve blind source separation in our setting, whereas our criterion does (see plots at the following link for the MCC evaluation: https://ibb.co/YtVBrWf , and this link for the nonlinear Amari distance: https://ibb.co/6bQK8QY ; more details in the individual replies to reviewers `989y` and `VVZG`).

Throughout our answers, we will refer to works in the literature through the numbering applied in the References list of our main manuscript; or, whenever citing papers not already present in our References list, by first author, title and year.

---

### Decision · Program_Chairs · 2021-09-27

**Decision:**

Accept (Poster)

**Comment:**

The authors propose a novel method for non-linear ICA. The idea is to constrain partial derivatives of the mixing function with respect to each source to be orthogonal. The authors connect this idea to the principle of independent mechanisms. The authors show that the constraint allows to deal with known counterexamples to identifiability in non-linear ICA. The method is evaluated on simulated toy examples. Reviewer hbzz praised the originality and significance of the work. Several reviewers (tDZV, ovUQ, 989y) criticized that the connection of IMA to other principles (such as algorithmic independence and information geometric causal inference) is vague or difficult to understand. In the responses, the authors expanded on the connections between the principles and proposed to de-emphasise the connection to algorithmic independence, while emphasizing the connection to information geometric causal inference. Several reviewers (VVZG, ovUQ, hbzz) agree that the paper is well-written. Overall, the authors have addressed the main concerns in the discussion.